# SpectraLLM: Uncovering the Ability of LLMs for Molecular Structure Elucidation from Multi-Spectral Data

**Yunyue Su[1]\*, Zao Jiang[2], Zhenyi Zhong[3], Liang Wang[1],**
**Qiang Liu[1]†, Zhaoxiang Zhang[1]**

[1]New Laboratory of Pattern Recognition, Institute of Automation, Chinese Academy of Sciences
[2]The Hong Kong Polytechnic University
[3]Tianjin University Peiyangyuan Campus

## Abstract

Automated molecular structure elucidation remains challenging, as existing approaches often depend on pre-compiled databases or restrict themselves to single spectroscopic modalities. Here we introduce **SpectraLLM**, a large language model that performs end-to-end structure prediction by reasoning over one or multiple spectra. Unlike conventional spectrum-to-structure pipelines, SpectraLLM represents both continuous (IR, Raman, UV-Vis, NMR) and discrete (MS) modalities in a shared language space, enabling it to capture substructural patterns that are complementary across different spectral types. We pretrain and fine-tune the model on small-molecule domains and evaluate it on four public benchmark datasets. SpectraLLM achieves state-of-the-art performance, substantially surpassing single-modality baselines. Moreover, it demonstrates strong robustness in unimodal settings and further improves prediction accuracy when jointly reasoning over diverse spectra, establishing a scalable paradigm for language-based spectroscopic analysis. Code is available at https://github.com/OPilgrim/SpectraLLM.

## 1 Introduction

Structure elucidation is fundamental across chemistry, biology, and materials science, enabling the determination of molecular and crystal structures (Shklover et al., 2021; Ali et al., 2022; Beermann & Brockamp, 2005; Als-Nielsen & Materlik, 1995; Ishchenko et al., 2011; Filipponi et al., 1995; Le et al., 2024; Fang et al., 2024; Ju et al., 2025). A wide array of spectroscopic techniques, including infrared (IR), Raman, ultraviolet-visible (UV-Vis), nuclear magnetic resonance (NMR), and mass spectrometry (MS), have become indispensable tools for this purpose, each probing distinct physicochemical properties through interactions with electromagnetic radiation or ionization (Stuart, 2004; Smith, 2018; Butler et al., 2016; Perkampus, 2013; Picollo et al., 2019; Bovey et al., 1988; James, 2012; Quinn et al., 2020). Table 1 summarizes their detection mechanisms, sensitivities, and characteristic outputs. Taken together, these modalities provide complementary molecular insights, and human experts routinely rely on their joint interpretation to resolve structural ambiguities.

Machine learning has emerged as a promising approach to automate spectrum-to-structure inference, showing effectiveness in functional group recognition (Fine et al., 2020; Judge et al., 2008; Klawun & Wilkins, 1996; Fessenden & Györgyi, 1991; Hemmer & Gasteiger, 2000) and molecular property regression from spectral data (Wang et al., 2022; Chen et al., 2023). Early work has primarily adopted conventional architectures such as convolutional neural networks (LeCun et al., 2002; Li et al., 2021; O'shea & Nash, 2015) and Transformers (Vaswani et al., 2017; Han et al., 2022; 2021), framing spectral analysis as a sequence-to-sequence task. Examples include Spec2Mol (Litsa et al., 2023), which encodes MS/MS spectra using a CNN and decodes SMILES via an RNN, as well as subsequent efforts leveraging Transformer decoders (Hu et al., 2023; Kanakala et al., 2024). Other

---

\*Yunyue Su and Jiahui Chen contributed equally to this work.
†Corresponding author: qiang.liu@nlpr.ia.ac.cn

Table 1: Overview of spectroscopic modalities.

| Modality | Detection Principle | Structural Sensitivity | Common Output Features |
|---|---|---|---|
| IR | Molecular vibration absorption | Functional groups with dipole moment changes | Peak positions, intensities |
| Raman | Inelastic light scattering | Symmetric bonds and polarizability changes | Shifted peak patterns |
| UV-Vis | Electronic transitions | Conjugated systems, $\pi - \pi*$ and $n - \pi*$ transitions | Absorbance wavelengths, spectra |
| $^{13}$C NMR | Carbon-13 spin resonance | Carbon backbone structure, hybridization | Chemical shifts of $^{13}$C nuclei |
| $^1$H NMR | Proton spin resonance (1H) | Proton environments, local chemical shifts | Peak multiplicity, integration |
| HSQC NMR | $^{13}$C-$^1$H correlation spectroscopy | Coupling between proton and carbon atoms | 2D cross-peaks ($^{13}$C-$^1$H pairs) |
| Mass (MS) | Mass-to-charge ratio (m/z) | Molecular weight and fragmentation patterns | m/z peaks |

extensions combine molecular formulas with spectra (Alberts et al., 2024a), or achieve reconstruction from IR alone without expert rules (French et al., 2025). Recently, diffusion-based approaches such as DiffMS (Bohde et al., 2025) have framed inverse mass spectrometry as a conditional molecular generation task, improving both diversity and synthesizability. Despite these advances, most existing models remain confined to single modalities and lack the flexibility to integrate heterogeneous spectroscopic evidence.

A key limitation of these approaches lies in their architectural rigidity: spectra are treated as fixed numerical features, which hinders the incorporation of additional modalities or contextual information such as experimental conditions and annotations. In contrast, large language models (LLMs) offer a natural foundation for this problem. By transforming spectral peaks into textual representations that encode physical attributes, LLMs can unify heterogeneous modalities within a shared semantic space, enabling joint reasoning over diverse spectroscopic signals. This paradigm leverages LLMs' strengths in symbolic reasoning, contextualization, and compositional generalization, which conventional architectures cannot easily replicate, which motivates our design of SpectraLLM.

In this study, we present SpectraLLM, a large language model for molecular structure elucidation from spectroscopic data. Unlike previous modality-specific approaches, SpectraLLM employs a unified language-based architecture that accepts structured descriptions of one or more spectral modalities and directly infers molecular structures via natural language reasoning. Although we initially experimented with vision-language models (VLMs), their performance was not as good as the language-centered approaches (Appendix A.5), which is likely due to the cross-modal encoding artifacts. By leveraging LoRA-based adaptation on top of a frozen foundation model, SpectraLLM aligns spectral features with linguistic priors while maintaining scalability. Our systematic evaluation across four public benchmark datasets demonstrates that SpectraLLM achieves state-of-the-art performance, outperforming both traditional pipelines and recent multimodal transformer-based baselines. Notably, the model exhibits strong generalization across modality combinations: it supports inference from single and multiple spectra, with accuracy improving monotonically as spectral diversity increases.

The main contributions of this work are as follows:

- We propose a language-based model for molecular structure elucidation that performs symbolic reasoning over multi-spectral data.

- By expressing spectral peaks and experimental conditions in natural language, the model supports both single-spectrum and multi-spectral inputs, generalizing effectively across combinations of spectral data, benefiting from spectral complementarity.

- SpectraLLM achieves state-of-the-art performance on four public chemical datasets, showing strong robustness in unimodal settings and substantial gains when jointly reasoning over multiple spectra.

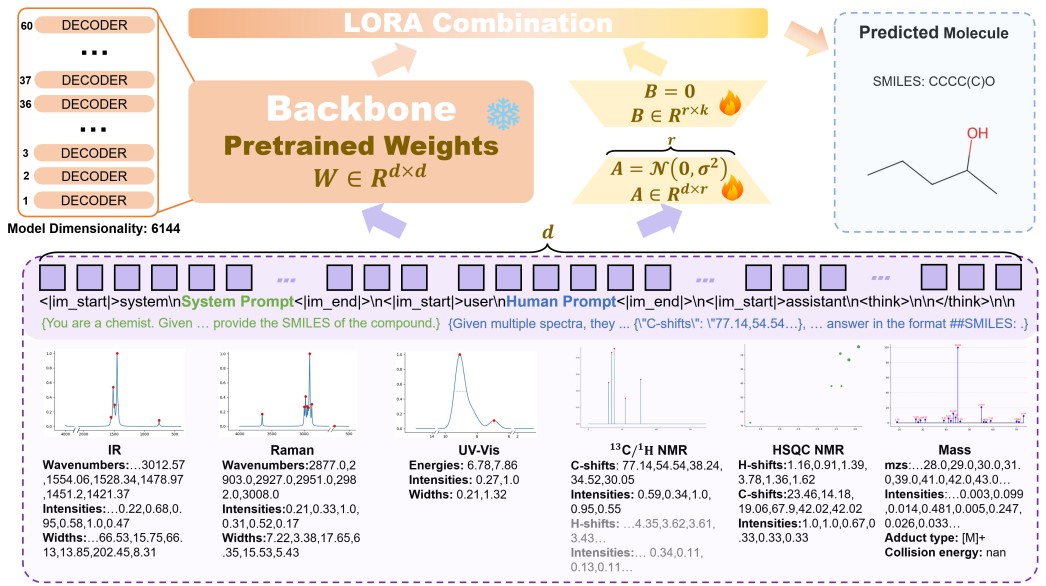

Figure 1: Overview of the training pipeline for structure elucidation. Characteristic spectral peaks are extracted from raw IR, Raman, UV, NMR, or MS data and used to construct natural language prompts. These are input to a frozen large language model fine-tuned via LoRA. The model is trained to autoregressively generate molecular structures in SMILES format, supervised by the ground-truth sequence.

## 2 SEMANTIC-DRIVEN SPECTROSCOPIC REASONING

### 2.1 PRELIMINARIES

We consider the task of molecular structure elucidation from spectroscopic data. Formally, let $\mathbf{s} = \{s_1, s_2, ..., s_k\}$ denote a set of one or more spectra corresponding to the same molecule. Each spectrum $s_i$ arises from a modality $m_i \in \{\text{IR}, \text{Raman}, \text{UV}, \text{NMR}, \text{MS}\}$. A single spectrum can be described as a finite set of peak observations, $s_i = \{(p_j, a_j)\}_{j=1}^{n_i}$, where $n^{(i)}$ is the number of peaks, $p_j$ denotes a physical measurement characteristic of the modality (such as wavenumber, chemical shift, or mass-to-charge ratio), and $a_j$ is the corresponding normalized intensity or abundance.

The objective is to infer the molecular structure, represented as a SMILES sequence $y = (y_1, y_2, \ldots, y_T)$, where $T$ is the number of tokens in the sequence. This task is cast as an autoregressive modeling problem, in which the conditional probability of generating a sequence given the spectra factorizes as

$$p_\theta(y|\mathbf{s}) = \prod_{t=1}^{T} p_\theta(y_t|\mathbf{s}, y_{<t}), \tag{1}$$

where $y_{<t}$ denotes the prefix of previously generated tokens, and $\theta$ are the model parameters.

Given a training corpus of $N$ paired examples $(\mathbf{s}^{(i)}, y^{(i)})_{i=1}^{N}$, where $\mathbf{s}^{(i)}$ is the spectroscopic input and $y^{(i)}$ is the corresponding SMILES sequence, the model is trained to minimize the negative log-likelihood of the reference sequences. This corresponds to the cross-entropy objective:

$$\mathcal{L}_{\text{CE}} = -\sum_{i=1}^{N} \sum_{t=1}^{T} \log p_\theta(y_t^{(i)}|\mathbf{s}^{(i)}, y_{<t}^{(i)}). \tag{2}$$

## 2.2 JOINT FINE-TUNING OF MULTIPLE SPECTRA

### 2.2.1 CORPUS CONSTRUCTION AND PREPROCESSING

A prerequisite for joint spectroscopic reasoning is a sufficiently large and diverse corpus of molecules annotated with multiple types of spectral data. To this end, we combined four complementary sources spanning simulated and experimental data. The QM9spectra (QM9s) dataset (Zou et al., 2023) provided simulated IR, Raman, and UV-Vis spectra for approximately 134,000 small organic molecules, generated by frequency analysis and time-dependent density functional theory at the B3LYP/def2-TZVP level. The Multimodal Spectroscopic dataset (Alberts et al., 2024b) extends this coverage with simulated IR, MS, and multiple NMR spectra, including $^1$H, $^{13}$C, and HSQC correlations, for over 790,000 molecules extracted from USPTO reactions (Lowe, 2012). For experimental validation, we included MassSpecGym (Bushuiev et al., 2024) with 230,000 high-resolution spectra across 29,000 compounds, and MassBank (Horai et al., 2010), which contributed additional MS and MS/MS spectra obtained under varied instrument conditions. Together, these resources contribute more than 5.5 million spectra paired with over 940,000 unique molecules, forming one of the largest corpora for end-to-end structure elucidation. Table 5 summarizes their composition.

Raw spectra were then processed to produce compact, interpretable peak-level features. For vibrational and electronic spectra, IR and Raman signals were truncated to the 500–4000 cm$^{-1}$, region and uniformly sampled, while UV-Vis spectra were truncated to 1.0–15.0 eV at 0.02 eV intervals. Intensities were normalized by their maximum value, and characteristic peaks were extracted using local maxima and prominence thresholds. NMR spectra were processed by cropping to the minimal region containing significant peaks and discarding signals below one percent of the maximum intensity. For $^1$H and $^{13}$C spectra, chemical shifts and intensities were retained, while HSQC spectra preserved two-dimensional correlations between proton and carbon shifts along with estimated proton counts. Mass spectra were standardized by rounding m/z values to two decimals, scaling intensities into the range [0,100], and preserving auxiliary metadata such as ionization mode, collision energy, and instrument type. To support downstream model training, all spectra were ultimately represented as structured arrays of peaks and intensities, serialized in JSON-like format. Appendix A.2 contains more details for processing.

### 2.2.2 LANGUAGE-BASED FORMULATION OF SPECTRA

To unify heterogeneous spectra within a single modeling framework, we transformed peak arrays into natural language descriptions. A mapping function $\phi$ converts each spectrum $s_i$ into a textual prompt $\phi(s_i) \in \mathcal{V}^n$, where $\mathcal{V}$ denotes the model vocabulary. Multiple spectra can be jointly encoded via concatenation:

$$x = \phi(s_1) \oplus \phi(s_2) \oplus \cdots \oplus \phi(s_k). \tag{3}$$

Prompts explicitly specify modality and report salient peaks in textual form, thereby embedding vibrational frequencies, chemical shifts, and fragmentation patterns into the same linguistic space. In addition to peak values, we also incorporate relevant experimental conditions and metadata, such as ionization mode, collision energy, solvent, and instrument type, so that the model can contextualize spectra under varying acquisition settings. Training examples are cast in an instruction-response format: the `"Human"` side presents the spectroscopic description and the query to infer structure, while the `"GPT"` side provides the reference SMILES. This formulation enables symbolic reasoning over complementary spectroscopic cues, bridging continuous and discrete modalities.

### 2.2.3 AUTOREGRESSIVE BACKBONE AND PARAMETER-EFFICIENT FINE-TUNING

As illustrated in Fig. 1, we adopt Qwen3 (Yang et al., 2025) as the backbone model, chosen for its scalability and strong reasoning performance. To specialize it for spectroscopy-to-structure generation, we employ Low-Rank Adaptation (LoRA) (Hu et al., 2022), freezing the backbone weights while introducing lightweight trainable matrices in each transformer layer. This parameter-efficient tuning preserves the general linguistic competence of the model while enabling effective alignment with spectroscopic inputs.

Given a prompt $x$ derived from one or more spectra and a reference SMILES sequence $y = (y_1, \ldots, y_T)$, the model is optimized using the autoregressive cross-entropy loss introduced in Sec. 2.1. This objective encourages the model to maximize the conditional likelihood $p_\theta(y_t|x, y_{<t})$

Table 2: Comparative evaluation of SpectraLLM and conventional approaches under individual spectral inputs.

| Spectrum | Method | Validity ↑ | Tanimoto ↑ | Cosine ↑ | MCES ↓ | Functional Group ↑ | Tanimoto (MACCS) ↑ | Fraggle ↑ |
|---|---|---|---|---|---|---|---|---|
| | | | | | **QM9s** | | | |
| | IR-to-Structure | 100.00% | 0.0718 | 0.1311 | 11.3187 | 0.3151 | 0.1585 | 0.1747 |
| IR | Spectra2Structure | 100.00% | 0.0965 | 0.1695 | 10.1081 | 0.4383 | 0.2162 | 0.2308 |
| | SpectraLLM | 99.82% | **0.1921** | **0.3120** | **7.5651** | **0.6599** | **0.4330** | **0.3194** |
| | IR-to-Structure | 100.00% | 0.0766 | 0.1395 | 11.3516 | 0.3525 | 0.1639 | 0.1959 |
| Raman | Spectra2Structure | 100.00% | 0.1089 | 0.1901 | 9.4164 | 0.4419 | 0.2388 | 0.2504 |
| | SpectraLLM | 99.08% | **0.2500** | **0.3786** | **6.4076** | **0.7317** | **0.5071** | **0.2500** |
| | IR-to-Structure | 100.00% | 0.0728 | 0.1326 | 11.424 | 0.3151 | 0.1512 | 0.1837 |
| UV-Vis | Spectra2Structure | 100.00% | 0.0716 | 0.1313 | 11.1222 | 0.3901 | 0.1418 | 0.2092 |
| | SpectraLLM | 100.00% | **0.0790** | **0.1426** | **10.6374** | **0.3713** | **0.2026** | **0.2100** |
| | | | | | **Multimodal Spectroscopic** | | | |
| | NMR2Struct | 47.62% | 0.0433 | 0.1029 | 30.6938 | 0.1718 | 0.1294 | 0.0962 |
| NMR | SpectraLLM | 98.92% | **0.4151** | **0.5322** | **8.3091** | **0.7209** | **0.6367** | **0.5862** |

for each token $y_t$, with $\theta$ denoting the LoRA parameters. By training over millions of paired examples, the model learns to map spectroscopic patterns into chemically valid and structurally faithful sequences.

### 2.2.4 INFERENCE AND DECODING

During inference, one or more spectra are preprocessed, converted into natural language prompts, and fed into the fine-tuned model. Candidate SMILES strings are generated autoregressively:

$$\hat{y} = \arg\max_{y} p_\theta(y|x) \tag{4}$$

We employ nucleus sampling with threshold $p = 0.7$ and temperature scaling $\tau$ to balance determinism and diversity. As detailed in Appendix A.6, moderate temperatures ($\tau = 0.4$) yield the best trade-off between structural accuracy and output diversity. Notably, no modality-specific encoders, handcrafted rules, or retrieval components are introduced, ensuring a fully end-to-end generative process.

## 3 EXPERIMENTS

### 3.1 EXPERIMENTAL SETUP

We evaluate SpectraLLM across multiple types of spectra and benchmark datasets. Following the preprocessing described in Sec. A.2, we constructed training and evaluation sets from QM9s, the Multimodal Spectroscopic dataset, MassSpecGym, and MassBank, covering simulated and experimental IR, Raman, UV-Vis, NMR, and MS spectra.

Model performance is assessed using a suite of complementary metrics that quantify both exact structure recovery and chemically meaningful similarity. Specifically, we report functional group overlap, fingerprint-based Tanimoto coefficients (ECFP4 and MACCS), cosine similarity, maximum common edge substructure (MCES), and Fraggle similarity. These criteria jointly capture correctness at the level of global structure, local substructures, and functional groups. Full definitions and calculation details are provided in Appendix A.3.

For baseline comparison, we include representative state-of-the-art spectrum-to-structure models across different modalities: DiffMS (Bohde et al., 2025) and Spec2Mol (Litsa et al., 2023) for mass spectrometry, IR-to-Structure (Alberts et al., 2024a) and Spectra-to-Structure (Kanakala et al., 2024) for vibrational and electronic spectra, and NMR2Struct (Hu et al., 2024) for NMR-based prediction. Detailed descriptions of these baselines, including training configurations and evaluation protocols, are given in Appendix A.4.

### 3.2 STATE-OF-THE-ART PERFORMANCE OF SPECTRALLM

To provide a direct comparison with established spectrum-to-structure approaches, we evaluated SpectraLLM under unimodal inputs across IR, Raman, UV-Vis, NMR, and MS. As shown in Table 2

Table 3: Benchmarking SpectraLLM against established mass spectrometry-based inference models.

| Method | Validity ↑ | Tanimoto ↑ | Cosine ↑ | Functional Group ↑ | Tanimoto (MACCS) ↑ | Fraggle ↑ |
|---|---|---|---|---|---|---|
| **MassSpecGym** | | | | | | |
| Spec2Mol | 62.86% | 0.0849 | 0.1511 | 0.3111 | 0.2709 | 0.2065 |
| Diffms | 57.16% | **0.1597** | 0.2422 | 0.4890 | 0.4305 | 0.3539 |
| SpectraLLM | 99.74% | 0.1533 | **0.2558** | **0.5003** | **0.4723** | **0.3610** |
| **Multimodal Spectroscopic** | | | | | | |
| Spec2Mol | 75.39% | 0.0988 | 0.1739 | 0.3042 | 0.2440 | 0.2587 |
| Diffms | 78.77% | 0.1535 | 0.2351 | 0.4248 | 0.3730 | 0.3635 |
| SpectraLLM | 99.64% | **0.1844** | **0.2993** | **0.4929** | **0.4254** | **0.4282** |
| **MassBank** | | | | | | |
| Spec2Mol | 71.63% | 0.0857 | 0.0006 | 0.2999 | 0.1539 | 0.1102 |
| Diffms | 23.63% | 0.0742 | 0.2088 | 0.1795 | 0.1007 | 0.0238 |
| SpectraLLM | 98.44% | **0.1286** | **0.2229** | **0.4539** | **0.3787** | **0.3150** |

and Table 3, SpectraLLM consistently outperforms existing baselines in both predictive accuracy and structural fidelity.

On the QM9s dataset, SpectraLLM surpasses traditional spectrum-to-structure pipelines across all three unimodal conditions. With IR spectra alone, the model achieves a Tanimoto similarity of 0.1921 and a cosine similarity of 0.3120, more than doubling the scores of previous neural baselines, while simultaneously lowering MCES from over 10 to 7.5651, reflecting improved substructural alignment. The gains are even more pronounced for Raman spectra: SpectraLLM reaches a Tanimoto similarity of 0.2500 and cosine similarity of 0.3786, outperforming the strongest baseline by over 30%, and achieving the highest functional group recovery (0.7317). Even with UV-Vis spectra, which are intrinsically less structurally informative, SpectraLLM maintains superior performance, with modest but meaningful improvements in Tanimoto similarity (0.0790) and functional group recovery (0.3713). These results indicate that the model can extract structural cues even from sparse or globally delocalized modalities, highlighting its robustness under information-limited conditions.

The advantage of SpectraLLM becomes even clearer in the NMR domain. On the Multimodal Spectroscopic dataset, the model achieves a Tanimoto similarity of 0.4151 and a cosine similarity of 0.5322, compared to 0.0433 and 0.1029 for NMR2Struct. MCES decreases dramatically from 30.6938 to 8.3091, underscoring the improved recovery of substructural scaffolds. Moreover, functional group accuracy rises from 0.1718 to 0.7209, reflecting the model's ability to map NMR patterns onto interpretable chemical substructures with far greater reliability than prior specialized models. Importantly, validity remains near 99%, eliminating the decoding instability observed in traditional architectures.

Mass spectrometry presents a distinct challenge owing to its fragmented and context-dependent signal. Across the three evaluation settings—MassBank, MassSpecGym, and the Multimodal Spectroscopic benchmark—SpectraLLM consistently delivers state-of-the-art performance while preserving near-perfect structural validity (Table 3). On MassBank, SpectraLLM substantially improves all similarity metrics relative to baseline models, including a functional-group recovery of 0.4539 and a MACCS-based Tanimoto of 0.3787, outperforming both Spec2Mol and Diffms by sizeable margins. Moving to MassSpecGym, which expands upon MassBank with additional curated spectra and molecular diversity, SpectraLLM maintains this strong performance profile: it attains 99.74% validity, the highest cosine similarity (0.2558), and the strongest functional-group recovery (0.5003), while achieving Tanimoto similarity (0.1533) competitive with Diffms (0.1597). Finally, on the Multimodal Spectroscopic dataset, SpectraLLM again obtains the best overall structural alignment, with the highest Tanimoto similarity (0.1844), cosine similarity (0.2993), and functional-group accuracy (0.4929). Collectively, these results highlight SpectraLLM's capacity to generalize across diverse MS conditions while consistently surpassing specialized architectures on detailed structural-similarity metrics.

Taken together, these results demonstrate the breadth and flexibility of SpectraLLM. Even when restricted to a single spectroscopic modality, the model extracts chemically meaningful signals and consistently surpasses the strongest baselines in both structural similarity and functional group recovery. This robustness under unimodal constraints not only underscores the strength of its linguistic

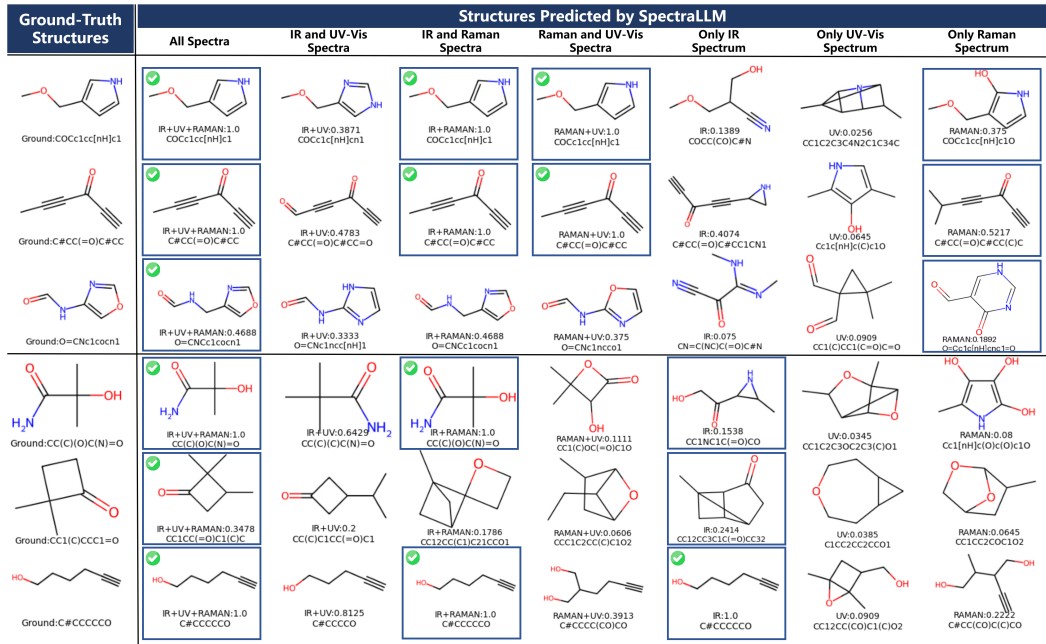

Figure 2: Case studies illustrating complementary roles of Raman and IR spectra in joint structure elucidation. Top: Three representative examples where incorporating Raman spectra corrects mispredictions made using only IR or UV-Vis inputs, reflecting Raman's sensitivity to polarizability-dependent substructures. Bottom: Three representative examples where IR spectra are indispensable for identifying carbonyl groups and resolving branched chain configurations; without IR input, predictions based on Raman and UV-Vis remain ambiguous or incorrect.

Table 4: Enhanced structure elucidation through fusion of complementary spectroscopic modalities.

| Inputs | Validity ↑ | Tanimoto ↑ | Cosine ↑ | MCES ↓ | Functional Group ↑ | Tanimoto (MACCS) ↑ | Fraggle ↑ |
|---|---|---|---|---|---|---|---|
| **QM9s** | | | | | | | |
| IR | 99.82% | 0.1921 | 0.3120 | 7.5651 | 0.6599 | 0.4330 | 0.3194 |
| Raman | 99.08% | 0.2500 | 0.3786 | 6.4076 | 0.7317 | 0.5071 | 0.2500 |
| UV-Vis | 100.00% | 0.0790 | 0.1426 | 10.6374 | 0.3713 | 0.2026 | 0.2100 |
| Jointly | 98.72% | **0.3355** | **0.4560** | **4.9647** | **0.7934** | **0.5785** | **0.4117** |
| **Multimodal Spectroscopic** | | | | | | | |
| IR | 99.63% | 0.1720 | 0.2868 | 15.3234 | 0.6023 | 0.4031 | 0.3906 |
| MS | 99.64% | 0.1844 | 0.2993 | 11.3243 | 0.4929 | 0.4254 | 0.4282 |
| IR+MS | 99.25% | 0.2300 | 0.3519 | 10.4164 | 0.6345 | 0.4887 | 0.4566 |
| $^{13}$C NMR | 99.64% | 0.1016 | 0.1801 | 14.8865 | 0.4249 | 0.2952 | 0.3607 |
| $^{1}$H NMR | 99.10% | 0.0720 | 0.1341 | 18.6141 | 0.3329 | 0.2203 | 0.2572 |
| HSQC NMR | 99.64% | 0.2058 | 0.3221 | 13.4919 | 0.5495 | 0.4392 | 0.4274 |
| Jointly NMR | 98.92% | 0.4151 | 0.5322 | 8.3091 | 0.7209 | 0.6367 | 0.5862 |
| Jointly NMR+IR | 99.60% | 0.4121 | 0.5341 | 8.3855 | 0.7764 | 0.6575 | 0.5809 |
| Jointly NMR+MS | 98.37% | 0.4518 | 0.5601 | 8.1682 | 0.7618 | 0.6760 | 0.6063 |
| Jointly NMR+IR+MS | 99.79% | **0.4875** | **0.5973** | **8.1151** | **0.8103** | **0.7099** | **0.6222** |

representation of spectra, but also provides the foundation for the more substantial gains observed in multi-spectral integration.

## 3.3 Synergistic Interaction among Multiple Spectra

In line with our central hypothesis that integrating complementary spectroscopic modalities enhances molecular structure prediction, we systematically benchmarked SpectraLLM across single-spectrum and multi-spectral configurations on two representative benchmarks: QM9s, which provides simulated IR, Raman, and UV–Vis spectra, and the Multimodal Spectroscopic dataset, which includes IR, MS, and multiple NMR modalities ($^{1}$H, $^{13}$C, HSQC). Importantly, the results reported

in Table 4 reflect controlled test-time conditions: although SpectraLLM is trained on datasets containing mixtures of single- and multi-spectral inputs, at inference we restrict the prompting to either single-spectrum or multi-spectral inputs to isolate the contribution of each spectra-fusion strategy. As shown in Table 4, single-spectrum inputs alone impose clear limits on predictive accuracy, whereas their fusion yields smooth and substantial performance gains across all structural metrics. On QM9s, for example, jointly leveraging IR, Raman, and UV–Vis increases Tanimoto similarity from 0.2500 (Raman alone) to 0.3355 and lowers MCES from more than 10 to 4.9647, highlighting the additive value of combining vibrational and electronic signatures. A similar trend holds for the Multimodal Spectroscopic benchmark: integrating $^1$H, $^{13}$C, and HSQC NMR yields a strong boost in substructural fidelity (Tanimoto 0.4151; MCES 8.3091), with further improvements when NMR is fused with MS and IR (functional-group recovery 0.8103; Fraggle 0.6222).

To more precisely disentangle inter-spectra interactions and quantify which spectra contribute most to joint reasoning, we include an extended set of ablation experiments in Appendix A.9. Concretely, on QM9s we train a separate model for every possible input configuration and evaluate each model both in-domain and cross-domain. These experiments reveal a clear hierarchy of informational complementarity across spectroscopic signals. Cross-domain evaluation further shows that models trained on multi-spectral inputs generalize better to unseen single-spectrum conditions than the reverse, indicating that fused spectra representations act as a more expressive structural prior.

Overall, these ablations corroborate our main finding that the benefits of multi-spectral fusion stem not only from aggregating signal strength but also from enabling SpectraLLM to form a shared latent space that captures spectra-invariant structural cues.

To gain mechanistic insight into the synergistic effects of multi-spectral integration, we conducted a qualitative case analysis focusing on representative molecules that were successfully reconstructed only when specific combinations of modalities were available (Fig. 2). The analysis provides a clearer understanding of how Raman and IR spectra contribute uniquely to accurate molecular reconstruction and why their fusion is essential in overcoming the limitations of unimodal spectral inputs.

In the top panel, where Raman spectra play a crucial role, we observe several cases where IR or UV-Vis alone misassigned key structural motifs, but the inclusion of Raman spectra rectified these errors. For instance, the molecule `COCc1cc[nH]c1` (imidazole scaffold) is correctly reconstructed with the joint IR + Raman input. While IR spectra alone struggle to accurately assign the heteroaromatic ring structure, Raman's sensitivity to polarizability-dependent substructures allows for precise identification of the imidazole ring, which otherwise remains ambiguous in IR spectra. Similarly, for `C#CC(=O)C#CC`, the carbon-carbon triple bond with adjacent ketone groups is correctly identified only when Raman spectra are combined with IR or UV-Vis data, revealing how Raman's ability to detect bond polarizability is key to distinguishing these motifs. The case of `CC(C)(C)C(=O)C` highlights Raman's power in resolving functional groups and differentiating between isomers that IR alone cannot clarify. In the bottom panel, we observe the indispensable role of IR spectra in resolving critical structural details, especially for carbonyl functionalities and complex branching configurations. For example, `CC(C)(C)C(=O)C` is correctly identified when IR is included, but remains ambiguous with Raman and UV-Vis alone. This reflects IR's ability to distinctly capture the position and environment of carbonyl groups, which cannot be unambiguously inferred from Raman's more general vibrational modes or UV-Vis electronic transitions. Similarly, when `CC1CC(C)(O)C1` is reconstructed, the inclusion of IR data resolves the branching pattern, which Raman and UV-Vis could not distinguish. This emphasizes the importance of IR in delineating carbonyl positioning and the molecular architecture at specific branching points, showing its unique contribution to correct isomeric assignment.

These examples confirm that while no single spectrum suffices, the integration of multiple modalities provides complementary structural constraints that significantly reduce ambiguity and improve the accuracy of molecular reconstruction. However, it is important to acknowledge that the analysis presented here showcases only the successful and accurate cases. To better understand the limitations and boundaries of this approach, we further explore `"bad cases"` in Appendix A.13. These analyses reveal that the upper bound of Tanimoto similarity and the observed performance ceiling are often constrained by the inherent ambiguity in spectral data, where structurally different molecules share similar spectral features. These challenging cases demonstrate that while multi-spectral fusion

provides substantial gains in structural prediction, the model's ability to distinguish between these highly similar spectra remains a fundamental challenge.

## 4 RELATED WORKS

Structure elucidation supports mechanistic studies and functional analysis in biomolecules (Krishnan & Rupp, 2012; Brünger et al., 1998), guides material design and defect analysis (Hemath et al., 2020; Gu et al., 2024), and optimizes product performance (Rantanen & Khinast, 2015). Powered by techniques such as Cryo-EM (Adrian et al., 1984), NMR (Hall, 1964), IR (Ng & Simmons, 1999), and MS (De Hoffmann & Stroobant, 2007), it involves interpreting spectral features to identify functional groups and connectivity patterns (Okada & Kotani, 1992; Coates et al., 2000). These are matched against spectral databases (e.g., NIST (Linstrom & Mallard, 2001), MoNA (Horai et al., 2010), SDBS (Punjabi et al., 2025), GNPS (Aron et al., 2020)) to infer candidate structures (Shiferaw et al., 2020; Platte & Heise, 2014).

IR, Raman, and UV-Vis spectroscopy are widely used for molecular structure elucidation, probing functional groups, vibrational modes, and electronic transitions, respectively Stuart (2004); Perkampus (2013); Picollo et al. (2019); Wilson et al. (1980); Zhang et al. (2020); Kim et al. (2013); Skinnider et al. (2021). While they are fast and non-destructive, these techniques suffer from peak overlap, weak signals, and background interference, and their interpretation often requires expert knowledge. NMR spectroscopy offers richer structural resolution, revealing atomic connectivity, chemical environments, and stereochemistry through one-dimensional ($^1$H, $^{13}$C) and two-dimensional (COSY, HSQC, HMBC) spectra Bovey et al. (1988); James (2012); Binev et al. (2007); Meiler (2003); Elson & Magde (1974); Mueller (1979). However, NMR analysis remains time-consuming and combinatorially complex. Mass spectrometry (MS), by contrast, provides high sensitivity and low sample requirements Quinn et al. (2020); Gentry et al. (2024), inferring partial structures or formulas from fragmentation patterns Coelho & Franco (2013); Nalbantoglu & Amri (2019). Yet, fragmentation depends strongly on experimental conditions, and spectra often exhibit high ambiguity and poor reproducibility across settings Bittremieux et al. (2022); da Silva et al. (2015); Stein (2012); Seger (2012); Dührkop et al. (2015). These diverse limitations underscore why no single spectroscopic modality suffices for robust, automated structure elucidation.

Automating structure elucidation has therefore become a longstanding challenge. Machine learning and deep learning methods (Li et al., 2022; Specht et al., 2024; Sridharan et al., 2022; Sapegin & Bear, 2024; Rippel et al., 2015; Defferrard et al., 2016; Cho et al., 2014b;a; Liu et al., 2017; Tang et al., 2020; Berman et al., 2023; Zhang et al., 2024; Xu et al., 2017; He et al., 2021; Achiam et al., 2023) have advanced tasks such as substructure identification, candidate filtering, and molecular property prediction. Yet, most remain confined to a single spectral modality. By contrast, human experts routinely integrate complementary spectra to resolve ambiguities and narrow the candidate space. Recent works support this direction: multi-spectral models combining IR, $^1$H-NMR, and $^{13}$C-NMR achieve notable accuracy gains when aided by formulas or partial structures Alberts et al. (2023); Yao et al. (2023), and even without formulas, $^1$H/$^{13}$C-NMR alone can reconstruct molecules up to 19 heavy atoms Alberts et al. (2023); Hu et al. (2024). These results demonstrate that end-to-end multi-spectral fusion is not only feasible but also critical for improving accuracy and generalization in automated structure elucidation.

## 5 CONCLUSION

In this work, we presented SpectraLLM, a unified language-based framework for molecular structure elucidation. By transforming diverse spectroscopic signals into a shared textual representation, the model leverages the reasoning capabilities of large language models to capture complementary structural constraints across modalities. Systematic evaluation on four benchmark datasets demonstrated that SpectraLLM not only surpasses existing baselines under unimodal settings, but also achieves consistent and substantial gains when integrating multiple spectra, mirroring expert analytical practice. Beyond empirical performance, our results highlight a broader conceptual finding: a general-purpose language model can internalize spectrum–structure relationships purely from paired data, without spectroscopy-specific encoders or fragmentation heuristics. This demonstrates that non-linguistic scientific measurements can be mapped into a language token space in which

structured chemical reasoning becomes feasible. We view this text-only reframing of spectroscopic inference as a promising direction for building general scientific inverse solvers and for extending language-model-based reasoning to a wider range of physical measurement domains.

## ETHICS STATEMENT

All datasets used in this study are derived from publicly available and standardized chemical databases. No personally identifiable, proprietary, or otherwise sensitive information was included. The research does not involve human subjects, animals, or environmental interventions, and therefore poses no foreseeable ethical risks.

## REPRODUCIBILITY STATEMENT

We have taken steps to ensure reproducibility of our results. The datasets employed are all publicly available and are cited in the main text. Detailed descriptions of data preprocessing, model architecture, training objectives, and evaluation protocols are provided in the Methods and Appendix sections. Hyperparameters, prompt templates, and implementation details are specified in the supplementary material. Source code and processed data will be released upon publication to facilitate independent verification.

## ACKNOWLEDGMENTS

The authors thank all colleagues and collaborators for their valuable discussions and constructive feedback during the preparation of this work. We also gratefully acknowledge the financial support from the Strategic Priority Research Program of the Chinese Academy of Sciences (Grant No. XDA0480102) and the National Natural Science Foundation of China (Grant Nos. 62576339, 92570204, 62236010).

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

# A APPENDIX

## A.1 DATASETS

We pre-trained the model using diverse data compiled from multiple publicly available datasets, Table 1 presents the individual characteristics of each spectrum, while Table 5 presents the basic information of all the datasets we collected.

Table 5: Overview of four spectroscopic datasets and their modality coverage.

| Dataset | IR | Raman | UV | NMR | Mass | Molecule | Spectral |
|---|---|---|---|---|---|---|---|
| QM9s | $\checkmark$ | $\checkmark$ | $\checkmark$ | - | - | 129,817 | 389,451 |
| Multimodal Spectroscopic dataset | $\checkmark$ | - | - | $\checkmark$ $^1$H,$^{13}$C,HSQC | $\checkmark$ Positive, Negative | 794,403 | 4,766,418 |
| MassSpecGym | - | - | - | - | $\checkmark$ | 231,104 | 231,104 |
| MassBank | - | - | - | - | $\checkmark$ | 122,746 | 122,746 |
| ALL | $\checkmark$ | $\checkmark$ | $\checkmark$ | $\checkmark$ | $\checkmark$ | 943,732 | 5,510,655 |

Our data preprocessing pipeline consists of four key steps: SMILES standardization and molecular alignment, dataset merging and splitting, spectral feature extraction, and prompt formulation for language modeling. We first standardized all SMILES strings across the four constituent datasets using RDKit [1]. For each molecule, we parsed the provided SMILES using `MolFromSmiles` to obtain an internal molecular graph, followed by `SanitizeMol` to canonicalize valence states, remove invalid structures, and generate a consistent, RDKit-normalized SMILES representation. This canonicalization step ensures that molecules originating from different datasets share an identical structural encoding, even when the raw SMILES strings differ in ordering, stereochemical formatting, or atom-group conventions. With the unified SMILES representation as a unique key, we merged all datasets and naturally deduplicated molecules appearing across multiple sources. Table 6 illustrates how spectra from four datasets are integrated under a single standardized SMILES identifier. Molecules with multiple spectra are concatenated using the "|" separator.

Table 6: Mapping of Molecular Identifiers Across Datasets

| SMILES | QM9s_number | MassBank_file_name | multimodal_spectroscopic dataset_Idx | MassSpecGym_identifier |
|---|---|---|---|---|
| Cc1ccncn1 | 898 | MSBNK-CASMI_2016-SM826401 | 124352 | MassSpecGymID0068222 MassSpecGymID0068223 MassSpecGymID0068224 |

After consolidation, we obtained 943,730 unique molecules, each associated with one to five spectrum types and up to five total spectral instances. Before performing dataset splitting, we preserved the official train/validation/test partition provided by MassSpecGym, ensuring strict comparability with their benchmark. The remaining molecules—those originating from QM9s, MassBank, and the Multimodal Spectroscopic dataset—were then randomly divided into an 8:1:1 train/validation/test split based on standardized molecular identity. This guarantees that all spectra belonging to the same molecule are placed in the same split and that each spectral modality is sufficiently represented across all subsets (Table 7). All reported metrics are computed on the resulting held-out test set. Next, we applied modality-specific preprocessing procedures to convert each spectrum into a structured textual representation suitable for language model training.

**Vibrational and electronic Spectra**. For continuous spectra, we retained effective signals through modality-specific truncation and normalization procedures. In the IR modality, wavenumbers were restricted to 500-4000 $cm^{-1}$ for the QM9s dataset and 400-4000 $cm^{-1}$ for the Multimodal Spectroscopic dataset, uniformly resampled at 1 $cm^{-1}$ intervals (3501 and 3601 points, respectively). Raman spectra adopted the same 500-4000 $cm^{-1}$ range and sampling. UV-Vis spectra were truncated to the 1.0-15.0 eV range, with 0.02 eV sampling intervals (700 points). All spectral intensities were normalized by their maximum value to ensure comparability across molecules and suppress variation introduced by experimental conditions. To further support model interpretability and reduce input complexity, we extracted discrete peak-based representations from the preprocessed spectra.

---

[1]https://www.rdkit.org/

Table 7: Data distribution across spectral modalities and splits.

| | Spectrum | Train | Val | Test | ALL |
|---|---|---|---|---|---|
| | Mass | 301,133 | 15,849 | 14,247 | 331,229 |
| | MS/MS Positive | 643,605 | 71,511 | 79,287 | 794,403 |
| | $^{13}$C NMR | 643,604 | 71,511 | 79,287 | 794,402 |
| | $^1$H NMR | 643,586 | 71,509 | 79,285 | 794,380 |
| Single | HSQC NMR | 643,605 | 71,511 | 79,286 | 794,402 |
| | IR | 743,352 | 82,594 | 87,688 | 913,634 |
| | Raman | 105,080 | 11,675 | 13,062 | 129,817 |
| | UV-Vis | 105,078 | 11,675 | 13,062 | 129,815 |
| | IR+Raman+UV-Vis | 104,948 | 11,660 | 12,416 | 129,024 |
| Multi | $^{13}$C+$^1$H+HSQC NMR | 494,173 | 54,908 | 78,882 | 627,963 |
| | IR+Raman+UV-Vis+NMR+Mass | 383,870 | 42,652 | 30,937 | 457,459 |
| **ALL** | | 4,812,034 | 517,055 | 567,439 | 5,896,528 |

More realistically, characteristic peaks from all continuous spectra were extracted using built-in functions from the SciPy library (Virtanen et al., 2020), based on local maxima and prominence thresholds. Each spectrum was encoded in structured JSON format, recording only the coordinates and relative intensities of characteristic peaks, such as: `"Wavenumbers": ["wavenumber`$_1$`", "wavenumber`$_2$`"], "Intensities": ["intensity`$_1$`", "intensity`$_2$`"]`.

**Nuclear magnetic resonance spectrum**. For nuclear magnetic resonance (NMR) spectra, we applied modality-specific preprocessing and discrete feature extraction. For $^{13}$C NMR and $^1$H NMR, chemical shift ranges were set to 220-0 ppm and 12-0 ppm, respectively. A dynamic threshold (with the threshold being 1% of the maximum spectral intensity) was used to filter background noise, and the spectra were cropped to the minimal region containing significant peaks. The intensity values have all been normalization. For HSQC NMR, which contain 2D correlation peaks between hydrogen and carbon atoms, only peak-level information was retained. Each peak was characterized by its $^1$H and $^{13}$C chemical shifts and associated proton count (nH), defaulted to 1 if unspecified. To support downstream prompt generation, we represented NMR signals in structured formats. For $^{13}$C and $^1$H spectra, data were stored as key-value pairs of shift and intensity values, e.g., `"C-shifts": ["shift`$_1$`", "shift`$_2$`"], "Intensities": ["intensity`$_1$`", "intensity`$_2$`"]`. For HSQC, the 2D correlations were encoded as: `"H-shifts": ["h`$_1$`", "h`$_2$`"], "C-shifts": ["c`$_1$`", "c`$_2$`"], "Intensities": ["intensity`$_1$`", "intensity`$_2$`"]`.

**Mass Spectrometry**. For mass spectrometry (MS), we retained spectra containing between 2 and 1280 peaks. All mass-to-charge ratio (m/z) values were rounded to two decimal places, and peak intensities were converted to relative abundances within a [0, 100] range, also to two decimal places. To preserve experimental context and enable condition-aware modeling, auxiliary metadata (such as instrument type, collision energy, and adduct ion species) was retained when available, distinguishing spectra acquired under different experimental settings for the same molecule. For downstream prompt construction, MS spectra were represented as structured arrays of m/z-intensity pairs in JSON format, for example: `"mzs": ["m/z`$_1$`", "m/z`$_2$`"], "intensities": ["intensity`$_1$`", "intensity`$_2$`"]`.

Given that the prediction target is the SMILES representation of molecular structures, we naturally formulated the structure elucidation task as a sequence-to-sequence generation problem. As illustrated in Table 30 and Table 29, training data were organized in a dialogue-style format, where each example consists of a `"Human"` prompt and a corresponding `"GPT"` response. The `"Human"` component encodes the query, including the spectrum modality, extracted features, and a task directive to infer molecular structure. The `"GPT"` component provides the expected model output, i.e., the target SMILES string.

To promote generalization across varying data configurations, we designed a diverse set of input formats:

1. Single-spectrum setting (Table. 30): molecular structure prediction based solely on one spectrum type, including individual IR, Raman, UV-Vis, $^1$H NMR, $^{13}$C NMR, HSQC NMR, or MS spectra.

2. Multi-spectral setting (Table. 29): molecular structure prediction using two or more spectra from the same molecule. This includes:

- Joint IR, Raman, and UV spectra from the QM9s dataset;
- Joint IR, NMR, and MS spectra from the Multimodal Spectroscopic dataset;
- Combined $^1$H NMR, $^{13}$C NMR, and HSQC NMR spectra;

This formulation allows the model to reason across heterogeneous spectral evidence while remaining robust to missing modalities and variations in experimental settings. We finally construct a large-scale multi-spectral question-answering dataset, whose composition is summarized in Table 7. To ensure compatibility with the input length constraints of the base language model, we exclude examples with tokenized prompts exceeding 1024 tokens. Leveraging this dataset, we perform supervised fine-tuning of the foundational Qwen3-32B language model to enable spectrum-informed molecular reasoning.

## A.2 DATASETS ANALYSIS

We conducted a detailed exploratory data analysis (EDA) on our datasets, examining molecular composition and functional group distributions as shown in Figures 3 and 4. The datasets encompass a total of 943,729 molecules, reflecting both high structural diversity and strong chemical relevance.

The molecular weight (MW) distribution indicates that the vast majority of molecules fall within the 100–500 Da range, peaking around 250 Da. This confirms that our datasets are predominantly composed of small molecules, which remain critically important across multiple domains. For instance, in metabolomics and clinical diagnostics, most metabolites and disease-related biomarkers are below 300 Da. Environmental and food safety monitoring typically targets small organic molecules such as pesticides, plasticizers, and pollutants. In chemical and materials manufacturing, quality control of monomers, solvents, intermediates, and byproducts similarly relies on small-molecule spectral analysis. The prevalence of small molecules in widely used public spectral datasets (e.g., NIST, MassBank, GNPS, QM9) further supports their relevance and generalizability.

Heavy atom count (HAC) is mostly concentrated in the 10–45 range, suggesting that the datasets cover molecules of moderate structural complexity. LogP values are primarily between 0 and 5, indicating that the dataset includes compounds spanning from moderately hydrophilic to moderately lipophilic, which is critical for modeling both solubility and spectral properties across different chemical environments.

Overall, the EDA confirms that our datasets are chemically diverse, structurally complex, and rich in functionally relevant substructures. This combination ensures that models trained on these data can learn robust representations suitable for a wide range of small-molecule spectral prediction tasks, making the datasets both practically relevant and scientifically challenging.

## A.3 METRICS FOR STRUCTURAL SIMILARITY

Our evaluation goes beyond assessing the exact recovery of molecular structures, and also probes whether the model has internalized the fundamental chemical cues embedded within the spectra, such as generating chemically plausible candidates, reconstructing key substructures, and correctly identifying functional groups. To quantify this, we not only evaluate the accuracy of functional group prediction as a coarse-grained measure of structural fidelity, but also assess molecular similarity at a finer granularity using fingerprint-based Tanimoto metrics and maximum common substructure (MCES) analysis. The subsequent section provides a detailed account of each evaluation criterion.

### A.3.1 FUNCTIONAL GROUP SIMILARITY METRIC

To quantify the similarity between two molecules in terms of their functional group composition, we define a functional group similarity score based on SMARTS pattern matching. Using a curated dictionary of 17 common functional groups (e.g., alcohols, ketones, ethers; see Table 8), we identify the presence of each group via substructure search implemented with RDKit (Landrum et al., 2016). Let $G_1$ and $G_2$ denote the sets of functional group types identified in molecule 1 and molecule 2, respectively. The functional group similarity $S_{FG}$ is then computed as the Jaccard index over these sets:

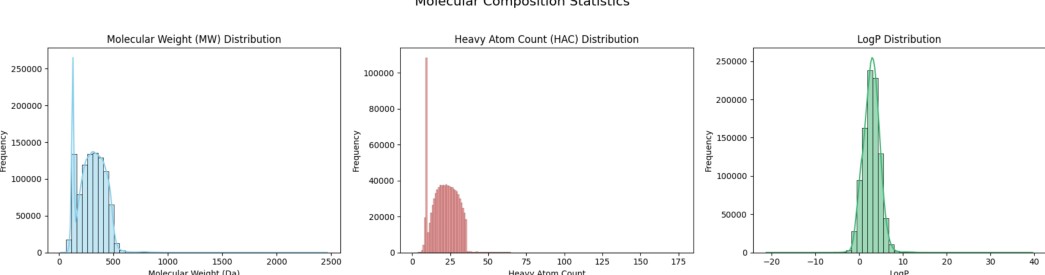

Figure 3: Molecular Property Distributions of the Dataset. This panel of three histograms summarizes critical molecular properties of the dataset:Molecular Weight (MW) Distribution (Left): Shows the frequency of molecules across a weight range (0–2500 Da), with a concentration around 500 Da (consistent with typical small-molecule datasets). Heavy Atom Count (HAC) Distribution (Middle): Characterizes the number of non-hydrogen atoms per molecule, peaking at 25 heavy atoms (reflecting moderate molecular complexity). LogP Distribution (Right): Depicts the octanol-water partition coefficient (a measure of lipophilicity), with a sharp peak near 0 (indicating a skew toward moderately hydrophilic molecules).

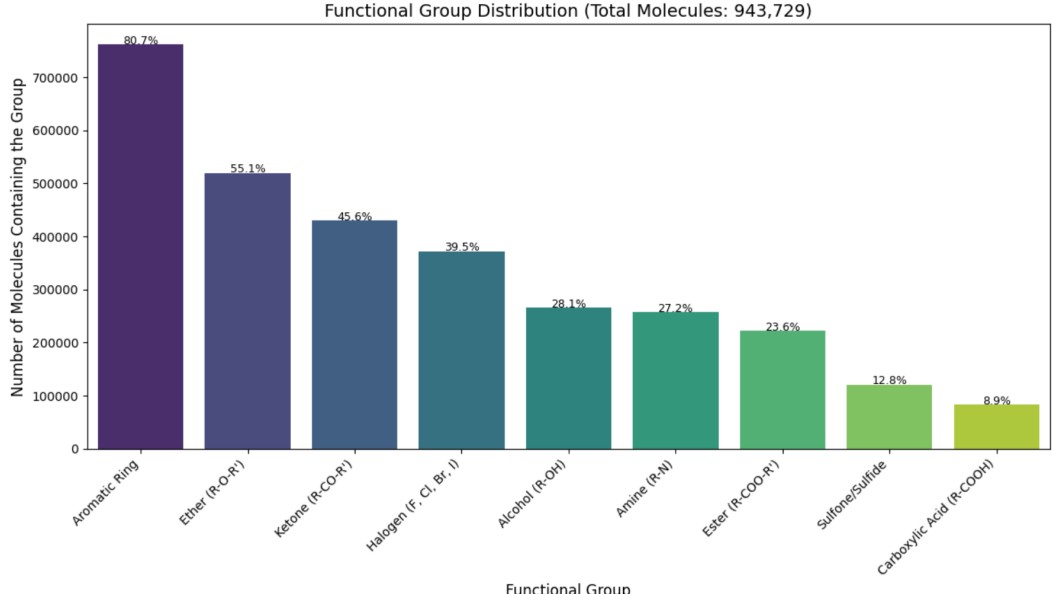

Figure 4: Functional Group Distribution in the Dataset. This bar plot quantifies the prevalence of key functional groups across the 943,729 molecules in our training dataset. The y-axis represents the number of molecules containing each functional group, with the corresponding percentage (relative to the total dataset size) labeled atop each bar.

Table 8: Functional group definitions used.

| Index | Group | Definition |
|---|---|---|
| 1 | Alkane | `[CX4]` |
| 2 | Alkene | `[CX3]=[CX3]` |
| 3 | Alkyne | `[CX2]#C` |
| 4 | Arene | `[$([cX3](:*):*),$([cX2+](:*):*)]` |
| 5 | Alcohol | `[#6][OX2H]` |
| 6 | Ether | `[OD2]([#6])[#6]` |
| 7 | Aldehyde | `[CX3H1](=O)[#6]` |
| 8 | Ketone | `[#6][CX3](=O)[#6]` |
| 9 | Carboxylic acid | `[CX3](=O)[OX2H1]` |
| 10 | Ester | `[#6][CX3](=O)[OX2H0][#6]` |
| 11 | haloalkane | `[#6][F,Cl,Br,I]` |
| 12 | Alkyl halide | `[CX3](=[OX1])[F,Cl,Br,I]` |
| 13 | Amine | `[NX3;$(NC=O)]` |
| 14 | Amide | `[NX3][CX3](=[OX1])[#6]` |
| 15 | Nitrile | `[NX1]#[CX2]` |
| 16 | Sulfide | `[#16X2H0]` |
| 17 | Thiol | `[#16X2H]` |

$$S_{\text{FG}}(mol_1, mol_2) = \frac{G_1 \cap G_2}{G_1 \cup G_2}$$

This metric reflects the qualitative overlap in functional group composition, regardless of group count or spatial arrangement. Ranging from 0 (no shared functional group types) to 1 (complete overlap), it offers a chemically interpretable measure that complements structure-level similarity metrics.

### A.3.2 MOLECULAR FINGERPRINT SIMILARITY METRICS

Molecular fingerprints are vector representations that encode the presence of specific substructures within a molecule, and are widely used to assess molecular similarity. In this work, we compute fingerprint-based similarities between predicted and reference molecules using RDKit and several established metrics:

**Tanimoto Similarity**. Given two binary fingerprints $A$ and $B$, with $a = |A|$, $b = |B|$, and $c = |A \cap B|$, the Tanimoto coefficient is defined as:

$$\text{Tanimoto}(A, B) = \frac{|A \cap B|}{|A \cup B|} = \frac{c}{a + b - c}$$

This metric is computed using ECFP4 circular fingerprints, which capture atom-centered substructures within a radius of 2 bonds.

**MACCS Tanimoto Similarity**. Identical in form to the standard Tanimoto coefficient, but computed over 166-bit MACCS keys, which encode the presence of predefined structural fragments rather than circular environments.

**Cosine Similarity**. For continuous-valued fingerprints (e.g., neural embeddings), similarity is computed as the cosine of the angle between vectors:

$$\text{Cosine}(A, B) = \frac{A \cdot B}{\| A \| \| B \|}$$

where $A$ and $B$ denote the continuous fingerprint vectors (embeddings) of the predicted molecule and the reference molecule, respectively. This metric captures the overall similarity of their learned feature representations in high-dimensional space.

**Fraggle Similarity**. Based on fragment matching rather than fixed-length vectors, this metric decomposes both molecules into substructures and computes alignment-based similarity. It is implemented via RDKit's `FraggleSim` module, and reflects partial structural overlap beyond atom-level matching.

### A.3.3 DERIVED STATISTICAL METRICS BASED ON FINGERPRINT SIMILARITY

In addition to raw similarity scores, we report threshold-based classification metrics that stratify prediction quality according to Tanimoto similarity levels. These derived measures capture not only exact matches but also structurally plausible alternatives that retain key substructures of the target molecule.

**Approximate Match Rate**. The proportion of predicted molecules with Tanimoto similarity $\geq$ 0.675 to the reference structure. This threshold is widely adopted to indicate moderate-to-high structural similarity in cheminformatics.

**Acceptable Match Rate**. The proportion of predictions with Tanimoto similarity $\geq$ 0.4, often associated with weak but potentially chemically relevant similarity.

### A.3.4 MAXIMUM COMMON SUBSTRUCTURE SIMILARITY

The Maximum Common Edge Subgraph (MCES) similarity quantifies the extent of structural overlap between two molecules by identifying their largest common subgraph under graph isomorphism. Given two molecular graphs $G_1$ and $G_2$, the MCES is defined as the largest subgraph $G_{\text{common}}$ such that $G_{\text{common}} \subseteq G_1$ and $G_{\text{common}} \subseteq G_2$. The similarity is computed as the ratio of shared bonds to the maximum number of bonds in either molecule:

$$\text{MCES Similarity} = \frac{E(G_{\text{common}})}{\max(|E(G_1)|, |E(G_2)|)}$$

where $E(G)$ denotes the set of edges (chemical bonds) in graph $G$. This metric captures partial structural correctness and is particularly informative when global fingerprint-based measures fail to reflect local substructure similarity.

## A.4 BASELINE MODELS

To benchmark the performance of SpectraLLM, we compare it with state-of-the-art spectrum-to-structure models spanning various spectroscopic modalities, including mass spectrometry, vibrational spectroscopy, and NMR.

For mass spectrometry data, we compare our model with DiffMS (Bohde et al., 2025). DiffMS is a diffusion-based molecular generation framework specifically designed for mass spectrometry-based structure elucidation. The encoder of DiffMS uses a Transformer architecture to model spectral features such as peak formulas and neutral losses, while its decoder is a discrete graph diffusion model conditioned on the chemical formula. During training, the decoder is pre-trained on a large-scale fingerprint-structure dataset, allowing it to robustly map latent spectrum embeddings to molecular structures. DiffMS demonstrates superior performance across various benchmarks, with stable improvements as the amount of pre-training data increases.

In comparison, another baseline model for mass spectrometry structure elucidation is Spec2Mol (Litsa et al., 2023). Spec2Mol adopts an end-to-end encoder-decoder framework inspired by speech recognition (Speech2Text). The encoder maps MS/MS spectra to embeddings, and the decoder, a structure auto-encoder pre-trained on large compound datasets such as PubChem and ZINC, decodes these embeddings into SMILES strings to propose potential molecular structures. Spec2Mol measures the similarity between generated structures and the ground truth using metrics such as fingerprint similarity and maximum common substructure. The results show that Spec2Mol can identify key information (key substructures) from the spectra and generate structures with similarity to the real molecules comparable to traditional methods that rely on fragmentation trees.

Since our dataset is large and training time is extensive, we directly use the open-source embedding parameters of both baseline models without additional fine-tuning. Specifically, Spec2Mol's spectrum encoder was trained on the NIST Tandem MS library, and DiffMS's embedding comes from pre-training on MassSpecGym, which includes a subset of MassBank. Therefore, the evaluation of DiffMS on MassBank can be considered in-domain, while Spec2Mol, trained on a subset of NIST MS/MS, is evaluated out-of-domain. To ensure comparability with Spec2Mol, we set the number of generated results in DiffMS's decoding process to 1.

In our evaluation of vibrational and electronic spectroscopy data, we adopt two baseline models: Spectra-to-Structure (Kanakala et al., 2024) and IR-to-Structure (Alberts et al., 2024a). Spectra-to-Structure implements a dual-encoder architecture (named SMEN) comprising a transformer-based spectral encoder and a graph-based molecule encoder (EGNN), trained via contrastive loss so that correct molecule–spectrum pairs occupy nearby points in a shared embedding space; candidate molecules are then ranked by embedding similarity, and a supplementary generative module decodes top embeddings into SMILES when needed. IR-to-Structure deploys a sequence-to-sequence transformer that takes as input an IR spectrum (and in the original work also the molecular formula) and outputs a SMILES string describing the predicted molecule. To ensure a fair comparison under our own experimental constraints, we omit the molecular formula and supply only spectral data as input for both baselines. The original models were trained under different regimes (Spectra-to-Structure on small simulated-IR QM9 molecules, IR-to-Structure on simulated and experimental IR spectra for 6–13 heavy-atom molecules), yet we re-adopt their published training configurations and preprocessing pipelines, applying them to our adapted IR, Raman, and UV–Vis subsets drawn from QM9s dataset; summary statistics of training and test splits are provided in Table 9, and detailed training hyperparameters (epochs, batch sizes, learning rates) follow those in their respective sources (see Appendix A.12).

In our evaluation of NMR-based spectral prediction, we include NMR2Struct (Hu et al., 2024) as a baseline. NMR2Struct implements an end-to-end multitask machine learning framework that combines a convolutional neural network to encode raw one-dimensional $^1$H and/or $^{13}$C NMR spectra with a transformer-based decoder that predicts molecular connectivity and full structure without requiring a molecular formula or pre-defined fragments. The model is first pretrained on a substructure-to-structure task, assembling small molecular fragments into complete molecular graphs, then extended into a full spectrum-to-structure model. In our experiments, we re-trained NMR2Struct using the $^1$H and $^{13}$C NMR subsets drawn from our Multimodal Spectroscopic dataset under the same preprocessing and training protocols, and we report dataset statistics in Table 9.

Table 9: Dataset Split for Different Baseline Models

| Model | Dataset | Train/Val | Test |
|---|---|---|---|
| IR-to-Structure | QM9s(IR-only) | 105,079/11,676 | 13,062 |
| | QM9s(Raman-only) | 105,079/11,676 | 13,062 |
| | QM9s(UV-only) | 105,079/11,676 | 13,062 |
| Spectra2Structure | QM9s(IR-only) | 105,079/11,676 | 13,062 |
| | QM9s(Raman-only) | 105,079/11,676 | 13,062 |
| | QM9s(UV-only) | 105,079/11,676 | 13,062 |
| NMR2Struct | Multi(NMR-only) | 643,604/71,512 | 79,287 |

## A.5 SELECTION OF MODEL ARCHITECTURE

Designing an effective model architecture for structure elucidation from spectroscopic data requires careful consideration of the modality-specific characteristics and their interaction with large-scale language modeling. We explored two principal approaches for integrating multiple spectral inputs with large language models: (1) a vision-language-centric pipeline in which continuous spectra are first interpreted via a visual encoder, and their latent representation is subsequently merged with discrete mass spectra for structure generation; and (2) a unified language-based architecture that encodes both continuous and discrete spectra as structured textual descriptions, directly processed by a pretrained LLM.

Table 10: Comparison of SpectraLLM and VLM-LLM performance on the QM9s dataset, demonstrating superior performance of the unified language model in structure elucidation.

| Inputs | Validity ↑ | Tanimoto ↑ | Cosine ↑ | MCES ↓ | Functional Group ↑ | Tanimoto (MACCS) ↑ | Fraggle ↑ |
|---|---|---|---|---|---|---|---|
| **VLM-LLM** | | | | | | | |
| IR | 99.67% | 0.1434 | 0.2446 | 8.7010 | 0.5665 | 0.3510 | 0.2767 |
| Raman | 98.17% | 0.1616 | 0.2716 | 7.8993 | 0.6005 | 0.3676 | 0.3131 |
| UV-Vis | 99.63% | 0.0874 | 0.1576 | 10.2960 | 0.3645 | 0.2158 | 0.2112 |
| Jointly | 99.40% | 0.1954 | 0.3139 | 7.0543 | 0.6828 | 0.4205 | 0.3571 |
| **LLM** | | | | | | | |
| IR | 99.82% | 0.1921 | 0.3120 | 7.5651 | 0.6599 | 0.4330 | 0.3194 |
| Raman | 99.08% | 0.2500 | 0.3786 | 6.4076 | 0.7317 | 0.5071 | 0.2500 |
| UV-Vis | 100.00% | 0.0790 | 0.1426 | 10.6374 | 0.3713 | 0.2026 | 0.2100 |
| Jointly | 98.72% | **0.3355** | **0.4560** | **4.9647** | **0.7934** | **0.5785** | **0.4117** |

Inspired by recent advances in vision–language models (VLMs), we initially explored a two-stage architecture for multimodal spectral inference: IR , Raman, and UV-Vis spectra were rendered into 2D images and processed by a vision encoder (e.g., Qwen2.5-VL). The encoder output was decoded via beam sampling into a preliminary SMILES candidate. This intermediate representation, alongside the mass spectrum (encoded as m/z–intensity pairs or peak descriptors), was subsequently input into a large language model (LLM, e.g., Qwen3-32B) to generate the final molecular structure.

While this hybrid approach leveraged the representational flexibility of VLMs for interpreting unstructured plots, it introduced several critical limitations:

- The intermediate SMILES representation was lossy and error-prone, potentially propagating inaccuracies from the vision stage to the language model.

- The handoff between modalities disrupted end-to-end optimization, as the system lacked a shared latent space and consistent gradient flow.

- When input spectra were incomplete or noisy, the intermediate SMILES failed to preserve chemically meaningful cues, leading to reduced interpretability and robustness.

By contrast, the language-centric architecture presented in this study employs discrete peak features from all modalities as structured input to a unified LLM, eliminating the need for intermediate symbolic forms. As shown in Table 10 and Table 11, we benchmarked both architectures on the QM9s and Multimodal Spectroscopic datasets, reporting structure reconstruction performance across multiple criteria.

Across all metrics and datasets, the unified LLM paradigm consistently outperformed the VLM-LLM cascade. Notably, improvements were most pronounced under stricter evaluation thresholds, underscoring the LLM's superior reasoning capabilities in a purely linguistic input space. We attribute this performance gain to the model's ability to jointly process all modalities in a chemically coherent and interpretable manner, without cross-modal encoding artifacts. Furthermore, unlike the VLM-LLM pipeline, where the information from one modality (e.g., the vision-based representation of IR or Raman spectra) is susceptible to distortion before reaching the LLM, SpectraLLM directly processes peak-level features from all spectral inputs, ensuring consistent gradient flow and maximizing interpretability and robustness, particularly in challenging, noisy, or incomplete spectral data.

These findings reinforce the adoption of a language-based modeling framework for multi-spectral analysis. While VLMs remain valuable for raw spectrum visualization and downstream interpretability, symbolic generation, particularly in the context of small molecule structure elucidation, is best achieved via a unified language model trained on curated, peak-based prompts.

## A.6 DECODING STRATEGIES FOR SPECTRUM-TO-STRUCTURE GENERATION

To generate molecular structures from spectroscopic inputs, we use the fine-tuned SpectraLLM model in a greedy decoding setup with fixed temperature and nucleus sampling parameters. Specifically, we adopt a temperature-controlled sampling strategy with a nucleus sampling threshold of

Table 11: Comparison of SpectraLLM and VLM-LLM performance on the Multimodal Spectroscopic dataset, highlighting the advantages of multi-spectral fusion in SpectraLLM.

| Inputs | Validity ↑ | Tanimoto ↑ | Cosine ↑ | MCES ↓ | Functional Group ↑ | Tanimoto (MACCS) ↑ | Fraggle ↑ |
|---|---|---|---|---|---|---|---|
| | | | | **VLM-LLM** | | | |
| IR | 100.00% | 0.1392 | 0.2385 | 14.3495 | 0.5166 | 0.3373 | 0.3415 |
| MS | 98.56% | 0.1855 | 0.2965 | 10.8124 | 0.5268 | 0.4326 | 0.4258 |
| IR+MS | 99.66% | 0.1902 | 0.4450 | 10.2871 | 0.4812 | 0.4450 | 0.4664 |
| $^{13}$C NMR | 100.00% | 0.1404 | 0.2357 | 15.6020 | 0.4386 | 0.3140 | 0.3632 |
| $^1$H NMR | 100.00% | 0.1484 | 0.2515 | 17.5893 | 0.4098 | 0.3373 | 0.3252 |
| HSQC NMR | 99.82% | 0.1532 | 0.2582 | 16.5800 | 0.4239 | 0.3346 | 0.3438 |
| Jointly NMR | 99.64% | 0.2221 | 0.3425 | 13.2811 | 0.5242 | 0.4341 | 0.4520 |
| Jointly NMR+IR | 99.60% | 0.2598 | 0.3858 | 10.7038 | 0.6493 | 0.5070 | 0.4876 |
| Jointly NMR+MS | 98.37% | 0.2822 | 0.4102 | 12.4570 | 0.6130 | 0.5595 | 0.5415 |
| Jointly NMR+IR+MS | 98.74% | 0.2455 | 0.3681 | 9.9964 | 0.6045 | 0.5098 | 0.4785 |
| | | | | **LLM** | | | |
| IR | 99.63% | 0.1720 | 0.2868 | 15.3234 | 0.6023 | 0.4031 | 0.3906 |
| MS | 99.64% | 0.1844 | 0.2993 | 11.3243 | 0.4929 | 0.4254 | 0.4282 |
| IR+MS | 99.25% | 0.2300 | 0.3519 | 10.4164 | 0.6345 | 0.4887 | 0.4566 |
| $^{13}$C NMR | 99.64% | 0.1016 | 0.1801 | 14.8865 | 0.4249 | 0.2952 | 0.3607 |
| $^1$H NMR | 99.10% | 0.0720 | 0.1341 | 18.6141 | 0.3329 | 0.2203 | 0.2572 |
| HSQC NMR | 99.64% | 0.2058 | 0.3221 | 13.4919 | 0.5495 | 0.4392 | 0.4274 |
| Jointly NMR | 98.92% | 0.4151 | 0.5322 | 8.3091 | 0.7209 | 0.6367 | 0.5862 |
| Jointly NMR+IR | 99.60% | 0.4121 | 0.5341 | 8.3855 | 0.7764 | 0.6575 | 0.5809 |
| Jointly NMR+MS | 98.37% | 0.4518 | 0.5601 | 8.1682 | 0.7618 | 0.6760 | 0.6063 |
| Jointly NMR+IR+MS | 99.79% | **0.4875** | **0.5973** | **8.1151** | **0.8103** | **0.7099** | **0.6222** |

$p = 0.7$ (default setting), and vary the temperature $\tau$ to investigate its influence on generation diversity and accuracy. Lower temperatures bias the model toward more deterministic predictions, while higher temperatures encourage exploration by flattening the probability distribution. As shown in Table 12, a moderate temperature of $\tau = 0.4$ achieves the best overall performance across multiple metrics, including Tanimoto similarity, functional group recovery, and substructure alignment scores (e.g., Fraggle). Extremely low temperatures (e.g., $\tau = 0.2$) slightly improve structure similarity metrics such as cosine and MCES, but at the cost of reduced diversity. Conversely, high temperatures lead to degraded accuracy and increased structural inconsistency. We do not employ beam search or advanced decoding control mechanisms; instead, decoding is performed greedily (beam size = 1), with syntax validity implicitly learned from training data. Although this setup is relatively simple, it proves effective in generating syntactically valid and chemically plausible SMILES strings under constrained sampling.

Beyond temperature sampling, we further conduct a systematic study of decoding strategies to assess their impact on spectrum-to-structure generation. We extend our analysis from basic temperature sampling to a richer set of generation-control methods, including beam search, diverse sampling, and other commonly used decoding variants. These strategies are widely adopted in language generation to improve diversity, enhance global consistency, or mitigate mode collapse. To evaluate their behavior in our setting, we run experiments on the QM9s Jointly test subset with SpectraLLM, and compare 6 different decoding strategies. The configurations of each strategy are summarized in Table 13, and the corresponding generation quality metrics are reported in Table 14.

We observe that Qwen's default sampling configuration (temperature $T = 0.6$, Top-$k = 20$, Top-$p = 0.95$) achieves the best performance among these off-the-shelf strategies (Tanimoto = 0.3355). In comparison, greedy decoding leads to a substantial drop to 0.2746, indicating that spectrum-to-structure generation induces a highly non-convex search space where the absence of stochasticity easily traps the model in low-quality local optima. At the same time, further increasing Top-$k$ or Top-$p$ (i.e., injecting more randomness) typically degrades performance, suggesting that excessive exploration introduces noise and weakens structural accuracy. Diverse beam search (DBS) attains a slightly lower overall Tanimoto score than the default sampling, but it consistently produces multiple candidate structures with stable and chemically reasonable scores (around 0.18–0.19), exhibiting a diverse yet conservative behavior that is particularly desirable in scenarios without a single ground-truth answer. In contrast, Contrastive Search yields the worst performance, implying that its penalization of high-frequency tokens may harm the model's ability to represent regular patterns such as repeated functional groups or chain-like motifs. Importantly, even the best of these mainstream decoding strategies does not match the overall performance of our custom sampling setup,

further demonstrating that spectrum parsing benefits from domain-aware, task-specific optimization of decoding.

In summary, the different decoding strategies expose a clear trade-off between exploration and accuracy, and our proposed configuration achieves the most favorable balance among them. This highlights the importance of task-specific decoding design for spectrum-to-structure generation with large language models.

Table 12: Effect of Decoding Temperature on Molecular Structure Prediction Performance.

| Temperature ↓ | Tanimoto ↑ | Cosine ↑ | MCES ↓ | Functional Group ↑ | Tanimoto(MACCS) ↑ | Fraggle ↑ |
|---|---|---|---|---|---|---|
| 0.2 | 0.4872 | **0.5985** | 6.9840 | 0.8084 | 0.7070 | **0.6302** |
| 0.4 | **0.4875** | 0.5973 | 8.1151 | **0.8103** | **0.7099** | 0.6222 |
| 0.6 | 0.4740 | 0.5846 | **6.6784** | 0.8039 | 0.7022 | 0.6218 |
| 0.8 | 0.4555 | 0.5660 | 7.0897 | 0.7980 | 0.6819 | 0.6075 |
| 1.0 | 0.4367 | 0.5518 | 7.1902 | 0.7867 | 0.6752 | 0.6019 |
| 1.2 | 0.4161 | 0.5352 | 7.3024 | 0.7840 | 0.6592 | 0.5805 |

Table 13: Decoding configurations for SpectraLLM.

| Decoding Setting | do_sample | temperature | top_k | top_p | penalty_alpha | num_beams | num_beam_groups | diversity_penalty | length_penalty |
|---|---|---|---|---|---|---|---|---|---|
| Qwen default | true | 0.6 | 20 | 0.95 | - | - | - | - | - |
| Greedy decoding | false | 0.0 | -1 | 1.0 | - | 1 | - | - | - |
| Top-k sampling | true | 0.95 | {10, 50, 100} | - | - | - | - | - | - |
| Top-p (nucleus) sampling | true | 0.95 | -1 | {0.7, 0.9, 0.98} | - | - | - | - | - |
| Contrastive search | false | - | {4, 8} | - | {0.3, 0.5, 0.7} | - | - | - | - |
| Diverse beam | false | - | - | - | - | {4, 6} | 2 | {0.3, 0.5, 0.7} | 1.0 |

Table 14: Impact of decoding strategies on spectrum-to-structure generation on the QM9s Jointly benchmark.

| SpectraLLM QM9s Jointly | Setting | Validity ↑ | Tanimoto ↑ | Cosine ↑ | MCES ↓ | Functional Group ↑ | Tanimoto (MACCS) ↑ | Fraggle ↑ |
|---|---|---|---|---|---|---|---|---|
| Default | | 98.72% | 0.3355 | 0.4560 | 4.9647 | 0.7934 | 0.5785 | 0.4117 |
| Greedy | | 93.41% | 0.1746 | 0.2782 | 7.9078 | 0.5835 | 0.3711 | 0.2881 |
| Top-k | 10 | 89.38% | 0.1357 | 0.2283 | 10.2059 | 0.5131 | 0.3129 | 0.2451 |
| | 50 | 90.66% | 0.1421 | 0.2363 | 9.8636 | 0.5052 | 0.3253 | 0.2495 |
| | 100 | 89.74% | 0.1392 | 0.2338 | 9.5745 | 0.5307 | 0.3172 | 0.2402 |
| Top-p | 0.7 | 94.51% | 0.1591 | 0.2590 | 8.6744 | 0.5538 | 0.3469 | 0.2546 |
| | 0.9 | 91.76% | 0.1420 | 0.2410 | 9.2804 | 0.5128 | 0.3298 | 0.2489 |
| | 0.98 | 91.39% | 0.1346 | 0.2278 | 9.3858 | 0.5025 | 0.3130 | 0.2412 |
| Diverse beam | [4, 0.3] | 96.15% | 0.1880 | 0.2939 | 8.0324 | 0.6106 | 0.3843 | 0.2966 |
| | [4, 0.5] | 96.15% | 0.1865 | 0.2926 | 7.2010 | 0.6051 | 0.3814 | 0.2947 |
| | [4, 0.7] | 96.34% | 0.1860 | 0.2923 | 7.1473 | 0.6102 | 0.3825 | 0.2913 |
| | [6, 0.3] | 95.97% | 0.1908 | 0.2969 | 7.1126 | 0.6041 | 0.3872 | 0.2936 |
| | [6, 0.5] | 95.42% | 0.1885 | 0.2955 | 9.1881 | 0.6042 | 0.3835 | 0.2851 |
| | [6, 0.7] | 94.69% | 0.1872 | 0.2941 | 9.9826 | 0.6030 | 0.3808 | 0.2858 |
| Contrastive search | [0.3, 4] | 95.60% | 0.1677 | 0.2714 | 8.0077 | 0.5512 | 0.3576 | 0.2744 |
| | [0.3, 8] | 95.60% | 0.1724 | 0.2770 | 10.3884 | 0.5673 | 0.3629 | 0.2747 |
| | [0.5, 4] | 97.99% | 0.1593 | 0.2628 | 7.8598 | 0.5402 | 0.3500 | 0.2615 |
| | [0.5, 8] | 98.35% | 0.1643 | 0.2659 | 8.1965 | 0.5535 | 0.3617 | 0.2523 |
| | [0.7, 4] | 97.62% | 0.1507 | 0.2494 | 8.2458 | 0.4948 | 0.3346 | 0.2470 |
| | [0.7, 8] | 87.36% | 0.1608 | 0.2584 | 9.0210 | 0.5217 | 0.3488 | 0.2610 |

## A.7 IMPACT OF FEATURE PEAK ON MODEL SENSITIVITY

For the parameter choices in feature-peak extraction, we first systematically evaluate the impact of the peak-intensity filtering threshold on model performance. As shown in Table 15, when the threshold is increased from 0% to 1% and 2%, the model exhibits consistent improvements across multiple key metrics (e.g., Tanimoto, Cosine, MCES, MACCS), with the 1% threshold achieving the best overall performance (Tanimoto = 0.4404). This result indicates that moderate filtering of low-intensity peaks is crucial for suppressing background noise and highlighting chemically relevant peak patterns: a too-low threshold introduces many weak, non-discriminative spurious peaks, while

a too-high threshold removes chemically important peaks and makes it difficult for the model to recover structural details. Notably, when the threshold is raised to 10% or 20%, almost all structure-related metrics degrade significantly, suggesting that the loss of chemically informative peaks is the main cause of the performance drop. Overall, the 1%–2% range strikes the best balance between noise suppression and information retention, and serves as the optimal and stable parameter choice in our $\phi$ mapping.

In addition, this ablation reveals a typical yet reasonable phenomenon: the model trained only on QM9s slightly outperforms the mixed-data SpectraLLM on the QM9s test set. This reflects the presence of negative transfer, and at the same time highlights a structural trade-off between general-ization and performance on a homogeneous dataset. QM9s is highly clean, with regular peak shapes and strong spectral consistency, so a model fine-tuned purely on QM9s can more easily achieve a local optimum on its in-distribution test set. In contrast, the mixed-training variant must simul-taneously adapt to real-world spectra from MassBank, MassSpecGym, etc., which contain noisy peaks, distorted peak shapes, and missing peaks, forcing the model to learn a more compromise-oriented and robust spectrum–structure mapping. Thus, this local degradation is an inevitable cost of pursuing cross-domain generalization and noise robustness—just as shown in Table 3 of the main text, SpectraLLM's consistently superior performance on real experimental spectra demonstrates the value of this trade-off.

In the ablation on peak-width features (Table 16), we further compare the model's behavior with and without peak-width descriptors. For single spectral modalities, the influence of peak width exhibits clear differences. On IR data, including peak width slightly degrades performance, while on Raman data we observe the opposite trend: moderate peak-width information improves the model's ability to discriminate continuous band-like peaks. For the UV modality, the impact of peak width is unstable, mainly because UV absorption spectra are inherently sparse and highly susceptible to noise, so peak width contributes relatively little to structure prediction. However, when all three spectra are jointly used as input (`"All"`), the model with peak width consistently achieves better overall performance, suggesting that peak width is not universally beneficial, but in multi-spectral fusion acts as an additional complementary structural cue that helps reduce representation bias across different spectral modalities.

Taken together, these two sets of ablations clearly illustrate the key design trade-offs in spectral preprocessing: (1) an appropriate peak-intensity filtering threshold is a decisive factor for model performance, while thresholds that are too low or too high will both damage the spectrum–structure mapping; (2) the utility of peak-width information is modality-dependent and becomes more valu-able under multi-spectral integration. These results not only validate the rationality of our prepro-cessing strategy, but also underscore that, for realistic chemical applications, joint optimization of spectral feature engineering and model decoding strategies is indispensable.

Table 15: Effect of peak-intensity threshold on spectrum-to-structure performance on QM9s Jointly.

| SpectraLLM QM9s Jointly | Validity ↑ | Tanimoto ↑ | Cosine ↑ | MCES ↓ | Functional Group ↑ | Tanimoto (MACCS) ↑ | Fraggle ↑ |
|---|---|---|---|---|---|---|---|
| 0% | 99.45% | 0.4149 | 0.5388 | 4.36 | 0.8743 | 0.6769 | 0.4696 |
| 1% | **99.82%** | 0.4404 | 0.5613 | 4.1606 | **0.8754** | **0.6978** | **0.4615** |
| 2% | 99.63% | **0.4411** | **0.5622** | **3.9908** | 0.8744 | 0.6917 | 0.4568 |
| 5% | 99.08% | 0.414 | 0.5357 | 4.3983 | 0.865 | 0.6748 | 0.4495 |
| 10% | 99.08% | 0.4071 | 0.53 | 4.4732 | 0.86 | 0.6683 | 0.4505 |
| 20% | 98.90% | 0.369 | 0.4988 | 4.7213 | 0.8513 | 0.6366 | 0.4441 |

## A.8 ROBUSTNESS OF $\phi$ TO SPECTRAL NOISE

To investigate the sensitivity of the mapping $\phi$ from spectra to text, as well as the impact of peak ex-traction on downstream performance, we conduct a series of controlled noise experiments. We first inject synthetic noise directly into the input spectra of the QM9s test set to study how different noise patterns affect the robustness of SpectraLLM. Concretely, we apply three levels of simulated noise (mild, moderate, severe) to IR, Raman, and UV-Vis spectra, with parameterizations summarized in Table 17.

Table 16: Effect of peak-width features and spectral modality on spectrum-to-structure performance on QM9s Jointly.

| SpectraLLM QM9s Jointly | Validity ↑ | Tanimoto ↑ | Cosine ↑ | MCES ↓ | Functional Group ↑ | Tanimoto (MACCS) ↑ | Fraggle ↑ |
|---|---|---|---|---|---|---|---|
| All | 98.53% | 0.4213 | 0.5272 | 4.0046 | 0.8349 | 0.6394 | 0.4504 |
| -wo peak width | 98.53% | 0.4047 | 0.5195 | 4.0288 | 0.8342 | 0.6481 | 0.4397 |
| IR | 99.82% | 0.1921 | 0.3120 | 7.5651 | 0.6599 | 0.4330 | 0.3194 |
| -wo peak width | 99.08% | 0.1459 | 0.2427 | 7.2218 | 0.4978 | 0.3345 | 0.2795 |
| Raman | 99.27% | 0.2500 | 0.3786 | 6.4076 | 0.7317 | 0.5071 | 0.3681 |
| -wo peak width | 99.08% | 0.2977 | 0.4223 | 5.8364 | 0.7597 | 0.5443 | 0.3883 |
| UV-Vis | 93.96% | 0.0640 | 0.1160 | 13.3928 | 0.2978 | 0.2129 | 0.0685 |
| -wo peak width | 99.82% | 0.0812 | 0.1467 | 10.7138 | 0.3453 | 0.1967 | 0.2108 |

Table 17: Noise configurations for simulated IR, Raman, and UV-Vis spectra at three severity levels.

| | IR | Raman | UV-Vis |
|---|---|---|---|
| Mild | rel_noise_sigma=0.02 baseline_amp=0.01 conv_sigma=0.5 spike_prob=0.001 freq_jitter=0.1 | rel_noise_sigma=0.04 baseline_amp=0.03 fluorescence_type=exp fluorescence_strength=1.0 conv_sigma=1.0 spike_prob=0.002 freq_jitter=0.2 | rel_noise_sigma=0.01 baseline_amp=0.0 conv_sigma=3.0 spike_prob=0.001 freq_jitter=0.1 |
| Moderate | rel_noise_sigma=0.05 baseline_amp=0.03 conv_sigma=1.5 spike_prob=0.003 freq_jitter=0.3 | rel_noise_sigma=0.07 baseline_amp=0.08 fluorescence_strength=1.3 conv_sigma=2.0 spike_prob=0.004 freq_jitter=0.4 | rel_noise_sigma=0.04 baseline_amp=0.0 conv_sigma=4.0 spike_prob=0.003 freq_jitter=0.3 |
| Severe | rel_noise_sigma=0.08 baseline_amp=0.08 conv_sigma=3.0 spike_prob=0.01 freq_jitter=0.6 | rel_noise_sigma=0.10 baseline_amp=0.15 fluorescence_strength=1.8 conv_sigma=3.5 spike_prob=0.01 freq_jitter=0.6 | rel_noise_sigma=0.07 baseline_amp=0.0 conv_sigma=5.0 spike_prob=0.01 freq_jitter=0.6 |

We add noise to multi-spectral prompts that jointly contain IR, Raman, and UV-Vis. In total, there are 21 possible constructions: (i) adding one of the three noise levels to exactly one spectrum (e.g., noisy IR, clean Raman/UV); (ii) keeping exactly one spectrum clean while adding the same noise level to the remaining two; and (iii) adding the same noise level to all three spectra. For each setting, we re-construct the noisy test set using a dynamic filtering threshold, set to 1% of the maximum spectral intensity. Table 18 reports a representative subset of 9 configurations, where a superscript "*" denotes the spectrum to which noise is applied.

Overall, the results show a clear monotonic relationship between performance degradation, noise intensity, and the number of corrupted spectra. When only mild noise is injected into a single modality (e.g., noisy IR with clean Raman/UV), the drop in all metrics is negligible. For instance, the Tanimoto score changes only slightly from 0.3355 (clean I+R+U) to 0.3365, 0.2205, or 0.2487 under different single-modality noise configurations, indicating that the model does not catastrophically fail when one spectrum deteriorates, but instead degrades gracefully. Moreover, even when moderate or severe noise (corresponding to heavily distorted spectra rarely observed in practice) is applied to a single modality, the model still maintains high validity and non-trivial Tanimoto scores. This suggests that SpectraLLM effectively exploits the remaining clean, orthogonal spectral information (e.g., Raman and UV-Vis) to compensate for the corrupted channel and continue performing meaningful structural inference.

At the same time, although Tanimoto similarity decreases under stronger noise, the corresponding increase in MCES (Maximum Common Edge Subgraph) distance is relatively mild. In other words, when the model fails, it tends to predict molecules that preserve a similar core scaffold to the ground

Table 18: Effect of modality-specific simulated noise on spectrum-to-structure performance on QM9s Jointly. `"I"`, `"R"`, and `"U"` respectively represent IR, Raman, and UV-Vis.

| SpectraLLM QM9s Jointly | Noisy Strong | Validity ↑ | Tanimoto ↑ | Cosine ↑ | MCES ↓ | Functional Group ↑ | Tanimoto (MACCS) ↑ | Fraggle ↑ |
|---|---|---|---|---|---|---|---|---|
| I+R+U | - | 98.72% | 0.3355 | 0.4560 | 4.9647 | 0.7934 | 0.5785 | 0.4117 |
| I*+R+U | mild | 96.89% | 0.3365 | 0.4659 | 5.3299 | 0.7819 | 0.5955 | 0.4011 |
| | moderate | 97.99% | 0.2688 | 0.3963 | 6.2112 | 0.6851 | 0.5212 | 0.3598 |
| | severe | 87.18% | 0.1712 | 0.2799 | 8.4433 | 0.5010 | 0.3774 | 0.2556 |
| I+R*+U | mild | 98.17% | 0.2487 | 0.3785 | 6.4272 | 0.6669 | 0.5073 | 0.3498 |
| | moderate | 95.79% | 0.1608 | 0.2691 | 7.8891 | 0.5168 | 0.3735 | 0.2811 |
| | severe | 97.44% | 0.0827 | 0.1495 | 9.5714 | 0.2058 | 0.2611 | 0.3719 |
| I+R+U* | mild | 98.90% | 0.4076 | 0.5302 | 4.4417 | 0.8744 | 0.6746 | 0.4624 |
| | moderate | 99.27% | 0.3972 | 0.5223 | 4.4511 | 0.8608 | 0.6611 | 0.4554 |
| | severe | 99.08% | 0.3901 | 0.5153 | 4.5018 | 0.8547 | 0.6530 | 0.4478 |
| I*+R*+U | mild | 96.34% | 0.2163 | 0.3398 | 7.0894 | 0.6063 | 0.4592 | 0.3115 |
| | moderate | 89.56% | 0.1546 | 0.2607 | 8.4029 | 0.4828 | 0.3653 | 0.2655 |
| | severe | 57.88% | 0.0749 | 0.1372 | 10.1472 | 0.3348 | 0.2466 | 0.1663 |
| I+R*+U* | mild | 98.53% | 0.2550 | 0.3824 | 6.4526 | 0.6704 | 0.5060 | 0.3494 |
| | moderate | 96.15% | 0.1602 | 0.2678 | 7.9476 | 0.5319 | 0.3795 | 0.2787 |
| | severe | 95.97% | 0.0850 | 0.1535 | 9.5544 | 0.2074 | 0.2612 | 0.3704 |
| I*+R+U* | mild | 97.07% | 0.3431 | 0.4719 | 5.2047 | 0.8098 | 0.6091 | 0.4124 |
| | moderate | 98.35% | 0.2796 | 0.4048 | 6.1825 | 0.6789 | 0.5240 | 0.3682 |
| | severe | 80.40% | 0.1694 | 0.2784 | 8.5068 | 0.4890 | 0.3712 | 0.2488 |
| I*+R*+U* | mild | 96.70% | 0.2205 | 0.3473 | 6.9820 | 0.6272 | 0.4676 | 0.3154 |
| | moderate | 87.91% | 0.1527 | 0.2556 | 8.4740 | 0.4659 | 0.3505 | 0.2548 |
| | severe | 39.01% | 0.0792 | 0.1427 | 10.2981 | 0.3342 | 0.2508 | 0.1718 |

truth rather than producing random structures. This structural conservatism is an important form of predictable failure and provides useful interpretability for chemists.

As the noise level increases from mild to moderate to severe, or as the number of corrupted modalities grows from one to two to three, all metrics continue to decline. For example, when all three spectra are severely corrupted, validity drops to 57.88%, the Tanimoto score falls to 0.0749, and functional group accuracy decreases to 0.3348. This pattern is fully consistent with intuition: the mapping $\phi$ from spectra to textual peak representations is the sole information interface between the raw data and the language model, so stronger perturbations or loss of peak fidelity directly weaken the chemical constraints available to the LLM.

These observations support three key conclusions:

1. The peak-extraction-based mapping $\phi$ is robust under realistic noise conditions.

2. As spectra become increasingly corrupted, the model degrades in a predictable and graceful manner rather than collapsing abruptly.

3. The practical performance boundary of SpectraLLM is determined by simultaneous severe disruption of multiple modalities, rather than by a single low-quality spectrum.

## A.9 MODALITY CONTRIBUTIONS AND MULTISPECTRAL SYNERGY

In Table 4, we have already conducted a preliminary ablation from single-spectrum inputs to multi-spectral joint inputs on the test data, and observed that joint inference with multiple spectra indeed leads to better performance. To perform a more systematic spectral-modality ablation, we further train a separate model for every modality combination in the QM9s dataset, so as to avoid the distribution shift that may arise from simply masking inputs. Concretely, QM9s contains three spectra, IR, Raman, and UV-Vis; we train one model for each single-spectrum setting, one for each pairwise combination among IR, Raman, and UV-Vis, and one for the three-spectrum combination, yielding in total 7 models, which are evaluated independently. The results are shown in Table 19. The performance differences among single-modality models first reveal the baseline contribution strength of each spectrum to the structure recovery task. Among them, the Raman-Only model performs the best (Tanimoto = 0.3251), far exceeding IR-Only and UV-Vis-Only, which is consistent with the strong structural constraints Raman provides on carbon skeleton vibrations and symmetry modes. In contrast, UV-Vis used alone carries limited information, making it difficult to provide

stable bond-order or functional-group constraints; consequently, it performs the weakest on almost all structural metrics.

When we further examine the bimodal combinations, the performance gain of IR+Raman is the most significant (Tanimoto = 0.5320), achieving more than 60% relative improvement over the best single-modality Raman model. This strong synergy indicates that the two spectra offer highly complementary structural information: Raman mainly encodes backbone and symmetry, while IR is particularly sensitive to polar functional groups and local environments. When both are provided as input, the model can jointly lock down the global molecular framework and the positions of key functional groups, substantially shrinking the uncertainty of the chemical space. Other bimodal combinations (such as Raman+UV-Vis or IR+UV-Vis) also yield consistent gains, but the magnitude is smaller than that of IR+Raman, further indicating that the marginal contribution of UV-Vis to structure recovery is weaker than that of the two vibrational spectra.

Interestingly, although the three-spectra input (IR+Raman+UV–Vis) provides the most complete coverage, its performance is slightly worse than that of the IR+Raman combination. For example, its Tanimoto score reaches only 0.4053, clearly below the 0.5320 achieved by IR+Raman. This phenomenon suggests that, on clean synthetic spectra, UV–Vis information does not always yield additional benefits and can in some cases introduce noise or redundant cues, increasing the burden of aligning multiple spectroscopic signals. In other words, incorporating more spectra does not guarantee monotonically improved performance; rather, the effectiveness of fusion depends on the complementarity and information correlation among the spectra being combined.

Table 19: Effect of IR, Raman, and UV-Vis modality combinations on spectrum-to-structure performance on QM9s.

| SpectraLLM Q9MS retrained | Validity ↑ | Tanimoto ↑ | Tanimoto (MACCS) ↑ | Cosine ↑ | MCES ↓ | Functional Group ↑ | Fraggle ↑ |
|---|---|---|---|---|---|---|---|
| IR-Only | 100.00% | 0.1474 | 0.3436 | 0.2462 | 7.1245 | 0.5309 | 0.2686 |
| Raman-Only | 98.90% | 0.3251 | 0.5748 | 0.4490 | 5.4009 | 0.7986 | 0.3958 |
| UV-Vis-Only | 100.00% | 0.0777 | 0.2081 | 0.1407 | 10.3874 | 0.3600 | 0.2137 |
| IR+Raman | 99.63% | 0.5320 | 0.7363 | 0.6288 | 3.0312 | 0.8834 | 0.4806 |
| IR+UV-Vis | 99.63% | 0.2088 | 0.4471 | 0.3280 | 6.0551 | 0.6791 | 0.3333 |
| Raman+UV-Vis | 99.45% | 0.4594 | 0.6718 | 0.5699 | 4.0405 | 0.8431 | 0.4561 |
| IR+Raman+UV-Vis | 98.53% | 0.4053 | 0.6501 | 0.5202 | 3.9926 | 0.8370 | 0.4479 |

On the other hand, we summarize the train–test transfer performance across different spectral combinations in Table 20 to further analyze the generalization patterns and spectrum robustness of multi-spectral models. Overall, when the test spectra match the spectra seen during training, the model typically achieves the highest Tanimoto similarity; once the input spectra differ from those observed during training, performance drops sharply and can in some cases approach random. This indicates that the inverse mapping from spectra to structures does not naturally share a modality-invariant latent space; instead, the mapping heavily depends on spectrum-specific structural constraints, which also explains why single-spectrum models generally fail under cross-spectrum transfer.

At the same time, training with multiple spectra yields substantially stronger robustness under cross-spectrum testing. For example, the model trained on IR+Raman not only reaches the highest Tanimoto score (0.532) when evaluated on the same spectral pair, but also remains significantly better than single-spectrum models when tested with only IR or only Raman as input. This suggests that multi-spectral training helps the model acquire a more unified and adaptable structural prior, enabling it to maintain stable prediction capability even when certain spectra are absent. A similar pattern is observed for the IR+Raman+UV–Vis model: joint training across all spectra provides a more complete representational basis, allowing the model to partially recover structural information under missing-spectrum or mismatched-spectrum conditions.

However, this robustness is clearly directional. When evaluated on UV–Vis alone, all models—regardless of training spectra—exhibit limited performance, indicating that UV–Vis inherently provides weaker constraints for structure recovery and that its predictive utility cannot be substantially improved via transfer from other spectra. This again reflects the unequal information density across spectroscopic signals and explains why most multi-spectral gains originate from IR and Raman rather than from simply increasing the number of spectra.

Overall, the transfer matrix supports two key conclusions: (i) single-spectrum models rely strongly on spectrum-specific cues and struggle with cross-spectrum generalization; (ii) multi-spectral training substantially improves robustness, allowing the model to retain structure recovery capability under diverse spectral combinations and missing-spectrum conditions. These observations align with the independent-training ablations discussed above and further highlight the necessity and practical advantages of building multi-spectral models that remain functional even under incomplete spectroscopic inputs.

Table 20: Train-test modality transfer matrix on QM9s (Tanimoto similarity).

| test\train | IR | Raman | UV-Vis | IR+Raman | IR+UV-Vis | Raman+UV-Vis | IR+Raman+UV-Vis |
|---|---|---|---|---|---|---|---|
| IR | 0.1474 | 0.0674 | 0.0605 | 0.2012 | 0.2012 | 0.0629 | 0.1921 |
| Raman | 0.0669 | 0.3251 | 0.0721 | 0.3951 | 0.0639 | 0.4112 | 0.2977 |
| UV-Vis | 0.0592 | 0.0632 | 0.0777 | 0.0579 | 0.086 | 0.0877 | 0.0812 |
| IR+Raman | 0.1372 | 0.1209 | 0.0671 | 0.532 | 0.1468 | 0.1319 | 0.4231 |
| IR+UV-Vis | 0.1379 | 0.0642 | 0.0704 | 0.1803 | 0.2088 | 0.0645 | 0.1962 |
| Raman+UV-Vis | 0.0671 | 0.2746 | 0.0741 | 0.3687 | 0.0786 | 0.4594 | 0.3353 |
| IR+Raman+UV-Vis | 0.1373 | 0.1161 | 0.0706 | 0.4922 | 0.17 | 0.1305 | 0.4213 |

## A.10 IMPACT OF METADATA AND EXPERIMENTAL CONDITIONS ON STRUCTURE PREDICTION

Experimental metadata, such as spectral acquisition conditions, sample descriptions, or collision energies, often play a critical role in both spectral patterns and molecular structure prediction. To quantitatively assess the model's dependence on such metadata, we design an ablation setting denoted as `"-wo experiment"`. Specifically, we remove all experimental metadata from the prompts, as shown in Table 21, and then retrain the model on this new prompt corpus, and then evaluate it on each single-spectrum and multi-spectral test set. The results are reported in Table 22. Overall, the full model with experimental conditions consistently outperforms the variant without them, with even larger gains observed under cross-spectrum combined inputs.

This effect is particularly pronounced for mass spectrometry tasks. For example, on the MassBank dataset, the full model achieves an average Tanimoto similarity of 0.1306, which drops to 0.1259 when experimental metadata are removed. The underlying reason is that the core of mass spectrometry lies in fragmentation patterns, while experimental parameters such as collision energy directly influence the formation of fragment peaks and their intensity distributions. SpectraLLM can encode these experimental conditions into the prompt and, during autoregressive generation, jointly condition on both the spectral peaks and the experimental settings. For instance, when the prompt indicates a low collision energy, the model expects weaker, low-mass fragment peaks, whereas a higher collision energy corresponds to more intense, deeper fragmentation peaks.

For multi-spectral joint training tasks, experimental metadata similarly act as a bridge, enabling the model to share contextual information across different spectral types. When IR and Raman spectra are provided jointly, for example, the model can exploit their respective characteristic peaks while using metadata to perform conditional reasoning over spectral patterns, thus improving the accuracy of the joint prediction. This suggests that language-based metadata prompts not only compensate for potential precision loss introduced by discretizing spectral peaks but also substantially enhance the model's reasoning ability across spectra and experimental conditions.

Taken together, this series of ablations demonstrates that SpectraLLM successfully integrates non-numerical context into the structure prediction pipeline. By expressing experimental conditions in natural language, the model can map diverse spectra and measurement parameters into a shared semantic space, offering a new and general computational paradigm for complex spectral data analysis. This highlights the potential of multi-spectral large language models for scientific data reasoning, establishing a powerful framework for cross-spectral analysis and complex data interpretation.

## A.11 ZERO-SHOT LIMITATIONS OF PRETRAINED LLMS ON SPECTRAL INTERPRETATION

To rule out the possibility that the base large language model may have indirectly acquired chemical-structure knowledge during pre-training—potentially causing knowledge leakage or spurious performance gains—we conduct a strict zero-shot evaluation on the pretrained Qwen3-32B model without

Table 21: Prompt Design Examples (Conditional vs. Unconditional)

**Conditional Prompt**

```
{
  "prompt": "Mass spectrum data: {\"mzs\":
      \"91.05,125.02,154.05,155.06,185.1,200.11,229.09,246.11\", \"
      intensities\": \"0.245,1.0,0.08,0.355,0.349,0.045,0.142,0.735\"}\
      nAdduct type: [M+H]+\nInstrument: Orbitrap\nCollision energy: 30.0
      eV\nPlease predict the compound's SMILES representation with LESS
      THAN 1000 characters thinking. The final output must strictly begin
       with '#SMILES:'",
  "response": "#SMILES: CC(=O)N[C@@H](CC1=CC=CC=C1)C2=CC(=CC(=O)O2)OC",
  "system": "You are a professional mass spectrometry analysis model.
      Your task is to predict the compound's SMILES based on the given
      mass spectrum data. The input contains m/z and intensity list, as
      well as experimental conditions (such as adduct type, instrument
      type, collision energy, and precursor m/z). Please predict the
      compound's SMILES representation."
}
```

**Unconditional Prompt**

```
{
  "prompt": "Mass spectrum data: {\"mzs\":
      \"91.05,125.02,154.05,155.06,185.1,200.11,229.09,246.11\", \"
      intensities\": \"0.245,1.0,0.08,0.355,0.349,0.045,0.142,0.735\"}\
      nPlease predict the compound's SMILES representation with LESS THAN
       1000 characters thinking. The final output must strictly begin
      with '#SMILES:'",
  "response": "#SMILES: CC(=O)N[C@@H](CC1=CC=CC=C1)C2=CC(=CC(=O)O2)OC",
  "system": "You are a professional mass spectrometry analysis model.
      Your task is to predict the compound's SMILES based on the given
      mass spectrum data. The input contains m/z and intensity list.
      Please predict the compound's SMILES representation."
}
```

Table 22: Effect of removing experimental metadata from the prompt on spectrum-to-structure prediction across datasets.

| SpectraLLM Qwen3-32B | Validity ↑ | Tanimoto ↑ | Tanimoto (MACCS) ↑ | MCES ↓ | Cosine ↑ | Functional Group ↑ | Fraggle ↑ |
|---|---|---|---|---|---|---|---|
| QM9s Jointly | 99.45% | 0.4586 | 0.5639 | 3.7164 | 0.8659 | 0.6785 | 0.4621 |
| -wo experiment | 99.82% | 0.4408 | 0.5477 | 3.8339 | 0.8459 | 0.6608 | 0.4607 |
| Multi Jointly | 99.57% | 0.5750 | 0.6638 | 5.4829 | 0.8448 | 0.7700 | 0.6740 |
| -wo experiment | 98.94% | 0.5527 | 0.6446 | 5.7462 | 0.8279 | 0.7524 | 0.6658 |
| MassBank | 98.46% | 0.1306 | 0.2264 | - | 0.4540 | 0.3848 | 0.3201 |
| -wo experiment | 98.62% | 0.1259 | 0.2215 | - | 0.4336 | 0.3791 | 0.3092 |

any spectrum-related fine-tuning. Specifically, we directly feed spectral text prompts into the base model and evaluate its structure generation ability on the QM9s Jointly and Multi Jointly test sets. The results are presented in Table 23.

Overall, the unfine-tuned model is unable to recover any meaningful structural information from spectra. On QM9s Jointly, the Tanimoto similarity is only 0.0308, and all structural metrics remain near random. On the more heterogeneous Multi Jointly set, performance degrades even further: all structure-recovery metrics (Tanimoto, MACCS, Fraggle) collapse to 0, and distance-based metrics such as MCES indicate severe deviation from the ground truth. This shows that the base LLM essentially lacks the ability to map spectral signals to molecular structures, yielding chemically inconsistent outputs.

These findings indicate that although a pretrained LLM may have encountered chemical text or isolated molecular representations, such knowledge does not transfer to the spectrum-to-structure inverse problem. In other words, the base model does not understand spectra and cannot exploit pretraining statistics for structure recovery. The performance gains of SpectraLLM therefore stem almost entirely from spectrum–text alignment during fine-tuning rather than retrieval or memorization of pretraining knowledge. Consequently, spectrum-to-structure generation is not a conventional language modeling task that benefits from corpus-level generalization, but a cross-modal inverse problem that must be learned through dedicated training.

Table 23: Zero-shot performance of the pretrained Qwen3-32B model on spectrum-to-structure prediction.

| SpectraLLM Qwen3-32B | Validity ↑ | Tanimoto ↑ | Tanimoto (MACCS) ↑ | Cosine ↑ | MCES ↓ | Functional Group ↑ | Fraggle ↑ |
|---|---|---|---|---|---|---|---|
| QM9s Jointly | 0.92% | 0.0308 | 0.1963 | 0.0595 | 7.6000 | 0.2567 | 0.0157 |
| Multi Jointly | 0.43% | 0.0000 | 0.0000 | 0.0000 | 29.0000 | 0.0000 | 0.0000 |

## A.12 TRAINING AND INFERENCE EFFICIENCY OF SPECTRALLM

To comprehensively evaluate computational efficiency, we compare lightweight spectrum-specific models (IR-to-Structure, Spectra2Structure, NMR2Struct, Spec2Mol) with the large-scale multi-spectral SpectraLLM. As shown in Table 24, we report training compute in FLOPs ($10^{15}$), using Spectra2Structure as the 1.0× baseline. Lightweight models incur relatively modest training costs; for example, IR-to-Structure requires 2.81× the baseline FLOPs, while NMR2Struct requires 53.48×. In contrast, fine-tuning SpectraLLM on multi-spectral data demands $429{,}655.20 \times 10^{15}$ FLOPs—approximately $4.3 \times 10^6 \times$ the baseline—primarily due to its 32B parameters and long multi-spectral input sequences.

For inference (Table 25), we measure per-sample latency (ms/sample) and throughput (samples/s), using Spec2Mol (6.02 ms/sample) as the 1.0× baseline. Spectrum-specific models achieve millisecond-level latency suitable for high-throughput use. In contrast, SpectraLLM exhibits an average latency of 5.78 s/sample, corresponding to a 959× overhead. This slowdown stems mainly from autoregressive decoding and the large parameter count. We also observe spectrum-dependent differences; for example, Raman spectra typically require more decoding steps than IR, reflecting differing spectral complexity. Nevertheless, SpectraLLM delivers unified multi-spectral reasoning and significantly higher accuracy than its lightweight counterparts—making it suitable for low-throughput, high-precision analytical scenarios.

Overall, while SpectraLLM incurs substantially higher computational costs compared to spectrum-specific models, these costs are offset by improvements in cross-spectral generality and structural prediction accuracy. For large-scale spectral analysis, practitioners can balance efficiency and accuracy based on application needs, with lightweight models remaining preferable for high-throughput settings.

## A.13 BAD CASE STUDY

In addition to the quantitative ablations, we performed qualitative stress tests by manually deleting chemically meaningful fragment peaks from single-spectrum mass spectrometry (MS) examples.

Table 24: Training compute and wall-clock cost for SpectraLLM and modality-specific baselines.

| Datasets | Model | #Samples (train/val) | GPUs used | Batch | epoch | Approx FLOPs ($10^{15}$) | Wall-clock Time (h) | Training Cost (relative) |
|---|---|---|---|---|---|---|---|---|
| QM9s, (IR-only, Raman-only, UV-only) | IR-to-Structure | 105,079 / 11,676 | 1x RTX 3090 24G | 4096 | 200 | 0.2566 | 24.73h | 2.81× |
| | Spectra2Structure | 105,079 / 11,676 | 1x A100 80GB | 800 | 136 for IR 334 for Raman 157 for UV | 0.0914 | 9.25 h for IR 22.71 h for Raman 10.68 h for UV | 1.00× (baseline) |
| Multimodal Spectroscopic (NMR-only) | NMR2Struct | 643,604 / 71,512 | 1x RTX 3090 24G | 256 | 500 | 4.888 | 24 h | 53.48× |
| Combined | SpectraLLM | 2,367,865 / - | 4x H100 80G | 16 | 1 | 429,655.20 | 100h | 4.7008e+06× |

Table 25: Inference latency, throughput, and relative cost across SpectraLLM and baseline models.

| Datasets | Model | #Sample | Inference latency | | | Inference Cost |
|---|---|---|---|---|---|---|
| | | | Avg Latency (ms/sample) | Throughput (samples/s) | Params (B) | |
| QM9s (IR-only, Raman-only, UV-only) | IR-to-Structure Spectra2Structure | 13,062 13,062 | 16.06 7.49 | 62.27 133.46 | 0.069 0.024 | 2.67× 1.24× |
| Multimodal Spectroscopic NMR-only | NMR2Struct | 79287 | 119.05 | 8.4 | 0.0136 | 19.78× |
| MassSpecGym Mass-only | Spec2Mol Diffms | 16264 16264 | 73.03 2700.97 | 13.15 0.37 | 0.0518 0.0849 | 12.13× 448.67× |
| MassBank Mass-only | Spec2Mol Diffms | 7905 7905 | 46.32 873.97 | 21.59 1.14 | 0.0518 0.0849 | 7.69× 145.18× |
| Multimodal Spectroscopic Mass-only | Spec2Mol Diffms | 79287 79287 | 6.02 2800.72 | 166.20 0.36 | 0.0518 0.0849 | 1.00× (baseline) 465.24× |
| Combined | SpectraLLM | depends on subsets | 5778.39 | 0.17 | 32 | 959.86× |

This approach allowed us to assess how the model performs under conditions where key spectral features are missing, testing its reliance on spectral information versus its ability to infer molecular structures. Two representative cases are shown in Fig 5, each illustrating a situation where important fragment peaks are removed, leading to significant changes in the model's predictions. In the first case, the ground-truth molecule is the branched ester CC(C)OC(=O)C(C)C, which exhibits a prominent fragment at m/z = 43, corresponding to an isopropyl cation. This fragment is a hallmark of esters containing isopropyl groups. When this peak is removed, the model fails to predict an ester structure, instead generating a straight-chain aldehyde CCCCCC=O, which retains a terminal carbonyl group but lacks the ester motif and the branched isopropyl substituent. This demonstrates that the model is heavily reliant on this particular fragment for its predictions, underscoring the importance of specific fragment peaks in its reasoning process. In the second case, the ground-truth molecule CN(C)C(C)=O (N,N-dimethylacetamide) also produces a strong fragment at m/z = 43, indicative of the acetyl cation $[CH_3CO]^+$. After removing this peak, the model predicts $[N-]=[N^+]=NC[C@@H]1CN1$, a nitrogen-rich heterocycle that fits the overall mass of the molecule but loses the oxygen-containing fragment characteristic of m/z = 43. This suggests that the model has learned to associate certain fragment peaks with specific functional groups and substructures, and when key peaks are missing, its predictions shift accordingly.

These controlled failures provide important insights into how SpectraLLM processes spectral data. The model's "incorrect" predictions are not random—they systematically avoid the specific functional motifs (isopropyl ester, acetyl carbonyl) that correspond to the fragment peaks we have artificially removed, while retaining other spectral aspects such as mass and heteroatom composition. This behavior indicates that SpectraLLM relies on chemically interpretable spectral features rather than merely exploiting superficial dataset biases.

In the next case study (Fig 6), we performed a systematic analysis of failure modes across QM9S, MassSpecGym, MassBank and the Multimodal Spectroscopic dataset to understand where SpectraLLM still struggles and why these failures depress structure-level metrics (e.g. Tanimoto). The empirical bad cases cluster into three dominant categories that reflect fundamentally distinct sources of uncertainty: intrinsic spectral ambiguity among isomers, high structural complexity with rare or congested fragmentation patterns, and a small fraction of output errors traceable to SMILES formatting or decoding failures. Below we examine each category in turn and explain how the spectroscopic physics, dataset statistics, and sequence-generation mechanics conspire to produce the observed prediction errors.

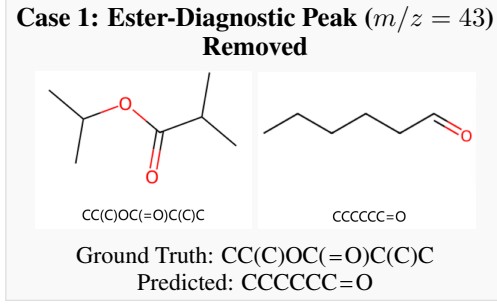 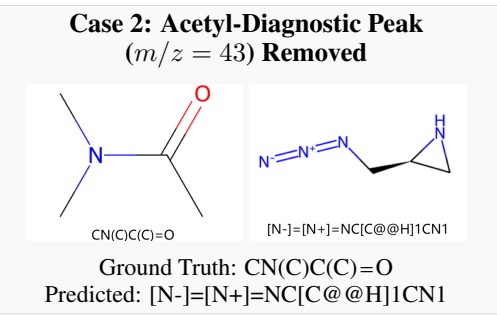

**Case 1: Ester-Diagnostic Peak ($m/z = 43$) Removed**

CC(C)OC(=O)C(C)C     CCCCCC=O

Ground Truth: CC(C)OC(=O)C(C)C
Predicted: CCCCCC=O

**Case 2: Acetyl-Diagnostic Peak ($m/z = 43$) Removed**

CN(C)C(C)=O     [N-]=[N+]=NC[C@@H]1CN1

Ground Truth: CN(C)C(C)=O
Predicted: [N-]=[N+]=NC[C@@H]1CN1

Figure 5: Qualitative counterfactual examples obtained by deleting diagnostic MS fragment peaks. Left group: Ground-truth molecule CC(C)OC(=O)C(C)C (left) and SpectraLLM prediction CCCCCC=O (right) after removing the ester-diagnostic m/z = 43 isopropyl cation peak from the input spectrum. The model preserves a terminal carbonyl but no longer predicts an ester or branched isopropyl substituent. Right group: Ground truth CN(C)C(C)=O (left) and prediction $[N-]=[N^+]=NC[C@@H]1CN1$ (right) after removing the m/z = 43 $[CH_3CO]^+$ acetyl fragment. The predicted structure matches the overall mass but eliminates oxygen-containing fragments consistent with m/z = 43.

The largest class of failures originates from intrinsic ambiguity: many constitutional isomers and positional isomers produce highly similar spectral fingerprints, and mass fragmentation in particular cannot uniquely disambiguate certain rearrangements or alkyl-chain permutations. Examples from Case1–Case3 illustrate this: spectra from structural isomers (e.g., $C_6H_{14}$ isomers, positional alcohol isomers, or different substitution patterns on an aromatic ring) frequently share almost identical sets of fragment masses and vibrational features, so multiple chemically plausible SMILES strings yield near-identical spectral likelihoods. In these situations the model often outputs a chemically valid isomer that is spectrally consistent with the measurement but not the reference structure, which directly lowers Tanimoto despite generating reasonable chemistry. This limitation is not a model bug per se but a property of the inverse problem: the measurements lack the information necessary for unique reconstruction. As a result, performance metrics that require exact structural agreement are conservative estimates of practical utility in these regimes.

A second, qualitatively different failure mode arises for high-complexity molecules—polycyclic frameworks, heavily functionalized or heteroatom-rich species, and longer fused-ring or highly polar scaffolds (Cases 4–6). These compounds incur two compounding difficulties. First, their spectra are intrinsically more complex: dense peak forests, overlapping isotopic or adduct signals, strong neutral losses (e.g. phosphate, sugar moieties), and multiple low-abundance fragments that vary strongly with ionization and collision-energy conditions. Under typical MS/IR acquisition the diagnostic peaks that would uniquely indicate a particular scaffold are often missing, suppressed, or convolved with neighboring signals. Second, these chemotypes are under-represented in training corpora, so the model has limited prior experience to draw upon; rare stereochemical motifs and subtle regiochemical differences are therefore not strongly encoded in the learned latent space. Together these factors cause systematic misplacement or omission of key substructures (missed phosphate/sugar groups, incorrect ring fusion patterns, wrong stereochemical assignments), leading to large topological discrepancies captured by MCES and low fragment-level metrics like Fraggle and functional-group recall.

A third category consists of SMILES-formatting or decoding errors that produce syntactically invalid or truncated outputs. These failures typically occur for long or highly branched predictions where ring indices or parentheses are misaligned during autoregressive decoding. While such errors are largely orthogonal to the model's spectroscopic reasoning capabilities, they still impact aggregate statistics because invalid or malformed SMILES are excluded from fingerprint comparisons and therefore reduce measured Tanimoto and MACCS rates. In practice, improving decoding robustness (e.g., constrained decoding, grammar-aware postprocessing, or validity-guided reranking) mitigates a portion of these cases without changing the underlying spectroscopic model.

Taken together, these three categories highlight where future work should focus to close the gap between spectrally plausible predictions and exact-structure recovery. Intrinsic ambiguity implies

## Intrinsic Isomer Ambiguity (QM9s Dataset)

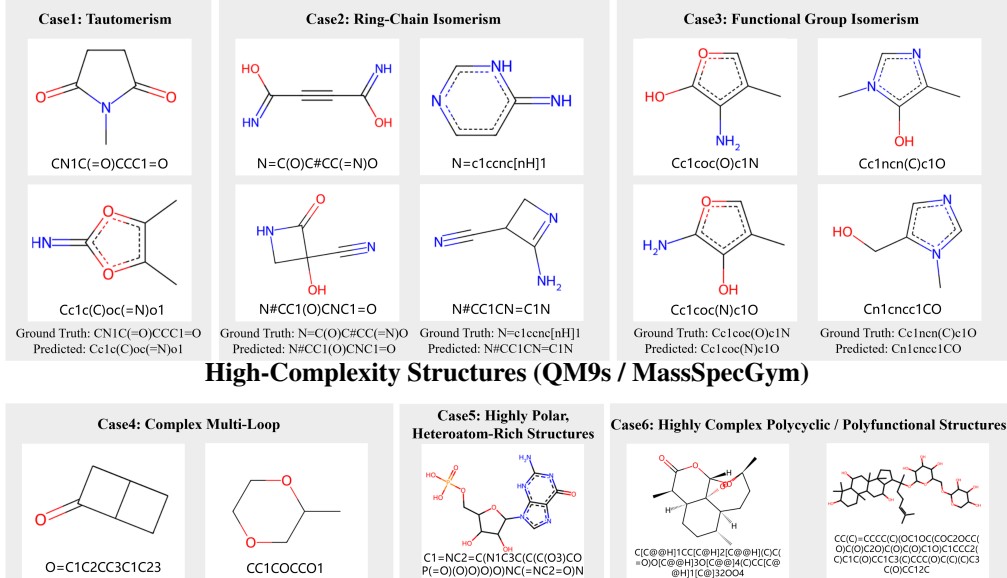

## High-Complexity Structures (QM9s / MassSpecGym)

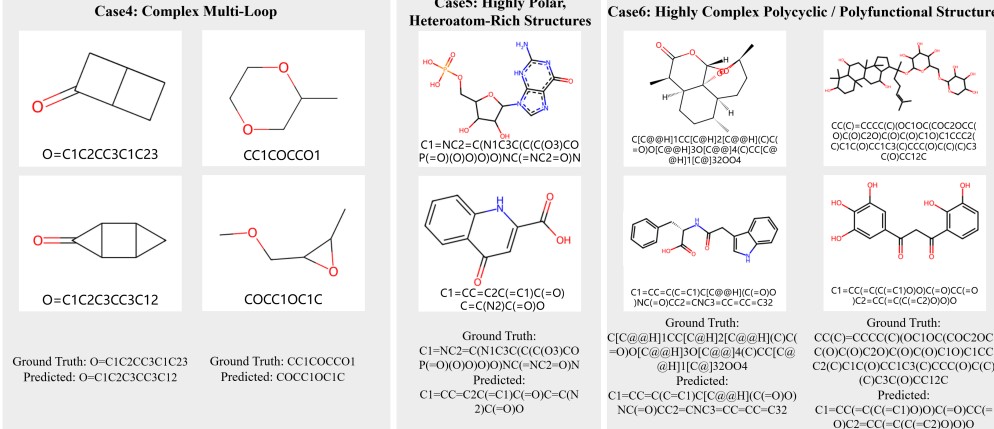

Figure 6: Bad Case Analysis of SpectraLLM Predictions. The upper panel shows bad cases from *intrinsic ambiguity*: Isomers (e.g., $C_6H_{14}$ isomers, alcohol positional isomers, aromatic substitution patterns) have nearly identical spectral features, leading the model to predict chemically reasonable but non-unique isomers (all cases from QM9s). The lower panel shows bad cases from *high-complexity structures*: Errors are more frequent for fused polycyclic molecules, long conjugated systems, or heteroatom-rich structures (case4 from QM9s; case5–6 from MassSpecGym), which are underrepresented in training data and suffer from peak congestion/shift.

a hard information-theoretic ceiling that can only be relaxed by adding orthogonal measurements or experimental metadata; high-complexity failures call for targeted data augmentation, domain-aware pretraining on rare scaffolds, and instrument-conditional modeling to account for acquisition-dependent fragmentation; and decoding errors can be reduced through constrained generation techniques. Understanding these distinct failure modes helps explain why single-number metrics such as Tanimoto remain modest in some regimes: the bottlenecks are a mixture of irreducible measurement ambiguity, dataset coverage, and generation mechanics.

### A.14 GENERALIZATION TO REAL EXPERIMENTAL SPECTRA

To rigorously assess the model's ability to generalize beyond simulated spectra, we evaluate SpectraLLM under two complementary settings involving real experimental data: (1) out-of-distribution reconstruction on MassSpecGym and MassBank after removing these datasets entirely from training, and (2) direct zero-shot transfer to NIST IR and UV-Vis spectra. Together, these experiments reveal the extent to which multi-spectral language modeling enables robust molecular structure prediction from real-world measurements, where spectral noise, instrument variability, and distributional shifts significantly challenge purely simulation-trained models.

We first re-train SpectraLLM using only simulated spectra from QM9s and the Multimodal Spectroscopic Dataset, explicitly excluding all MassSpecGym and MassBank molecules. The retrained model is then evaluated on the original real-world test sets. Table 26 reports detailed comparisons against DiffMS and Spec2Mol under both trained and untrained states. Across both real-world datasets, SpectraLLM consistently outperforms prior methods in all structure-sensitive similarity metrics, particularly Tanimoto similarity, MACCS-based similarity, Fraggle similarity, and functional-group accuracy. The performance gap is most pronounced in the untrained setting, where SpectraLLM—despite never observing these datasets—still reconstructs molecular structures with significantly higher fidelity than both DiffMS and Spec2Mol. These results indicate that SpectraLLM learns a spectra-to-structure mapping that is not tightly bound to the statistical properties of any specific dataset, enabling effective generalization to experimental spectra with unknown acquisition settings and noise characteristics.

Table 26: Model Performance on massbank/massspecgym Experimental Spectra.

| Setting | State | Model | Validity ↑ | Tanimoto ↑ | Cosine ↑ | Functional Group ↑ | Tanimoto (MACCS) ↑ | Fraggle ↑ |
|---|---|---|---|---|---|---|---|---|
| MassSpecGym | trained | Diffms | 57.16% | 0.159705 | 0.242187 | 0.489004 | 0.430529 | 0.353878 |
| | | SpectraLLM | 99.74% | 0.1537 | 0.2565 | 0.5016 | 0.4735 | 0.362 |
| | untrained | Spec2Mol | 62.86% | 0.084861 | 0.151115 | 0.311157 | 0.270927 | 0.206558 |
| | | SpectraLLM | 92.97% | 0.1139 | 0.2039 | 0.4354 | 0.4005 | 0.2988 |
| MassBank | trained | Diffms | 23.63% | 0.074269 | 0.100735 | 0.179588 | 0.156596 | 0.153949 |
| | | SpectraLLM | 98.46% | 0.1306 | 0.2264 | 0.454 | 0.3848 | 0.3201 |
| | untrained | Spec2Mol | 71.63% | 0.085741 | 0.153861 | 0.299915 | 0.241751 | 0.214962 |
| | | SpectraLLM | 92.79% | 0.1125 | 0.2019 | 0.187 | 0.3612 | 0.2922 |

To further evaluate zero-shot transferability, we collected all molecules from NIST [2] using the nistchempy toolkit [3] and applied the same QM9-style filtering protocol used in prior literature (Kanakala et al., 2024). Molecules containing elements outside C, N, O, H, F were removed, resulting in a pool of 73,354 unique QM9-like structures. Among these, 7,177 had downloadable IR spectra and 1,568 had UV-Vis spectra. After removing molecules overlapping with our training set—either exactly or via high MCES similarity—we obtained two challenging, strictly out-of-distribution test sets: 4,024 molecules-IR pairs and 911 molecules-UV-Vis pairs. Statistics are summarized in Table 27.

Across both IR and UV-Vis regimes (Table 28), SpectraLLM exhibits a markedly stronger ability to recover chemically faithful structures than all baseline systems, even under the strict zero-shot setting. Although validity remains uniformly high across methods, the structure-aware metrics—Tanimoto, MACCS, Fraggle, and functional-group accuracy—consistently distinguish SpectraLLM from embedding-driven baselines. On the NIST-IR benchmark, SpectraLLM achieves a Tanimoto similarity of 0.0727, representing an improvement of more than 5× over the strongest

---

[2] https://webbook.nist.gov/chemistry/
[3] https://github.com/IvanChernyshov/NistChemPy

Table 27: Statistics of NIST Dataset (SMILES and Spectral Availability).

| NIST | Total Molecule | QM9s-like Molecule | Download Spectral | Available | |
|---|---|---|---|---|---|
| | | | | **SMILES** | **Spectral** |
| IR | 144795 | 73354 | 7177 | 4024 | 4024 |
| UV-Vis | 144795 | 73354 | 1568 | 911 | 911 |

baseline. This gain is mirrored in MACCS and Fraggle similarity, each reflecting substantially closer fingerprint- and fragment-level agreement with ground-truth structures. A similar pattern holds for the NIST-UV subset. While cosine similarity among all models remains relatively close—likely reflecting the smoother, low-frequency nature of UV-Vis spectra—SpectraLLM continues to deliver the highest structure-level fidelity across all fingerprint- and fragment-based metrics.

Table 28: Model Performance on NIST-IR/UV Experimental Spectra.

| Datasets | Model | Validity ↑ | Tanimoto ↑ | Cosine ↑ | MCES ↓ | Functional Group ↑ | Tanimoto (MACCS) ↑ | Fraggle ↑ |
|---|---|---|---|---|---|---|---|---|
| NIST-ir | IR-to-Structure | 98.02% | 0.0095 | 0.0657 | 33.4585 | 0.1309 | 0.0699 | 0.0720 |
| | Spectra2Structure | 99.31% | 0.0128 | 0.0769 | 29.5700 | 0.2104 | 0.0923 | 0.1267 |
| | SpectraLLM | 99.43% | 0.0727 | 0.1368 | 22.3229 | 0.3191 | 0.1964 | 0.2176 |
| NIST-uv | IR-to-Structure | 99.12% | 0.0673 | 0.1147 | 25.9537 | 0.2799 | 0.1529 | 0.1101 |
| | Spectra2Structure | 99.37% | 0.0726 | 0.1285 | 23.3650 | 0.3058 | 0.1989 | 0.1407 |
| | SpectraLLM | 99.56% | 0.0744 | 0.1381 | 23.0612 | 0.3231 | 0.2084 | 0.1411 |

Across both evaluation settings, SpectraLLM demonstrates strong generalization, robustness to spectral noise, and cross-dataset transferability. We attribute this to its unified language-based representation, which jointly encodes multi-spectral information and molecular structural priors. Compared to architectures optimized for a specific spectral modality or dataset, our model is better equipped to handle previously unseen experimental distributions, distortions absent from the training data, and novel molecular scaffolds not covered in synthetic datasets. Overall, these results indicate that SpectraLLM serves as a highly versatile spectrum-to-structure model, effectively transferring the knowledge learned from synthetic spectra to real experimental measurements.

Table 29: Example prompts for multi-spectral reasoning.

| Multi-spectral Prompts |
| --- |
| **Human:** Given multiple spectra, they are Infrared Spectrum {Wavenumbers:3596.8, 3549.49, 3314.97, 3202.86, 3141.14, 1498.51, 1444.0, 724.0, Intensities:1.0, 0.16, 0.24, 0.12, 0.38, 0.51, 0.65, 0.76, Widths:98.69, 16.5, 77.31, 40.6, 114.88, 42.94, 142.13, 67.04}, Raman spectroscopy {Wavenumbers:2913.0, 2926.0, 2981.0, 3685.0, Intensities:0.75, 1.0, 1.0, 0.61, Widths:4.39, 22.34, 24.45, 10.08}, Ultraviolet-visible spectroscopy {Energies:6.54, 7.52, Intensities:1.0, 0.34, Widths:0.75, 0.37}. All of these spectra are determined by the same compound, with the wavenumber postions in reciprocal centimeters as Wavenumbers, the energy postions in eV as Energies and corresponding intensities as Intensities. Based on the information provided by these spectra, predict which compound the spectra correspond to and give the SMILES of that compound. Please answer strictly in the format ##SMILES: |
| **GPT:** `##SMILES: ON=C1CCCC1` |
| **Human:** Given multiple spectra, they are Carbon-13 Nuclear Magnetic Resonance {C-shifts:146.12, 105.1, 77.93, 42.24, Intensities:1.0, 0.37, 0.92, 0.78}, Proton Nuclear Magnetic Resonance {H-shifts:8.32, 8.31, 6.96, 6.95, 6.95, 6.95, 6.94, 6.94, 6.93, 6.31, 6.31, 6.3, 4.59, 4.57, 4.56, Intensities:0.5, 0.47, 0.14, 0.26, 0.26, 0.2, 0.19, 0.32, 0.16, 0.51, 1.0, 0.53, 0.33, 0.91, 0.34}, Heteronuclear Single Quantum Coherence {H-shifts:4.3, 2.57, 4.94, 6.32, C-shifts:68.39, 27.85, 99.64, 146.03, Intensities:1.0, 1.0, 1.0, 1.0}, Infrared Spectrum {Wave-numbers:3961.98, 3931.96, 3903.95, 3761.87, 3731.85, 3703.84, 3491.72, 3477.71, 3449.69, 3041.47, 2999.44, 2767.32, 2637.24, 2623.24, 2609.23, 2595.22, 1440.58, 1434.57, 1420.57, 1414.56, 1404.56, 1392.55, Intensities:0.21, 0.21, 0.38, 0.92, 0.64, 1.0, 0.13, 0.11, 0.16, 0.11, 0.14, 0.11, 0.13, 0.11, 0.11, 0.18, 0.12, 0.22, 0.16, 0.1, 0.16, 0.18, Widths:3.79, 3.15, 2.35, 3.7, 4.26, 4.22, 3.72, 2.37, 2.39, 4.23, 2.78, 2.49, 3.85, 3.15, 3.03, 2.76, 6.27, 4.73, 3.96, 3.23, 2.65, 4.28}, Mass spectrum data {mzs:39.02, 41.04, 45.03, 53.04, Intensities:0.28, 1.0, 0.22, 0.23}. All of these spectra are determined by the same compound, with the wavenumber postions in reciprocal centimeters as Wavenumbers, the energy postions in eV as Energies and corresponding intensities as Intensities. Based on the information provided by these spectra, predict which compound the spectra correspond to and give the SMILES of that compound. Please answer strictly in the format ##SMILES: |
| **GPT:** `##SMILES: C1=COCC1` |

Table 30: Representative prompts for single-modality spectroscopic inference.

| Single-spectrum Prompts |
|---|
| **Human:** Given **Infrared** Spectrum {Wavenumbers: 3596.8, 3549.49, 3314.97, 3202.86, 3141.14, 1498.51, 1444.0, 724.0, Intensities: 1.0, 0.16, 0.24, 0.12, 0.38, 0.51, 0.65, 0.76, Widths: 21.23, 9.14, 15.3, 11.94, 15.47, 17.52, 25.88, 15.47}, the spectra data includes the wavenumber positions in reciprocal centimeters as Wavenumbers and corresponding intensities as Intensities, corresponding width as Widths. Based on the information provided, predict which compound the spectra correspond to and give the SMILES of that compound. Please answer strictly in the format ##SMILES: |
| **GPT:** ##SMILES: ON=C1CCCC1 |
| **Human:** Given **Raman** spectroscopy {Wavenumbers: 2926.0, 2981.0, 3685.0, Intensities: 1.0, 1.0, 0.61, Widths: 22.34, 24.45, 10.08}, the spectra data includes the wavenumber positions in reciprocal centimeters as Wavenumbers and corresponding intensities as Intensities, correspond width as Widths. Based on the information provided, predict which compound the spectra correspond to and give the SMILES of that compound. Please answer strictly in the format ##SMILES: |
| **GPT:** ##SMILES: ON=C1CCCC1 |
| **Human:** Given **Ultraviolet-visible** spectroscopy {Energies: 6.54, 7.52, Intensities: 1.0, 0.34, Widths: 0.75, 0.37}, the spectra data includes the energy positions in eV as Energies and corresponding intensities as Intensities, corresponding width as Widths. Based on the information provided, predict which compound the spectra correspond to and give the SMILES of that compound. Please answer strictly in the format ##SMILES: |
| **GPT:** ##SMILES: ON=C1CCCC1 |
| **Human:** Given **Carbon-13 Nuclear Magnetic Resonance** {C-shifts: 146.12, 105.1, 77.93, Intensities: 1.0, 0.37, 0.92}, the spectra data includes the Chemical Shift positions in ppm as C-shifts and corresponding intensities as Intensities. Based on the information provided, predict which compound the spectra correspond to and give the SMILES of that compound. Please answer strictly in the format ##SMILES: |
| **GPT:** ##SMILES: C1=COCC1 |
| **Human:** Given **Proton Nuclear Magnetic Resonance** {H-shifts: 8.32, 8.31, 6.96, 6.95, 6.95, 6.95, 6.94, 6.93, 6.31, 6.31, 6.3, 4.59, Intensities: 0.5, 0.47, 0.14, 0.26, 0.26, 0.2, 0.19, 0.51, 1.0, 0.53, 0.33}, the spectra data includes the Chemical Shift positions in ppm as H-shifts and corresponding intensities as Intensities. Based on the information provided, predict which compound the spectra correspond to and give the SMILES of that compound. Please answer strictly in the format ##SMILES: |
| **GPT:** ##SMILES: C1=COCC1 |
| **Human:** Given **Heteronuclear Single Quantum Coherence** {H-shifts: 4.3, 2.57, 4.94, 6.32, C-shifts: 68.39, 27.85, 99.64, 146.03, Intensities: 1.0, 1.0, 1.0, 1.0}, the spectra data includes the Chemical Shift positions in ppm as C-shifts, H-shifts and corresponding intensities as Intensities. Based on the information provided, predict which compound the spectra correspond to and give the SMILES of that compound. Please answer strictly in the format ##SMILES: |
| **GPT:** ##SMILES: C1=COCC1 |
| **Human:** **Mass** spectrum data: {mzs: 134.1, 202.08, 216.14, 244.13, 266.11, 284.12, 384.18, 402.19, 545.26, 645.31, intensities: 0.056, 0.039, 0.036, 0.05, 0.204, 0.111, 0.999, 0.06, 0.119, 0.351} Adduct type: $[M + Na]^+$ Collision energy: 35(NCE). The final output must strictly begin with ##SMILES: |
| **GPT:** ##SMILES: CC(C)[C@@H]1C(=O)N |

