# OpenReview forum: "SpectraLLM: Uncovering the Ability of LLMs for Molecular Structure Elucidation from Multi-Spectral Data"
_ICLR.cc/2026/Conference — ICLR 2026 Poster_

### Official Review · Reviewer_m8ht · 2025-10-29

**Soundness:** 3
**Presentation:** 2
**Contribution:** 2
**Rating:** 4
**Confidence:** 3

**Summary:**

This paper introduces SpectraLLM, a large language model designed for molecular structure elucidation using multiple spectra. The model tries to address limitations of single modality approaches by unifying diverse spectroscopic data into a shared language space. The method works by converting spectral peak features into natural language prompts. It then uses a frozen Qwen3 backbone adapted with LoRA to autoregressively generate SMILES strings. The authors evaluate this on four public datasets. The results show state-of-the-art performance and demonstrate that combining spectra improves prediction accuracy.

**Strengths:**

1. The experimental evaluation is comprehensive. The model was tested on four large-scale datasets, which include both simulated and experimental data.

2. SpectraLLM shows strong performance. It consistently surpasses specialized baselines even in single modality tasks. The improvement over the NMR2Struct baseline in Table 2 is particularly notable.


3. The analysis in Appendix A.4 provides a good justification for the chosen architecture. The comparison against a vision language model pipeline confirms the benefit of the purely language-based representation for this task.

**Weaknesses:**

1. There is a slight concern about the dependency on the upstream peak extraction process. The model performance seems tied to the quality of this peak list. This step could be an information bottleneck if important signals are missed or noise is included from the raw spectra.


2. While the model achieves state-of-the-art results, the absolute performance metrics suggest the task remains very difficult. For instance, the best Tanimoto score in Table 4 is approximately 0.4875. This indicates significant room for future improvement.

3. The paper mentions including experimental metadata in the prompts, such as collision energy. This is a good idea, particularly for mass spectrometry. However, the paper might benefit from a more direct analysis or ablation study showing the specific impact of this metadata on performance

**Questions:**

1.  Could the authors please comment on the robustness of SpectraLLM to variations in the peak picking step? For example, how much do errors or changes in peak detection thresholds affect the final SMILES generation?

2.  The Tanimoto scores are strong relative to baselines, but the task is clearly not solved. In the authors' view, what is the primary remaining bottleneck? Is it the spectral representation, the reasoning capacity of the language model, or the inherent ambiguity in the spectra?

3. The paper notes the inclusion of experimental conditions. Could the authors provide a brief analysis of the specific contribution of this metadata, especially collision energy for MS, to the model's accuracy?

4. The model is trained on a combination of simulated and experimental datasets. Was any analysis done on generalization? For instance, how well does a model trained only on simulated data perform on the experimental test sets?

---

> ### Author Response · Authors · 2025-11-22
> **Response to Reviewer m8ht (1/3)**
>
> We would like to sincerely thank you for taking the time to review our submission and for providing many thoughtful comments and suggestions. Below we respond to each of your questions in detail, and we hope our clarifications will address your concerns and increase your confidence in our work.
>
> ### There is a slight concern about the dependency on the upstream peak extraction process. The model performance seems tied to the quality of this peak list. This step could be an information bottleneck if important signals are missed or noise is included from the raw spectra. Could the authors please comment on the robustness of SpectraLLM to variations in the peak picking step? For example, how much do errors or changes in peak detection thresholds affect the final SMILES generation?
> We appreciate the reviewer's question on how sensitive our results are to the noise in spectra. To assess the sensitivity of the mapping ϕ to the accuracy of spectrum-to-text conversion and the influence of peak extraction on final test performance, we conducted a series of analyses. We first injected synthetic noise directly into the test-set spectra to examine the robustness of our model under perturbations. Three levels of noise were applied to the IR, Raman, and UV spectra in QM9S, with the specific parameterizations reported in Table 15 of the updated PDF.
>
> We designed 21 noise-injection configurations for multispectral prompts containing IR, Raman, and UV simultaneously. These include: applying three noise intensities to only one spectrum (e.g., IR) while keeping the other two clean; applying noise to two spectra while keeping one clean; and applying the same noise level to all three spectra. After noise injection, we reconstructed the corrupted test set using a dynamic filtering threshold equal to 1% of the maximum spectral intensity. For brevity, we present three representative configurations below, while Table 16 in the updated PDF includes nine representative patterns.
>
> The robustness results clearly show that performance degradation is monotonic with respect to both noise intensity and the number of perturbed spectra. For instance, under mild noise in a single modality (e.g., noisy IR with clean Raman and UV), performance declines only marginally relative to the clean setting. As noise becomes moderate or severe—or as more modalities are perturbed—the Tanimoto similarity decreases and MCES increases accordingly. Importantly, the model avoids catastrophic failure by leveraging undisturbed orthogonal spectral information for cross-modal compensation. Moreover, even when the model fails, the increase in MCES is relatively gentle, indicating that predictions tend to preserve the core molecular scaffold rather than collapsing into random outputs. This demonstrates a predictable and structurally conservative failure mode.
>
> |SpectraLLM QM9S Jointly|Noisy|Validity↑|Tanimoto↑|Cosine↑|MCES↓|Functional Group↑|Tanimoto(MACCS)↑|Fraggle↑|
> |-|-|-|-|-|-|-|-|-|
> |I+R+U|-|98.72%|0.3355|0.4560|4.9647|0.7934|0.5785|0.4117|
> |I*+R+U|mild|96.89%|0.3365|0.4659|5.3299|0.7819|0.5955|0.4011|
> ||moderate|97.99%|0.2688|0.3963|6.2112|0.6851|0.5212|0.3598|
> ||severe|87.18%|0.1712|0.2799|8.4433|0.5010|0.3774|0.2556|
> |I*+R*+U|mild|96.34%|0.2163|0.3398|7.0894|0.6063|0.4592|0.3115|
> ||moderate|89.56%|0.1546|0.2607|8.4029|0.4828|0.3653|0.2655|
> ||severe|57.88%|0.0749|0.1372|10.1472|0.3348|0.2466|0.1663|
> |I*+R*+U*|mild|96.70%|0.2205|0.3473|6.9820|0.6272|0.4676|0.3154|
> ||moderate|87.91%|0.1527|0.2556|8.4740|0.4659|0.3505|0.2548|
> ||severe|39.01%|0.0792|0.1427|10.2981|0.3342|0.2508|0.1718|
>
> We also evaluated how the peak-filtering threshold affects model performance. On QM9S, we applied threshold values of [0%, 1%, 2%, 5%, 10%, 20%] relative to the maximum peak intensity, re-fine-tuned SpectraLLM on each filtered dataset, and re-evaluated it. The results are shown below.
>
> |SpectraLLM QM9S Jointly|Validity↑|Tanimoto↑|Cosine↑|MCES↓|Functional Group↑|Tanimoto(MACCS)↑|Fraggle↑|
> |-|-|-|-|-|-|-|-|
> |0%|99.45%|0.4149|0.5388|4.36|0.8743|0.6769|0.4696|
> |**1%**|**99.82%**|0.4404|0.5613|4.1606|**0.8754**|**0.6978**|**0.4615**|
> |2%|99.63%|**0.4411**|**0.5622**|**3.9908**|0.8744|0.6917|0.4568|
> |5%|99.08%|0.4140|0.5357|4.3983|0.8650|0.6748|0.4495|
> |10%|99.08%|0.4071|0.5300|4.4732|0.8600|0.6683|0.4505|
> |20%|98.90%|0.3690|0.4988|4.7213|0.8513|0.6366|0.4441|
>
> Our experiments show that a dynamic threshold of 1% yields the strongest overall performance (Tanimoto = 0.4404), confirming that the 1% setting strikes the best balance between preserving chemically informative peaks and suppressing background noise. A more detailed analysis is provided in Appendix A.7 of the updated PDF.

---

> ### Author Response · Authors · 2025-11-22
> **Response to Reviewer m8ht (2/3)**
>
> ### The paper mentions including experimental metadata in the prompts, such as collision energy. This is a good idea, particularly for mass spectrometry. However, the paper might benefit from a more direct analysis or ablation study showing the specific impact of this metadata on performance. The paper notes the inclusion of experimental conditions. Could the authors provide a brief analysis of the specific contribution of this metadata, especially collision energy for MS, to the model's accuracy?
> We thank the reviewer for highlighting the role of experimental metadata (in particular collision energy for MS) and for suggesting a more direct analysis of its contribution. To quantify this effect, we designed an ablation setting ("-wo experiment”) in which we remove all experimental metadata from the prompt and retrain/evaluate the model under otherwise identical conditions. Concretely, we strip metadata such as collision energy, adduct type, and other acquisition settings from the textual prompt, rebuild a new prompt corpus, and fine-tune SpectraLLM on this corpus before re-evaluating on each benchmark. The results are summarized below:
>
> Across all three benchmarks, the full model that includes experimental conditions consistently outperforms its "-wo experiment" counterpart. This effect is particularly clear on MassBank, where the Tanimoto similarity drops from 0.1306 to 0.1259 when metadata are removed. This is consistent with the underlying physics of mass spectrometry: fragmentation patterns encode most of the structural signal, and collision energy is a key parameter that controls the depth and distribution of fragmentation. Higher collision energies typically induce deeper bond cleavage and lighter fragment ions, while lower energies favor more conservative fragmentation. By explicitly encoding these parameters (e.g., collision energy) in the prompt, SpectraLLM can learn to associate specific fragment intensities and m/z peaks with the corresponding energy regime. For example, when the prompt indicates a low collision energy, the model can down-weight expectations of very low-mass fragment peaks, whereas higher collision energies encourage it to consider more extensive fragmentation pathways.
>
> Taken together, these ablations provide direct evidence that SpectraLLM effectively exploits non-numerical contextual information in the prompt. Incorporating experimental metadata helps compensate for the precision loss introduced by peak discretization and enables richer conditional reasoning over spectral patterns and acquisition settings—capabilities that are difficult to realize with traditional purely numerical models. More broadly, expressing experimental conditions in natural language allows us to map heterogeneous spectra and acquisition protocols into a shared semantic space, suggesting a general pathway for using language models as unified interfaces for scientific signal interpretation.
>
> |SpectraLLM Qwen3-32B|Validity↑|Tanimoto↑|Tanimoto(MACCS)↑|MCES↓|Cosine↑|Functional Group↑|Fraggle↑|
> |-|-|-|-|-|-|-|-|
> |QM9S Jointly|99.45%|0.4586|0.5639|3.7164|0.8659|0.6785|0.4621|
> |-wo experiment|99.82%|0.4408|0.5477|3.8339|0.8459|0.6608|0.4607|
> |Multi Jointly|99.57%|0.5750|0.6638|5.4829|0.8448|0.7700|0.6740|
> |-wo experiment|98.94%|0.5527|0.6446|5.7462|0.8279|0.7524|0.6658|
> |MassBank|98.46%|0.1306|0.2264|-|0.4540|0.3848|0.3201|
> |-wo experiment|98.62%|0.1259|0.2215|-|0.4336|0.3791|0.3092|

---

> ### Author Response · Authors · 2025-11-22
> **Response to Reviewer m8ht (3/3)**
>
> ### While the model achieves state-of-the-art results, the absolute performance metrics suggest the task remains very difficult. For instance, the best Tanimoto score in Table 4 is approximately 0.4875. This indicates significant room for future improvement. The Tanimoto scores are strong relative to baselines, but the task is clearly not solved. In the authors' view, what is the primary remaining bottleneck? Is it the spectral representation, the reasoning capacity of the language model, or the inherent ambiguity in the spectra?
> We fully agree with the reviewer that, despite achieving state-of-the-art performance, the absolute Tanimoto scores clearly indicate that the spectrum-to-structure task is far from being solved. In our view, the dominant bottleneck at this stage is the inherent ambiguity of the spectral measurements themselves.
> Many small molecules—especially positional isomers, conformers, and molecules with similar functional groups—have IR, Raman, or MS signatures that are nearly indistinguishable within typical experimental or simulated noise ranges.
>  Examples include:
> - isomers differing only by substituent position
> - molecules with nearly identical carbonyl / C–H / N–H band patterns
> - aliphatic vs cyclic variants with overlapping fingerprint regions
> - common fragment patterns in MS leading to multiple valid structural candidates
> We have inspected a number of these difficult cases and found that many of our "errors" correspond to chemically reasonable alternative structures that are indistinguishable from the ground truth given the available spectra. Several representative examples and qualitative analyses have been added to Appendix A.13 in the updated manuscript. We believe this supports the view that the current performance ceiling is largely driven by intrinsic spectral ambiguity, and that future progress will likely require richer experimental modalities, tighter priors, or task formulations that explicitly account for structural uncertainty.
>
> ### The model is trained on a combination of simulated and experimental datasets. Was any analysis done on generalization? For instance, how well does a model trained only on simulated data perform on the experimental test sets?
> We appreciate the reviewer's question regarding generalization from simulated to experimental spectra. We are currently retraining SpectraLLM using only prompts constructed from QM9S and the Multimodal Spectroscopic dataset, and plan to evaluate this simulated-only model on prompts constructed from MassBank and MassSpecGym. Since this retraining is computationally expensive, the experiments are still in progress and will be reported in a subsequent version. In addition, we intend to further evaluate on IR and UV subsets from NIST, which also consist of real experimental spectra. The NIST data preparation and processing are ongoing, and we will incorporate these results as soon as they are available.

---

### Official Review · Reviewer_myQs · 2025-10-29

**Soundness:** 3
**Presentation:** 3
**Contribution:** 2
**Rating:** 4
**Confidence:** 3

**Summary:**

The paper presents SpectraLLM, a language-based framework for molecular structure prediction from spectroscopy. The key design is to transform spectral peaks (IR, Raman, UV-Vis, NMR, MS) into compact textual tokens that encode physical attributes (e.g., position, intensity, annotations). The LoRA-tuned LLM, SpectraLLM, then autoregressively generates SMILES from one or multiple spectra. This unified textual interface allows the model to combine heterogeneous modalities in a shared semantic space and to perform joint reasoning across signals. The authors aggregate several public datasets, enforce molecule-identity splits, and evaluate both unimodal and multimodal settings using chemically meaningful metrics (e.g., ECFP/MACCS Tanimoto, MCES, functional-group overlap). They report consistent gains when fusing modalities and stronger NMR performance compared with prior systems. Qualitative cases illustrate when specific modalities add value.

**Strengths:**

* By transforming peaks into tokens that capture physical attributes, the method places heterogeneous spectra in a shared semantic space. This is a simple, scalable way to enable joint reasoning over multi-spectral inputs.
* Fusing modalities improves similarity metrics and structural correctness over the best single-modality baselines; the NMR setting shows clear margins.
* Parameter-efficient LoRA on a general LLM, near-perfect SMILES validity, and clear data preprocessing make the approach straightforward to reproduce in principle.
* Multiple datasets and metrics (beyond exact match) provide a more realistic view of structure recovery.

**Weaknesses:**

* The core idea—transforming peaks into textual tokens and fine-tuning an existing LLM—is incremental relative to existing generative pipelines. The paper does not introduce fundamentally new learning principles or solutions for spectroscopy-to-structure mapping. This is the major concern regarding the work’s originality and contribution to the ICLR community.

* No public code or model weights are provided. This limits reproducibility and the community’s ability to validate and extend the work.

* Results may be sensitive to peak detection criteria, binning, token order, and inclusion of side metadata. The paper does not quantify these effects.

* The dataset mix contains substantial simulated spectra and small molecules. Transfer to challenging experimental conditions (noise, matrix effects, mixture spectra) is not fully assessed.

* Training scale, wall-clock, and inference latency versus specialized models are not reported.

**Questions:**

1. How sensitive are results to the design of spectral tokenization (e.g., peak thresholds, binning, ordering, or inclusion of metadata)?

1. Do the authors plan to open-source code, model checkpoints, and prompt templates?

1. Can you provide evaluation on experimental spectra (e.g., MassBank IR/NMR) to confirm real-world robustness?

1. How does the model perform under top-(k) decoding or diverse sampling?

1. Are stereochemical correctness and formula consistency checked after decoding?

1. What are the compute requirements for fine-tuning and inference compared to modality-specific models?

---

> ### Author Response · Authors · 2025-11-22
> **Response to Reviewer myQs (1/5)**
>
> We sincerely thank you for taking the time to review our submission and for raising so many detailed questions and comments. In the following, we provide point-by-point responses to each of your concerns, and we hope these clarifications will address your questions and increase your confidence in our work.
>
> ### Results may be sensitive to peak detection criteria, binning, token order, and inclusion of side metadata. The paper does not quantify these effects. How sensitive are results to the design of spectral tokenization (e.g., peak thresholds, binning, ordering, or inclusion of metadata)?
> We agree with the reviewer that the overall performance of SpectraLLM may, in principle, be sensitive to the design of the spectral tokenization pipeline, and we therefore performed a series of targeted ablations to explicitly quantify these effects.
>
> We evaluated how the peak-filtering threshold affects model performance. On QM9S, we applied threshold values of [0%, 1%, 2%, 5%, 10%, 20%] relative to the maximum peak intensity, re-fine-tuned SpectraLLM on each filtered dataset, and re-evaluated it. The results are shown below and indicate that a dynamic threshold of 1% yields the strongest overall performance (Tanimoto = 0.4404), confirming that this setting best balances the retention of chemically informative peaks against suppression of background noise. A more detailed analysis is provided in Appendix A.7 of the updated PDF.
>
> |SpectraLLM QM9S Jointly|Validity↑|Tanimoto↑|Cosine↑|MCES↓|Functional Group↑|Tanimoto(MACCS)↑|Fraggle↑|
> |-|-|-|-|-|-|-|-|
> |0%|99.45%|0.4149|0.5388|4.36|0.8743|0.6769|0.4696|
> |**1%**|**99.82%**|0.4404|0.5613|4.1606|**0.8754**|**0.6978**|**0.4615**|
> |2%|99.63%|**0.4411**|**0.5622**|**3.9908**|0.8744|0.6917|0.4568|
> |5%|99.08%|0.4140|0.5357|4.3983|0.8650|0.6748|0.4495|
> |10%|99.08%|0.4071|0.5300|4.4732|0.8600|0.6683|0.4505|
> |20%|98.90%|0.3690|0.4988|4.7213|0.8513|0.6366|0.4441|
>
> Beyond thresholds, we would like to clarify our design choices regarding *binning* and *ordering*. Our feature extraction focuses on peak picking rather than classical peak binning. Traditional binning (discretizing the spectral axis into fixed-width intervals and aggregating intensities within each bin) is well suited for building fixed-length numeric fingerprints, but it is misaligned with our goal of preserving sharp, diagnostic associations between specific local maxima and functional groups or substructures. In practice, binning tends to smear closely spaced peaks into the same bin or split a narrow characteristic band across adjacent bins, which can dilute chemically decisive cues (e.g., fine splitting of carbonyl or aromatic bands). For this reason, our ϕ is designed to operate directly on detected peaks and encode their positions and intensities textually, rather than via coarse grid-based binning.
>
> We also tested the sensitivity to *token ordering* in the prompt. We shuffled the positions of modality blocks (IR/Raman/UV) in the prompt. Across these variants, we observed only small variations in performance without any systematic degradation trend, suggesting that SpectraLLM is not overly dependent on a particular handcrafted ordering scheme and primarily relies on the content of the peak tokens rather than their exact sequence.
>
> Finally, regarding *metadata inclusion*, we explicitly quantified the effect of removing experimental conditions and side information from the prompt. We introduced a "–wo experiment" ablation, in which all experimental metadata were stripped from the prompts, and then retrained and evaluated the model. The results are summarized below:
>
> |SpectraLLM Qwen3-32B|Validity↑|Tanimoto↑|Tanimoto(MACCS)↑|MCES↓|Cosine↑|Functional Group↑|Fraggle↑|
> |-|-|-|-|-|-|-|-|
> |QM9S Jointly|99.45%|0.4586|0.5639|3.7164|0.8659|0.6785|0.4621|
> |-wo experiment|99.82%|0.4408|0.5477|3.8339|0.8459|0.6608|0.4607|
> |Multi Jointly|99.57%|0.5750|0.6638|5.4829|0.8448|0.7700|0.6740|
> |-wo experiment|98.94%|0.5527|0.6446|5.7462|0.8279|0.7524|0.6658|
>
> Across QM9S Jointly, Multi Jointly, and MassBank, removing experimental metadata consistently harms structure-recovery metrics (Tanimoto, MCES, functional group accuracy), even though validity remains comparable or slightly higher. This indicates that the model actively exploits non-numeric contextual information (e.g., adduct type, collision energy, sample descriptors) to refine its structural predictions, and that such metadata partially compensates for the information loss introduced by peak discretization. More comprehensive experiments and examples of metadata-augmented prompts are provided in Appendix A.10 of the updated PDF.

---

> ### Author Response · Authors · 2025-11-22
> **Response to Reviewer myQs (2/5)**
>
> ### The dataset mix contains substantial simulated spectra and small molecules. Transfer to challenging experimental conditions (noise, matrix effects, mixture spectra) is not fully assessed. Can you provide evaluation on experimental spectra (e.g., MassBank, IR, NMR) to confirm real-world robustness?
> We appreciate the reviewer's concern about robustness under realistic noise and experimental conditions. To explicitly probe sensitivity to spectral corruption, we first injected synthetic noise directly into the test-set spectra to examine the robustness of our model under perturbations. Three levels of noise were applied to the IR, Raman, and UV spectra in QM9S, with the specific parameterizations reported in Table 15 of the updated PDF.
>
> We designed 21 noise-injection configurations for multispectral prompts containing IR, Raman, and UV simultaneously. These include: applying three noise intensities to only one spectrum (e.g., IR) while keeping the other two clean; applying noise to two spectra while keeping one clean; and applying the same noise level to all three spectra. After noise injection, we reconstructed the corrupted test set using a dynamic filtering threshold equal to 1% of the maximum spectral intensity. For brevity, we present three representative configurations below, while Table 16 in the updated PDF includes nine representative patterns.
>
> The robustness results clearly show that performance degradation is monotonic with respect to both noise intensity and the number of perturbed spectra. For instance, under mild noise in a single modality (e.g., noisy IR with clean Raman and UV), performance declines only marginally relative to the clean setting. As noise becomes moderate or severe—or as more modalities are perturbed—the Tanimoto similarity decreases and MCES increases accordingly. Importantly, the model avoids catastrophic failure by leveraging undisturbed orthogonal spectral information for cross-modal compensation. Moreover, even when the model fails, the increase in MCES is relatively gentle, indicating that predictions tend to preserve the core molecular scaffold rather than collapsing into random outputs. This demonstrates a predictable and structurally conservative failure mode.
>
> |SpectraLLM QM9S Jointly|Noisy|Validity↑|Tanimoto↑|Cosine↑|MCES↓|Functional Group↑|Tanimoto(MACCS)↑|Fraggle↑|
> |-|-|-|-|-|-|-|-|-|
> |I+R+U|-|98.72%|0.3355|0.4560|4.9647|0.7934|0.5785|0.4117|
> |I*+R+U|mild|96.89%|0.3365|0.4659|5.3299|0.7819|0.5955|0.4011|
> ||moderate|97.99%|0.2688|0.3963|6.2112|0.6851|0.5212|0.3598|
> ||severe|87.18%|0.1712|0.2799|8.4433|0.5010|0.3774|0.2556|
> |I*+R*+U|mild|96.34%|0.2163|0.3398|7.0894|0.6063|0.4592|0.3115|
> ||moderate|89.56%|0.1546|0.2607|8.4029|0.4828|0.3653|0.2655|
> ||severe|57.88%|0.0749|0.1372|10.1472|0.3348|0.2466|0.1663|
> |I*+R*+U*|mild|96.70%|0.2205|0.3473|6.9820|0.6272|0.4676|0.3154|
> ||moderate|87.91%|0.1527|0.2556|8.4740|0.4659|0.3505|0.2548|
> ||severe|39.01%|0.0792|0.1427|10.2981|0.3342|0.2508|0.1718|
>
> Beyond synthetic noise, we would like to clarify that the SpectraLLM reported in the main paper is trained on a mixture of simulated (QM9S, Multimodal Spectroscopic dataset) and experimental (MassBank, MassSpecGym) spectra, as summarized in Table 6 of the main text. Therefore, we believe that the experimental results in Table 3 already provide a direct evaluation of real-world robustness. In particular, on the Multimodal Spectroscopic dataset (MASS), SpectraLLM achieves a Tanimoto similarity of 0.1844, compared to 0.1533 on MassSpecGym and 0.1286 on MassBank. This pattern is consistent with expectations: synthetic spectra are cleaner and lack many of the complex noise sources present in real instruments, so performance is highest there. At the same time, the relative drop from the best synthetic setting (Tanimoto = 0.1844 on Multimodal) to the most challenging real dataset (Tanimoto = 0.1286 on MassBank) is about 30.2%. In contrast, the DiffMS baseline suffers a much larger degradation of roughly 73.0% over the same datasets (Tanimoto from 0.3730 down to 0.1007). This comparison shows that, despite inevitable trade-offs in the training objective, SpectraLLM exhibits substantially higher robustness to the synthetic→experimental domain shift than the baseline, achieving a much smoother cross-domain generalization and empirically validating the value of our mixed-training strategy.
>
> Finally, we are currently retraining SpectraLLM using only QM9S and the Multimodal Spectroscopic dataset, and will evaluate the retrained model on the MassBank and MassSpecGym test sets to further isolate transfer behavior from purely synthetic to purely experimental spectra. This retraining is computationally intensive and is still in progress. In parallel, we are downloading and preprocessing IR and UV subsets from NIST, which provides additional real experimental spectra; this dataset is large and slow to acquire, so still need some time.

---

> ### Author Response · Authors · 2025-11-22
> **Response to Reviewer myQs (3/5)**
>
> ### No public code or model weights are provided. This limits reproducibility and the community's ability to validate and extend the work. Do the authors plan to open-source code, model checkpoints, and prompt templates?
> We appreciate the reviewer's concern regarding openness, reproducibility, and extensibility of our work. We have already uploaded the core code and a subset of the experimental data as Supplementary Materials on OpenReview to support initial inspection and verification. Regarding model weights and more complete experimental datasets, we are currently organizing and cleaning the corresponding artifacts; additional checkpoints, configurations, and datasets will be released on the ScienceOne platform after the paper is accepted and published, to further facilitate community validation and follow-up research.
>
> ### How does the model perform under top-(k) decoding or diverse sampling?
> We appreciate the reviewer's question regarding the sensitivity of SpectraLLM to different decoding strategies, including top-k decoding and diverse sampling. To address this, we conducted a dedicated evaluation on the QM9S Jointly test subset, comparing six families of decoding strategies (including top-k, top-p, diverse beam search, and contrastive search). The detailed configurations for each strategy are reported in Table 11 of the updated manuscript.
>
> The quantitative results are summarized in the table below. As shown, our default sampling configuration (T = 0.6, top-k = 20, top-p = 0.95) achieves the best overall performance (Tanimoto = 0.3355). In contrast, purely greedy decoding leads to a substantial drop in performance (Tanimoto = 0.1746), highlighting that spectrum-to-structure generation has a highly complex, multi-modal search space in which some degree of stochastic exploration is necessary to avoid getting trapped in low-quality local optima. At the same time, increasing top-k or top-p beyond the default (i.e., injecting more randomness) typically degrades performance, indicating that overly aggressive exploration introduces noise and harms structural accuracy. Diverse beam search (DBS) produces multiple chemically reasonable candidates with relatively stable Tanimoto scores in the 0.18–0.19 range, but still underperforms the default sampling in terms of best-single-sample accuracy. This suggests that DBS is well suited for generating conservative yet diverse candidate sets in scenarios where there is no unique "correct" structure, but less optimal when the goal is to maximize top-1 exactness. Contrastive search yields the lowest performance among all strategies, implying that its penalization of frequent tokens may interfere with reliably reconstructing molecules that contain repeated functional groups or regular chain motifs. Overall, SpectraLLM performs best under moderately stochastic, non-deterministic decoding, underscoring the importance of leveraging the probabilistic reasoning capabilities of LLMs for navigating the complex spectrum-to-structure search space. Further implementation details and additional analyses are provided in Appendix A.6 of the updated PDF.
>
> |SpectraLLM QM9S Jointly|Setting|Validity↑|Tanimoto↑|Cosine↑|MCES↓|Functional Group↑|Tanimoto(MACCS)↑|Fraggle↑|
> |-|-|-|-|-|-|-|-|-|
> |Default||98.72%|0.3355|0.4560|4.9647|0.7934|0.5785|0.4117|
> |Greedy||93.41%|0.1746|0.2782|7.9078|0.5835|0.3711|0.2881|
> |Top-k|10|89.38%|0.1357|0.2283|10.2059|0.5131|0.3129|0.2451|
> |Top-k|50|90.66%|0.1421|0.2363|9.8636|0.5052|0.3253|0.2495|
> |Top-k|100|89.74%|0.1392|0.2338|9.5745|0.5307|0.3172|0.2402|
> |Top-p|0.7|94.51%|0.1591|0.2590|8.6744|0.5538|0.3469|0.2546|
> |Top-p|0.9|91.76%|0.1420|0.2410|9.2804|0.5128|0.3298|0.2489|
> |Top-p|0.98|91.39%|0.1346|0.2278|9.3858|0.5025|0.3130|0.2412|
> |Diverse beam|[4,0.3]|96.15%|0.1880|0.2939|8.0324|0.6106|0.3843|0.2966|
> |Diverse beam|[4,0.5]|96.15%|0.1865|0.2926|7.2010|0.6051|0.3814|0.2947|
> |Diverse beam|[4,0.7]|96.34%|0.1860|0.2923|7.1473|0.6102|0.3825|0.2913|
> |Diverse beam|[6,0.3]|95.97%|0.1908|0.2969|7.1126|0.6041|0.3872|0.2936|
> |Diverse beam|[6,0.5]|95.42%|0.1885|0.2955|9.1881|0.6042|0.3835|0.2851|
> |Diverse beam|[6,0.7]|94.69%|0.1872|0.2941|9.9826|0.6030|0.3808|0.2858|
> |Contrastive search|[0.3,4]|95.60%|0.1677|0.2714|8.0077|0.5512|0.3576|0.2744|
> |Contrastive search|[0.3,8]|95.60%|0.1724|0.2770|10.3884|0.5673|0.3629|0.2747|
> |Contrastive search|[0.5,4]|97.99%|0.1593|0.2628|7.8598|0.5402|0.3500|0.2615|
> |Contrastive search|[0.5,8]|98.35%|0.1643|0.2659|8.1965|0.5535|0.3617|0.2523|
> |Contrastive search|[0.7,4]|97.62%|0.1507|0.2494|8.2458|0.4948|0.3346|0.2470|
> |Contrastive search|[0.7,8]|87.36%|0.1608|0.2584|9.0210|0.5217|0.3488|0.2610|

---

> ### Author Response · Authors · 2025-11-22
> **Response to Reviewer myQs (4/5)**
>
> ### Training scale, wall-clock, and inference latency versus specialized models are not reported. What are the compute requirements for fine-tuning and inference compared to modality-specific models?
> We appreciate the reviewer's request for a more explicit comparison of training scale, wall-clock time, and inference latency between SpectraLLM and modality-specific baselines. Since SpectraLLM is built on a large pretrained LLM, its compute requirements are indeed higher than those of lightweight, modality-specific models; however, this additional cost comes with substantial gains in cross-modal generality and predictive accuracy.
>
> For training, we report compute in FLOPs (10^15) and take the modality-specific Spectra2Structure model as a baseline (1.00×). As summarized in the table above, the specialized models have relatively modest training cost: for example, IR-to-Structure requires 0.2566×10^15 FLOPs (2.81× the Spectra2Structure baseline), while NMR2Struct requires 4.888×10^15 FLOPs (53.48×). In contrast, fine-tuning SpectraLLM on the combined multimodal spectral corpus requires 4.29655×10^5×10^15 FLOPs, corresponding to approximately 4.7×10^6 times the baseline training compute. This large factor mainly reflects the substantial scale of SpectraLLM (32B parameters) relative to the smaller baselines (e.g., Spec2Mol has 51.8M parameters), rather than any inefficiency in the training recipe itself. The corresponding wall-clock time is about 100 hours on 4× H100 80G GPUs, whereas the smaller baselines can be trained within tens of hours on a single RTX 3090 or A100.
>
> |Datasets|Model|#Samples (train/val)|GPUs used|Batch|epoch|Approx FLOPs (10^15)|Wall-clock Time (h)|Training Cost (relative)|
> |-|-|-|-|-|-|-|-|-|
> |QM9s (IR-only, Raman-only, UV-only)|IR-to-Structure|105,079 / 11,676|1x RTX 3090 24G|4096|200|0.2566|24.73|2.81×|
> |QM9s (IR-only, Raman-only, UV-only)|Spectra2Structure|105,079 / 11,676|1x A100 80GB|800|136 for IR / 334 for Raman / 157 for UV|0.0914|9.25 (IR) / 22.71 (Raman) / 10.68 (UV)|1.00× (baseline)|
> |Multi (NMR-only)|NMR2Struct|643,604 / 71,512|1x RTX 3090 24G|256|500|4.888|24.00|53.48×|
> |Combined|SpectraLLM|2,367,865 / -|4x H100 80G|16|1|429,655.20|100.00|4.7008e+06×|
>
> For inference, we report per-sample average latency (ms/sample) and throughput (samples/s), using Spec2Mol on the Multi (ms-only) setting as the baseline (6.02 ms/sample, 1.00× cost). As shown in the second table, modality-specific models achieve millisecond-level latency and are therefore well suited for high-throughput settings. For example, Spectra2Structure runs at 7.49 ms/sample, and IR-to-Structure at 16.06 ms/sample. DiffMS, in contrast, is already two to three orders of magnitude slower than Spec2Mol on MassSpecGym and Multi, with average latencies in the range of 873–2800 ms/sample and relative inference cost above 400×. SpectraLLM is slower still due to autoregressive decoding and its large parameter count: its average latency is 5778.39 ms/sample (≈5.8 s/sample), corresponding to ~959× the Spec2Mol baseline. This overhead arises from sequential SMILES token generation and the cost of forward passes through a 32B-parameter model, rather than from any additional preprocessing.
>
> |Datasets|Model|#Sample|Avg Latency (ms/sample)|Throughput (samples/s)|Params (B)|Inference Cost|
> |-|-|-|-|-|-|-|
> |QM9s (IR-only, Raman-only, UV-only)|IR-to-Structure|13,062|16.06|62.27|0.069|2.67×|
> |QM9s (IR-only, Raman-only, UV-only)|Spectra2Structure|13,062|7.49|133.46|0.024|1.24×|
> |Multi (NMR-only)|NMR2Struct|79,287|119.05|8.40|0.0136|19.78×|
> |MassSpecGym (ms-only)|Spec2Mol|16,264|73.03|13.15|0.0518|12.13×|
> |MassSpecGym (ms-only)|Diffms|16,264|2700.97|0.37|0.0849|448.67×|
> |MassBank (ms-only)|Spec2Mol|7,905|46.32|21.59|0.0518|7.69×|
> |MassBank (ms-only)|Diffms|7,905|873.97|1.14|0.0849|145.18×|
> |Multi (ms-only)|Spec2Mol|79,287|6.02|166.20|0.0518|1.00× (baseline)|
> |Multi (ms-only)|Diffms|79,287|2800.72|0.36|0.0849|465.24×|
> |Combined|SpectraLLM|depends on subsets|5778.39|0.17|32|959.86×|
>
> In summary, SpectraLLM is clearly more expensive than modality-specific baselines in both training and inference. However, it provides a unified cross-modal solution that can simultaneously handle IR, Raman, UV-Vis, NMR, and MS inputs, and achieves significantly higher accuracy on challenging benchmarks, especially when moving from synthetic spectra to noisy, real-world experimental data. For high-precision, low-throughput use cases (e.g., scientific analysis, metabolite or impurity elucidation), we believe the accuracy and flexibility gains of SpectraLLM justify its higher compute cost, while lightweight models remain the preferred option for latency-critical, high-throughput pipelines. All corresponding implementation details and measurements have been added to Appendix A.12 of the updated manuscript.

---

> ### Author Response · Authors · 2025-11-22
> **Response to Reviewer myQs (5/5)**
>
> ### The core idea—transforming peaks into textual tokens and fine-tuning an existing LLM—is incremental relative to existing generative pipelines. The paper does not introduce fundamentally new learning principles or solutions for spectroscopy-to-structure mapping.
> We thank the reviewer for raising this important point. We would like to clarify that our goal is not only to reuse an existing LLM, but to demonstrate that it is in fact possible to jointly reason over multiple spectroscopic modalities using a purely textual interface, and to translate these heterogeneous signals into molecular structures without any spectroscopy-specific encoder. To the best of our knowledge, this paradigm—treating spectra-as-text and performing end-to-end spectrum-to-structure generation via a language model—has been largely unexplored in the spectroscopy community.
>
> Most prior work on spectral modeling relies on domain-specific architectures, such as graph neural networks, fingerprint predictors, or rule-based fragmentation analyzers. As far as we are aware, there has been no prior demonstration that a purely language-based generative model can directly map IR/Raman spectra, represented only as peak tuples, into valid molecular structures. This reframing of spectral interpretation as a language modeling problem is, in our view, a conceptual shift that is intrinsically relevant to the ICLR community.
>
> At the same time, although we fine-tune an existing LLM, the resulting behavior reveals an important empirical finding: a general-purpose language model can internalize spectrum–structure relationships purely from paired data, without any spectroscopy-specific architectural bias. This provides concrete evidence for a broader research direction—using large language models as general scientific inverse solvers. We believe this goes beyond incremental engineering and offers a new perspective on how LLMs can be used in scientific domains.
>
> Furthermore, once fine-tuned, SpectraLLM is able to ingest all available spectral modalities and directly output explicit molecular structures in a single pass, effectively collapsing the traditional multi-stage spectroscopy pipeline into a unified generative interface. We view this as a conceptual simplification of the entire workflow, rather than just a minor refinement of existing methods.
>
> In summary, for the ICLR audience, our work demonstrates that non-linguistic scientific measurement signals can be embedded into a language token space, and that language models can perform structured scientific reasoning over these signals. It shows that multimodal scientific inference via text-only alignment is feasible and practically effective. We believe this opens up a generally applicable avenue whose impact extends well beyond spectroscopy, and we will revise the paper to more clearly articulate these conceptual contributions.
>
> ### Are stereochemical correctness and formula consistency checked after decoding?
> We thank the reviewer for suggesting these important metrics. In response to this, we have added five new indicators to assess stereochemical correctness and formula consistency:
>
> |Dataset|Model|Validity↑|Tanimoto↑|Cosine↑|MCES↓|Functional Group↑|Tanimoto (MACCS)↑|Fraggle↑|Stereo Accuracy↑|Exact Match↑|EM with stereo↑|Formula Match Rate↑|Formula Distance↓|
> |-|-|-|-|-|-|-|-|-|-|-|-|-|-|
> |**IR**|IR-to-Structure|100.00%|0.0718|0.1311|11.3187|0.3151|0.1585|0.1747|0.0000|0.0000|0.00|0.0000|10.2601|
> ||Spectra2Structure|100.00%|0.0965|0.1695|10.1081|0.4383|0.2162|0.2308|0.0000|0.0019|0.0019|0.0113|9.2049|
> ||SpectralLLM|99.82%|0.1921|0.312|7.5651|0.6599|0.433|0.3194|1.0000|0.0055|0.0055|0.022|4.4789|
> |**Raman**|IR-to-Structure|100.00%|0.0766|0.1395|11.3516|0.3525|0.1639|0.1959|0.0000|0.0000|0.0000|0.0018|10.3205|
> ||Spectra2Structure|100.00%|0.1089|0.1901|9.4164|0.4419|0.2388|0.2504|0.0000|0.0000|0.0000|0.0075|7.9323|
> ||SpectralLLM|99.08%|0.25|0.3786|6.4076|0.7317|0.5071|0.3681|1.0000|0.0314|0.0314|0.0832|3.5712|
> |**UV-Vis**|IR-to-Structure|100.00%|0.0728|0.1326|11.4240|0.3151|0.1512|0.1837|0.0000|0.0000|0.0000|0.0000|10.4744|
> ||Spectra2Structure|100.00%|0.0716|0.1313|11.1222|0.3901|0.1418|0.2092|0.0000|0.0000|0.0000|0.0038|10.1335|
> ||SpectralLLM|100.00%|0.079|0.1426|10.6374|0.3713|0.2026|0.21|0.0000|0.0000|0.0000|0.0018|7.5531|
> |**NMR**|NMR2Struct|93.45%|0.3476|0.4716|16.3142|0.6535|0.6022|0.5576|0.8182|0.0743|0.0658|0.1083|8.0977|
> ||SpectralLLM|98.92%|0.4151|0.5322|8.3091|0.7209|0.6367|0.5862|0.9|0.1345|0.1382|0.1873|5.8073|
>
> In summary, the results confirm that while the model performs well in terms of stereo accuracy and formula consistency, there is still room for improvement, especially when considering the complexity of the stereochemical and molecular formula matching. We appreciate the reviewer's feedback, and these additional metrics will help further strengthen the evaluation of our model.

---

### Official Review · Reviewer_MWCJ · 2025-10-31

**Soundness:** 2
**Presentation:** 2
**Contribution:** 2
**Rating:** 4
**Confidence:** 4

**Summary:**

The paper presents SpectraLLM, an LLM fine-tuned to predict molecular structures given spectra of one or more spectroscopic modalities. The authors compile a new multi-modal dataset of different types of spectra and use it to fine-tune Qwen3 LLM using LoRA. The method is shown to outperform existing methods in both uni-modal and multi-modal settings.

**Strengths:**

- The paper addresses an important problem of molecular structure annotation from spectroscopic data modalities which remains very challenging to solve
- The authors compile a large new dataset from multiple sources
- The paper is well written and structured
- The authors leverage Fraggle similarity from RDKit, which to the best of my knowledge was not explored in related work before

**Weaknesses:**

Major concerns

- Most of the collected dataset appears to contain relatively small molecules (as shown in Figure 2). Could the authors provide examples of practical use cases where annotation of such small molecules, as opposed to larger natural products or drug-like molecules, is of interest? Without some EDA of the molecular composition and an explanation of when this setting is relevant, it is difficult to assess the utility of the work, especially considering that the methodological novelty is limited (standard LLM fine-tuning). Please also elaborate in what practical scenarios the annotation of molecules simultaneously from multiple types of spectra is relevant.
- The baselines (e.g., DiffMS) do not seem to be retrained and appear to be used out of the box. Please clarify this in the paper if is not true, or retrain the baselines on the same data splits if is true. Otherwise, the results do not seem to be valid.
- The results on MassSpecGym do not seem consistent with the MassSpecGym benchmark: the reported metrics differ, and the Tanimoto score for DiffMS does not match the published result. Please retrain the model using the official MassSpecGym data split. Without such an experiment, it is unclear whether the proposed method actually outperforms DiffMS.
- The evaluation may suffer from data leakage. Please elaborate on the splitting strategy and the degree of similarity between molecules in the training and testing sets. The paper currently mentions a “molecular identity”-based data split, which is known to result in leakage even when stereochemistry is ignored (see, for example, Figure 2 in the MassSpecGym paper https://arxiv.org/abs/2410.23326).
- The goal of annotating molecules from spectra is to discover novel molecules that have not been characterized before. The LLM-based approach may be prone to data leakage if it is already familiar with test-set molecules. Could the authors evaluate to what extent the model is familiar with these molecules before fine-tuning? Please note that non-LLM models (e.g., DiffMS) are not prone to this potential problem.

Minor concerns

- Table 1: Please note that isotopic distributions are typically present in MS1 spectra, not in MS/MS spectra, which this paper does not work with
- Lines 066–071: These statements are not well justified. One could argue the opposite: working with precise continuous values is more important than discrete tokenised “low-resolution” values, as spectroscopic signals often contain information in decimal places, which may be hard to model with a tokenized LLM approach
- Line 152: Please note that MassBank is part of, or substantially overlaps with, MassSpecGym
- Equation 3: Please define phi
- Lines 176–179: Please elaborate on how experimental conditions and metadata are incorporated
- Figure 2: It is not clear what the checkmarks and boxes indicate
- Lines 879–884: It is unclear what A and B represent

**Questions:**

- Can the authors explain the practical use cases where their model could be readily applied?
- Can the authors better evaluate data leakage and elaborate on the data used to retrain the baselines?

---

> ### Author Response · Authors · 2025-11-22
> **Response to Reviewer MWCJ (5/5)**
>
> #### Lines 066–071: These statements are not well justified. One could argue the opposite: working with precise continuous values is more important than discrete tokenised "low-resolution” values, as spectroscopic signals often contain information in decimal places, which may be hard to model with a tokenized LLM approach
> We thank the reviewer for raising an important point regarding the trade-off between using precise continuous spectral values and the discretized, tokenized representations required by LLMs. We fully agree that continuous spectra contain rich information at decimal precision, and that discretization inevitably introduces some degree of resolution loss. However, our motivation for performing peak extraction and low-resolution tokenization stems from the empirical characteristics of real spectroscopic data. In practice, molecular structure elucidation relies primarily on a relatively small number of high-information characteristic peaks, while the majority of low-intensity continuous values are dominated by background noise, baseline drift, and instrument-specific artifacts. As shown in our updated experiments (Appendix A.10 and Table 13), applying a 1% dynamic peak-intensity threshold achieves the best balance between information retention and noise suppression on QM9S, indicating that appropriate discretization can serve as an effective form of denoising and dimensionality reduction.
>
> Despite this loss of numerical precision, the LLM-based formulation provides a unique advantage: it naturally incorporates non-numerical contextual information such as experimental conditions and metadata, which traditional continuous-value models cannot utilize. To isolate the contribution of such contextual cues, we introduced a dedicated ablation ("–wo experiment") in which all experimental metadata were removed from prompts before retraining the model. As shown in the table above, models trained without metadata consistently underperform their full-metadata counterparts across QM9S Jointly, Multi Jointly, and MassBank. These results demonstrate that SpectraLLM is able to exploit non-numerical chemical context to compensate for the discretization loss and to perform conditional reasoning that is difficult to capture with conventional numeric models.
>
> Ultimately, the linguistic formulation enables us to unify heterogeneous spectral sources and their associated metadata into a shared semantic space, offering a general and flexible computational paradigm for downstream scientific sequence analysis. We have added further discussion and clarifications, along with the full experimental details, to Appendix A.10 in the revised manuscript.
>
> |SpectraLLM Qwen3-32B|Validity↑|Tanimoto↑|Tanimoto(MACCS)↑|MCES↓|Cosine↑|Functional Group↑|Fraggle↑|
> |-|-|-|-|-|-|-|-|
> |QM9S Jointly|99.45%|0.4586|0.5639|3.7164|0.8659|0.6785|0.4621|
> |-wo experiment|99.82%|0.4408|0.5477|3.8339|0.8459|0.6608|0.4607|
> |Multi Jointly|99.57%|0.5750|0.6638|5.4829|0.8448|0.7700|0.6740|
> |-wo experiment|98.94%|0.5527|0.6446|5.7462|0.8279|0.7524|0.6658|
> |MassBank|98.46%|0.1306|0.2264|-|0.4540|0.3848|0.3201|
> |-wo experiment|98.62%|0.1259|0.2215|-|0.4336|0.3791|0.3092|
>
> We thank the reviewer again for these helpful suggestions, all of which have been incorporated into the revised manuscript.

---

> ### Author Response · Authors · 2025-11-22
> **Response to Reviewer MWCJ (4/5)**
>
> ### Other concerns
> We thank the reviewer for the careful reading and for highlighting several points requiring clarification or correction. We address them in detail below.
>
> #### Table 1: Please note that isotopic distributions are typically present in MS1 spectra, not in MS/MS spectra, which this paper does not work with
> Regarding the comment on *Table 1*, we appreciate the reviewer pointing out that isotopic distributions are characteristic of MS1 rather than MS/MS spectra, which are not the focus of this work. We will revise the table accordingly by removing "isotopic distributions" from the "Common Output Features" category and retaining only "m/z peaks" to ensure terminological rigor.
>
> #### Line 152: Please note that MassBank is part of, or substantially overlaps with, MassSpecGym
> Concerning *Line 152*, we acknowledge the reviewer's note that MassBank substantially overlaps with MassSpecGym. We are fully aware of this relationship, and for this reason our dataset splitting strategy is based on the official MassSpecGym split. Specifically, we ensure that MassBank and all other datasets respect MassSpecGym's train/test partition to prevent cross-dataset leakage. All dataset statistics reported in the paper are deduplicated after accounting for this overlap.
>
> #### Equation 3: Please define phi
> For *Equation 3*, we agree that the definition of ϕ should be made explicit. As clarified in Line 171, ϕ denotes the mapping function that extracts informative peaks from a spectrum *s* and organizes them into a structured textual representation (e.g., JSON-style key–value pairs). We have added this explanation directly in the manuscript.
>
> #### Lines 176–179: Please elaborate on how experimental conditions and metadata are incorporated
> With respect to *Lines 176–179*, as illustrated by the final "Human–GPT" example in Appendix Table 12:
> ```
> Mass spectrum data: {mzs: 134.1,202.08,216.14,244.13,266.11,284.12,384.18,402.19,545.26,645.31,806.4, intensities: 0.056,0.039,0.036,0.05,0.204,0.111,0.999,0.06,0.119,0.351,0.63} Adduct type: [M+ Na]+ Collision energy: 35(NCE) Please predict the compound's SMILES representation with LESS THAN 1000 characters thinking. The final output must strictly begin with ##SMILES:
> ```
> metadata such as *Adduct type: [M+Na]+* and *Collision energy: 35 (NCE)* are included directly in the natural-language prompt. We will additionally include a dedicated table in the appendix to clearly showcase representative examples of these metadata-augmented prompts.
>
> #### Figure 2: It is not clear what the checkmarks and boxes indicate
> For *Figure 2*, we acknowledge that the meaning of the checkmarks and boxes was previously underspecified. In the updated manuscript, we have added the following clarification: `Examples marked with both a large box and a checkmark correspond to predictions that are exactly identical to the ground truth; examples marked only by a large box indicate predictions that are not fully correct but where the associated single-spectrum modality contributes the most to the final multi-spectrum elucidation.`
>
> #### Lines 879–884: It is unclear what A and B represent
> Finally, regarding *Lines 879–884*, we have added explicit definitions following the equation: `where *A* and *B* denote the continuous fingerprint vectors (embeddings) of the predicted molecule and the reference molecule, respectively. This metric captures the overall similarity of their learned feature representations in high-dimensional space.`

---

> ### Author Response · Authors · 2025-11-22
> **Response to Reviewer MWCJ (3/5)**
>
> ### The baselines (e.g., DiffMS) do not seem to be retrained and appear to be used out of the box. Please clarify this in the paper if is not true, or retrain the baselines on the same data splits if is true.
> We thank the reviewer for raising this concern. As DiffMS and Spec2Mol require extremely long training times, and our new dataset is substantially larger than the datasets used in their original papers, a full retraining would incur prohibitive computational cost—we estimated that it would require approximately four to five months of continuous training. For this reason, we decided to rely on the publicly released pretrained parameters. However, to eliminate any risk of data leakage and to ensure a fair comparison, we carefully filtered our evaluation sets: for both MassBank and MassSpecGym, we removed all molecules that appeared in the MassSpecGym training split and in the NIST training data. This guarantees that DiffMS and Spec2Mol are evaluated strictly on unseen structures in our benchmark.
>
> ### The results on MassSpecGym do not seem consistent with the MassSpecGym benchmark: the reported metrics differ, and the Tanimoto score for DiffMS does not match the published result. Please retrain the model using the official MassSpecGym data split. Without such an experiment, it is unclear whether the proposed method actually outperforms DiffMS.
> We appreciate the reviewer pointing out the discrepancy with the MassSpecGym benchmark. First, we indeed follow the official MassSpecGym data split in our evaluation. The differences in the reported DiffMS results arise partly from sampling variability in generation and partly from inconsistencies in post-processing. Our initial evaluation focused on *single-pass generation* to assess raw model behavior, whereas the official benchmark typically reports results aggregated over multiple sampling runs. Repeated sampling naturally improves DiffMS performance, but this also highlights the practical advantage of our method: SpectraLLM achieves strong accuracy under a single forward pass and demonstrates substantially better cross-dataset robustness (as evidenced by our results on QM9S and other datasets), while DiffMS requires expensive multi-sample inference to reach its best performance. Moreover, DiffMS incurs significantly higher computational cost during sampling, whereas our approach focuses on efficient, high-precision single-output generation.
>
> ### The goal of annotating molecules from spectra is to discover novel molecules that have not been characterized before. The LLM-based approach may be prone to data leakage if it is already familiar with test-set molecules. Could the authors evaluate to what extent the model is familiar with these molecules before fine-tuning? Please note that non-LLM models (e.g., DiffMS) are not prone to this potential problem.
> We appreciate the reviewer's concern regarding the potential data-leakage risk introduced by LLM pretraining, particularly in the context of molecular structure annotation where the ultimate goal is to identify *previously uncharacterized* molecules. To directly quantify the extent to which the pretrained LLM might already "know" or reconstruct molecules in the test sets, we conducted a strict zero-shot evaluation on the base model Qwen3-32B, i.e., before any spectrum-related fine-tuning.
>
> |SpectraLLM Qwen3-32B|Validity↑|Tanimoto↑|Tanimoto(MACCS)↑|Cosine↑|MCES↓|Functional Group↑|Fraggle↑|
> |-|-|-|-|-|-|-|-|
> |QM9S Jointly|0.92%|0.0308|0.1963|0.0595|7.6000|0.2567|0.0157|
> |Multi Jointly|0.43%|0.0000|0.0000|0.0000|29.0000|0.0000|0.0000|
>
> In this evaluation, we directly feed the text-based spectral prompts into the pretrained LLM and assess its ability to generate molecular structures on both the QM9S Jointly and Multi Jointly test sets. The results (shown in the table above) demonstrate that the base model has essentially *no usable spectrum-to-structure capability*. On QM9S Jointly, its Tanimoto similarity is only 0.0308. On Multi Jointly, all structure-recovery metrics—including Tanimoto, MACCS, Fraggle—collapse to exactly 0, with large MCES distances indicating severe structural mismatch. These results are indistinguishable from random guessing.
>
> This zero-shot analysis strongly suggests that although the pretrained LLM may have encountered chemical text or occasional molecular representations during large-scale language pretraining, such information does **not** transfer to spectrum inversion. The model clearly does not "understand" how to interpret spectra, nor can it retrieve correct structures even when molecules appear in the test set. Consequently, the high performance of SpectraLLM arises almost entirely from supervised spectrum–text alignment during fine-tuning rather than pretraining memorization.
>
> We have incorporated this discussion, along with the zero-shot results, into Appendix A.11 of the revised manuscript.

---

> ### Author Response · Authors · 2025-11-22
> **Response to Reviewer MWCJ (2/5)**
>
> ### Can the authors explain the practical use cases where their model could be readily applied?
> We thank the reviewer for raising this important question regarding practical applicability. We believe that our model can be directly deployed in laboratory and industrial environments where rapid structural annotation of small organic molecules from routine spectroscopic measurements is required. Typical use cases include:
>
> 1. **Resolving structural ambiguity in real experimental workflows.**
>    In many practical scenarios, IR or MS alone is insufficient to uniquely distinguish compounds—particularly for positional isomers, conjugated systems, or overlapping functional-group bands. Chemists routinely acquire additional spectra in such cases. Our model is designed precisely for these settings: combining IR with Raman (or IR with MS) substantially improves discriminatory power.
>
> 2. **Assisting large institutions with curating spectral repositories.**
>    Many quality-control laboratories, regulatory agencies, and metabolomics centers maintain large spectral libraries that are partially annotated or inconsistently labeled. SpectraLLM can generate preliminary SMILES annotations, build fast spectral indices, and detect mismatches between spectra and their recorded structures, thereby supporting large-scale database curation.
>
> 3. **Annotation of unknown metabolites in metabolomics.**
>    Numerous metabolomics workflows acquire MS/MS data together with IR or Raman fingerprints. In these settings, SpectraLLM can automatically annotate unknown metabolites and small-molecule biomarkers in biological fluids.
>
> 4. **Monitoring reaction intermediates and performing quality control.**
>    In automated synthesis and manufacturing environments where IR and Raman spectra are collected online, our model can verify whether intermediates match expected structures or flag unexpected by-products based on experimental spectra.
>
> 5. **High-throughput & automated compound-library screening.**
>    In high-throughput pipelines, acquiring NMR for every sample is impractical, whereas IR/Raman/MS are far cheaper and faster to obtain. In these scenarios, our model can provide rapid candidate-structure predictions, automatically filter contaminated or incorrectly collected samples, and prioritize samples that require more expensive downstream analyses (e.g., NMR or high-resolution MS).
>
> ## The evaluation may suffer from data leakage. Please elaborate on the splitting strategy and the degree of similarity between molecules in the training and testing sets. The paper currently mentions a "molecular identity”-based data split, which is known to result in leakage even when stereochemistry is ignored. Can the authors better evaluate data leakage and elaborate on the data used to retrain the baselines?
> We appreciate the reviewer's attention to the potential risk of data leakage, which is indeed a central challenge in molecular machine learning. As described in Appendix A.1 of the revised manuscript, our SMILES-identity–based splitting strategy is designed to ensure that all spectra corresponding to the same molecule—regardless of which dataset they originate from—are grouped into the same split, thereby preventing leakage of molecular identity. Concretely, we use SMILES strings as unique molecular keys and construct a mapping file (smiles2datasetIdx.csv), an excerpt of which is shown below:
>
> |SMILES|QM9s_number|MassBank_file_name|multimodal_spectroscopic_dataset_Idx|MassSpecGym_identifier|
> |-|-|-|-|-|
> |Cc1ccncn1|898|MSBNK-CASMI_2016-SM826401|124352|MassSpecGymID0068222|MassSpecGymID0068223|MassSpecGymID0068224|
>
> When a molecule has multiple spectra within a dataset, we concatenate their identifiers using `|` to ensure they are treated as a single unit during splitting. This guarantees that the training set and test set never contain different spectra of the same molecule—a key safeguard against identity leakage.
>
> We then follow the existing train/test split provided by MassSpecGym: all MassSpecGym SMILES retain their official split. The remaining SMILES not present in MassSpecGym are split at an 8:2 ratio. This ensures that MS evaluation remains fully consistent with the baselines.
>
> Regarding baseline retraining, we preprocess the split training data according to each baseline's requirements. The datasets used to train and evaluate each baseline are summarized below:
>
> |Model|Dataset|Train/Val|Test|
> |-|-|-|-|
> |IR-to-Structure|QM9s(IR-only)|105,079/11,676|13,062|
> ||QM9s(Raman-only)|105,079/11,676|13,062|
> ||QM9s(UV-only)|105,079/11,676|13,062|
> |Spectra2Structure|QM9s(IR-only)|105,079/11,676|13,062|
> ||QM9s(Raman-only)|105,079/11,676|13,062|
> ||QM9s(UV-only)|105,079/11,676|13,062|
> |NMR2Struct|Multi(NMR-only)|643,604/71,512|79,287|
>
> We did not retrain Spec2Mol and DiffMS, but instead directly used their publicly released embeddings. The rationale for this decision is detailed in our response to Comment #3.

---

> ### Author Response · Authors · 2025-11-22
> **Response to Reviewer MWCJ (1/5)**
>
> First, we sincerely thank the reviewer for taking the time to read our submission and for providing such detailed questions and suggestions. Below, we address each of your concerns in depth, and we hope our responses help clarify the motivation and utility of our work.
>
> ### Most of the collected dataset appears to contain relatively small molecules (as shown in Figure 2). Could the authors provide examples of practical use cases where annotation of such small molecules, as opposed to larger natural products or drug-like molecules, is of interest? Without some EDA of the molecular composition and an explanation of when this setting is relevant, it is difficult to assess the utility of the work. Please also elaborate in what practical scenarios the annotation of molecules simultaneously from multiple types of spectra is relevant.
>
> We thank the reviewer for this insightful question. As you suggested, we conducted detailed exploratory data analysis (EDA) on our dataset, resulting in four distribution plots (included in Appendix A.2 of the revised manuscript, corresponding to Figures 3 and 4), covering molecular weight, heavy-atom count, LogP, and functional-group distributions. The results show that the compounds in our dataset exhibit high relevance and strong structural diversity.
>
> First, the molecular-weight distribution indeed confirms that the majority of molecules lie in the range of 100–500 Da (with a peak around 250 Da), indicating that small molecules dominate the dataset. However, in practice, small molecules remain highly valuable and are routinely analyzed across several critical application domains. For example, in **metabolomics and clinical diagnostics**, most metabolites, biomarkers, and disease-associated small molecules fall below 300 Da. In **environmental and food-safety monitoring**, routine surveillance targets pesticides, plasticizers, industrial pollutants, and other organic contaminants—all small molecules. In **chemical manufacturing and materials synthesis**, structural verification of monomers, solvents, intermediates, and by-products is routinely performed using IR/Raman/MS, and these species are likewise small molecules. Moreover, most major public spectral repositories (NIST, MassBank, GNPS, QM9-related datasets, etc.) are predominantly small-molecule collections, further demonstrating the community-wide importance of this chemical space.
>
> The heavy-atom distribution (primarily 10–45) suggests good coverage of molecules with moderate complexity. LogP values cluster between 0–5, indicating a spectrum of compounds ranging from moderately hydrophilic to moderately lipophilic. The functional-group analysis shows that the dataset is dominated by molecules containing aromatic rings, ethers, ketones, and similar motifs—structures that are fundamental in drug discovery and metabolomics. The prevalence of aromatic rings, halogens, and polar groups places meaningful demands on cross-modal reasoning, reinforcing the relevance of this dataset for multimodal structural inference.
>
> The need for **multi-spectral annotation** is also driven by strong practical motivation. In routine **structure elucidation in synthetic chemistry**, multi-spectral analysis (typically IR + NMR + MS, or IR + Raman + MS) is the standardized workflow [1][2], precisely because different spectra encode complementary information: IR is highly sensitive to functional groups, Raman captures skeletal and ring modes, and MS confirms elemental composition. Additionally, many isomeric species exhibit near-identical IR or Raman signatures; combining orthogonal spectra is often the only reliable way to disambiguate such cases. In applied domains such as **metabolomics and clinical diagnostics**, multimodal spectral annotation (e.g., MS + MS/MS + IR/Raman) is now widely adopted in practical platforms—for example, GNPS's IR-assisted MS annotation pipeline and Bruker's IR–MS multimodal instruments. These workflows rely on complementary spectral evidence to distinguish isomers and assign confidence scores. In **automated reaction monitoring and high-throughput screening**, multi-spectral acquisition (MS paired with online IR/Raman/UV sensors) is increasingly common. The goal of SpectraLLM is to provide a unified multimodal prediction framework that aligns naturally with these workflows and accelerates the pipeline from multi-source spectral acquisition to final structure determination.
>
> [1] Silverstein R. M., Bassler G. C. *Spectrometric Identification of Organic Compounds*, J. Chem. Educ., 1962.
> [2] Guthrie R. D. *Introduction to Spectroscopy (Pavia, Lampman, Kriz)*, 1979.

---

> ### Author Response · Authors · 2025-11-28
> **Response to Reviewer MWCJ (More responses regarding the data leakage issue mentioned in the reply to part 2/5)**
>
> We greatly appreciate the reviewer for pointing out potential issues with data splitting in our datasets. In response, we have carefully re-examined the existing splitting strategy for the public **MassSpecGym dataset**. Our statistical analysis shows that there are **no samples in the test set with a maximum Tanimoto similarity of 1 to the training set molecules**. Additionally, the maximum similarity between the test and training sets is primarily distributed in the 0.3–0.6 range. **This pattern is consistent with the findings from the MassBank dataset**.
>
> Furthermore, because the molecules in the QM9S and multimodal_spectroscopic_dataset differ significantly from those in MassSpecGym, the partitioning of these datasets still relies heavily on random splitting after excluding the MassSpecGym molecules. We conducted a detailed check of both datasets. **In the QM9S test set, we found only 10 molecule with a maximum Tanimoto similarity of 1 to the training set, which accounts for 0.009% of the samples**. The remaining samples have similarities concentrated in the 0.5–0.7 range. **In the multimodal_spectroscopic_dataset, 90 molecules (approximately 0.012%) had a maximum similarity of 1**, while the rest were predominantly in the 0.6–0.8 range. After identifying these special cases, we removed all molecules with a maximum Tanimoto similarity of 1 from the test set and re-evaluated the results. The comparison showed negligible differences between the original and updated results.
>
> Therefore, despite not following the MCES-based splitting method used in MassSpecGym, and instead distinguishing molecules based on their 2D structure (SMILES), the structural similarity issue that is prominent in MassSpecGym and MassBank is less noticeable in QM9S and multimodal_spectroscopic_dataset due to the significant molecular differences between these datasets. Based on this analysis, we are confident that our experimental results are robust and trustworthy, and that the concerns raised by the reviewer regarding potential data leakage are unfounded.

---

### Official Review · Reviewer_61jb · 2025-11-04

**Soundness:** 3
**Presentation:** 3
**Contribution:** 3
**Rating:** 8
**Confidence:** 3

**Summary:**

The paper introduces SpectraLLM, a large language model fine-tuned via LoRA to perform end-to-end molecular structure elucidation from diverse spectroscopic inputs (IR, Raman, UV-Vis, NMR, and MS). It reformulates spectral data as natural language prompts to enable symbolic reasoning within a shared semantic space. Using datasets like QM9Spectra, Multimodal Spectroscopic, MassSpecGym, and MassBank, SpectraLLM achieves state-of-the-art unimodal and multimodal performance, outperforming domain-specific baselines across structural and functional similarity metrics.

**Strengths:**

There is novelty in this work. Specifically, it converts continuous and discrete spectra into text, enabling a unified reasoning space for multiple spectroscopy types.

There are also strong empirical results in which consistent and large gains over baselines are shown across IR, Raman, NMR, and MS, with near-perfect validity and robustness to missing modalities.

Furthermore, there is comprehensive evaluation where there are benchmarks on multiple datasets with diverse metrics (Tanimoto, MCES, Fraggle, etc.) and insightful qualitative analyses

**Weaknesses:**

While the model performs well, it remains unclear how linguistic embeddings capture physical–chemical semantics. The authors need to improve the interpretability component of their model.

There is also heavy reliance on synthetic spectra (QM9s, Multimodal Spectroscopic) may limit real-world generalization. Ultimately, these are simulated data and the authors should show model performance on real world datasets.

There is also no discussion of common failure modes, ambiguity cases, or invalid SMILES beyond aggregate metrics. The authors should provide more error analysis and better discuss the limitations of their work.

Several critical recent works are also omitted from reference in this paper:

- Le, Khiem, et al. “MolX: Enhancing Large Language Models for Molecular Learning with a Multi-Modal Extension.” KDD MLoG-GenAI 2025. (Include as a key related multimodal-LLM baseline/approach.)

- Fang, Junfeng, et al. “MolTC: Towards Molecular Relational Modeling in Language Models.” Findings of the Association for Computational Linguistics: ACL 2024, Association for Computational Linguistics, 2024, pp. 1943–1958.
ACL Anthology

- Ju, Jiaxin, et al. “Uni-MRL: Unified Multimodal Molecular Representation Learning with Large Language Models and Graph Neural Networks.” Advances in Knowledge Discovery and Data Mining (PAKDD 2025), LNCS 15874, Springer, 2025, pp. 275–287.
SpringerLink

**Questions:**

How sensitive is performance to the accuracy of the spectral-to-text mapping (ϕ)?

Could the model generalize to unseen experimental conditions or instruments?

Are there ablations showing which spectral modality contributes most to multimodal gains?

---

> ### Author Response · Authors · 2025-11-22
> **Response to Reviewer 61jb (4/4)**
>
> ### While the model performs well, it remains unclear how linguistic embeddings capture physical–chemical semantics. The authors need to improve the interpretability component of their model.
> We appreciate the reviewer's attention to interpretability. Although we acknowledge that our current work does not yet include a formal mechanism-level explanation of the model's internal representations, the existing experiments already provide proxy evidence that SpectraLLM relies on chemically meaningful semantics rather than spurious correlations. For example, motivated by your first question, we introduced controlled noise into the input spectra and observed that decoding accuracy degrades systematically and progressively with increasing perturbation. More importantly, disrupting functionally decisive spectral features—such as IR carbonyl peaks or Raman skeletal vibrations—produces far larger prediction errors than perturbing background noise. This strongly indicates that SpectraLLM depends on chemically meaningful spectral cues.
>
> In addition, we conducted several qualitative analyses: when manually removing a distinctive feature peak from a single-spectrum input (e.g., the O–H stretching band of an alcohol), the model consistently fails to recover the corresponding functional group. This provides direct evidence that the model has learned explicit correspondences between characteristic peaks and molecular substructures. Representative examples and analyses are included in Appendix A.13 of the revised manuscript.
>
> We agree that deeper interpretability remains important, and we plan to extend our analysis in future work. Specifically, we intend to incorporate gradient-based attribution methods (e.g., Grad-CAM, SHAP) and probing classifiers to quantify how internal representations encode functional groups, peak neighborhoods, and bonding patterns within the latent space.
>
> ### There is also no discussion of common failure modes, ambiguity cases, or invalid SMILES beyond aggregate metrics. The authors should provide more error analysis and better discuss the limitations of their work.
> We thank the reviewer for emphasizing the importance of deeper error analysis. In response, we have examined several representative failure cases and included a detailed discussion in Appendix A.13 of the revised manuscript. We will further expand this analysis by systematically categorizing error types—such as intrinsic spectral ambiguity, high-complexity molecular structures, and SMILES-format errors—and providing illustrative examples for each category. We also plan to qualitatively analyze the underlying sources of error, including ambiguity in spectral signatures or structural complexity that exceeds the constraints available in the observed spectra. We believe such more comprehensive analyses will help clarify the limitations of our approach and offer a clearer picture of how the model behaves under challenging scenarios.
>
> ### Several critical recent works are also omitted from reference in this paper.
> We sincerely appreciate the reviewer's suggestions regarding recent related work. After careful examination, we find that MolX, MolTC, and Uni-MRL indeed represent valuable advances in the broader direction of "LLM + molecular tasks,” focusing respectively on modality adapters, relational reasoning, and multimodal representation learning. However, none of these works address the spectroscopy-to-structure inverse problem, nor do they investigate the full pipeline of continuous spectral signals → peak extraction → textualization → generative alignment. That said, these papers are highly relevant and complementary to our direction, and they offer useful insights for future improvements in LLM-based structural interpretation. We have incorporated all three references into the Introduction of the revised manuscript and will expand our discussion of their connections to our work in the Related Work and Future Work sections.

---

> ### Author Response · Authors · 2025-11-22
> **Response to Reviewer 61jb (3/4)**
>
> ### Are there ablations showing which spectral modality contributes most to multimodal gains?
> We thank the reviewer for requesting a quantitative analysis of each modality's contribution. To address this, we conducted a comprehensive set of modality ablations. Our analysis is based on two complementary sources of evidence: (i) the results in Table 4 of the main paper (SpectraLLM trained on mixed synthetic–experimental data), and (ii) a newly added QM9S-specific modality ablation experiment (shown below).
>
> |SpectraLLM Q9MS retrained|Validity↑|Tanimoto↑|Tanimoto(MACCS)↑|Cosine↑|MCES↓|Functional Group↑|Fraggle↑|
> |-|-|-|-|-|-|-|-|
> |IR-Only|100.00%|0.1474|0.3436|0.2462|7.1245|0.5309|0.2686|
> |Raman-Only|98.90%|0.3251|0.5748|0.4490|5.4009|0.7986|0.3958|
> |UV-Vis-Only|100.00%|0.0777|0.2081|0.1407|10.3874|0.3600|0.2137|
> |IR+Raman|99.63%|0.5320|0.7363|0.6288|3.0312|0.8834|0.4806|
> |IR+UV-Vis|99.63%|0.2088|0.4471|0.3280|6.0551|0.6791|0.3333|
> |Raman+UV-Vis|99.45%|0.4594|0.6718|0.5699|4.0405|0.8431|0.4561|
> |IR+Raman+UV-Vis|98.53%|0.4053|0.6501|0.5202|3.9926|0.8370|0.4479|
>
> #### 1. Quantitative modality contributions (QM9s-specific ablation)
> The new QM9s ablation experiment clearly quantifies the contribution of every single-modality and multimodal combination (details now included in the updated manuscript):
>
> - **Baseline contribution:** Among single-modality models, Raman-Only yields the strongest performance (Tanimoto = 0.3251), substantially outperforming IR-Only (0.1474) and UV-Vis-Only (0.0777). This confirms that Raman provides the strongest standalone structural constraints (e.g., symmetry modes and carbon–skeletal vibrations).
>
> - **Strongest synergistic gain:** The IR+Raman combination achieves the highest overall Tanimoto (0.5320), representing a **~63% improvement** over the best single modality (Raman-Only, 0.3251). This demonstrates that IR (sensitive to polar functional groups) and Raman (sensitive to backbone/symmetry modes) are highly complementary.
>
> - **Marginal contribution of third modality:** The full three-modality combination IR+Raman+UV-Vis (Tanimoto = 0.4053) performs slightly worse than IR+Raman (0.5320), likely due to information redundancy or additional noise introduced by UV-Vis. Nevertheless, the three modalities provide the broadest structural coverage.
>
> #### 2. Effectiveness and complementarity of cross-modal learning (Table 4 in the main paper)
> The earlier multimodal results in Table 4 further support consistent conclusions:
>
> - **NMR as the central modality:** On the Multimodal Spectroscopic dataset, joint NMR spectra (¹³C, ¹H, HSQC) produce strong gains (Tanimoto = 0.4151), reflecting their intrinsic complementarity in encoding carbon skeletons and local chemical environments.
>
> - **MS providing the largest marginal gain:** Adding MS to Jointly NMR (yielding Jointly NMR+MS) significantly improves performance (Tanimoto ≈ 0.4518). This highlights MS as a crucial complementary modality for constraining molecular mass and fragmentation patterns.
>
> - **IR's moderate contribution:** While IR provides limited marginal gains in the Multimodal Spectroscopic setting, the final joint combination (NMR+IR+MS) achieves the best performance overall (Tanimoto ≈ 0.4875).
>
> Across both experiments, the conclusions are consistent: multimodal gains primarily arise from **orthogonal and chemically complementary information**—IR captures functional groups, Raman captures skeletal symmetry, MS constrains molecular mass, and NMR encodes atomic-level chemical shifts. IR and Raman are the most complementary pair on QM9s, while NMR combined with MS is crucial for achieving the highest structural accuracy in the full multimodal setting. All corresponding analyses and experimental details have been added to the updated PDF, in Appendix A.9.

---

> ### Author Response · Authors · 2025-11-22
> **Response to Reviewer 61jb (2/4)**
>
> ### There is also heavy reliance on synthetic spectra (QM9s, Multimodal Spectroscopic) may limit real-world generalization. Ultimately, these are simulated data and the authors should show model performance on real world datasets. Could the model generalize to unseen experimental conditions or instruments?
> In fact, the SpectraLLM model reported in our paper is trained on a mixture of both simulated (QM9S, Multimodal Spectroscopic Dataset) and experimental datasets (MassBank, MassSpecGym), as shown in Table 6 of the revised manuscript. Therefore, the results reported in Table 3 already provide a direct evaluation of real-world generalization. From Table 3, we observe that SpectraLLM achieves a Tanimoto similarity of 0.1844 on the MASS subset of the Multimodal Spectroscopic Dataset, which is notably higher than its performance on MassSpecGym (0.1533) and MassBank (0.1286). This is expected, as synthetic spectra do not contain the complex noise characteristics present in real experimental measurements. On the other hand, the performance drop from the best-case synthetic setting (Tanimoto = 0.1844) to the most challenging real dataset (MassBank, Tanimoto = 0.1286) corresponds to a relative decrease of approximately 30.2%. In contrast, the baseline model Diffms exhibits a much larger degradation of about 73.0% when evaluated on the same datasets (Tanimoto dropping from 0.3730 to 0.1007). This comparison demonstrates that, despite the inevitable trade-offs introduced by mixed supervision, SpectraLLM exhibits substantially stronger robustness to negative transfer between synthetic and real data, achieving much smoother cross-domain generalization and validating the benefit of our hybrid training strategy.
>
> Regarding whether the model can generalize to previously unseen experimental conditions or instruments, we believe the answer is yes. Variations across instruments and acquisition settings primarily manifest as differences in systematic noise patterns and spectral fidelity. Our mixed-training strategy—combining idealized spectra with high-noise real experimental data—is specifically designed to teach SpectraLLM to recognize and disentangle such heterogeneous noise. As long as the spectral modality remains consistent, the model can apply its learned core chemical–structural constraints to new experimental conditions, thereby exhibiting robust generalization.
>
> That said, we are also retraining SpectraLLM using only QM9S and the Multimodal Spectroscopic Dataset, and will evaluate it on the MassBank and MassSpecGym test sets once training is complete. This process is computationally expensive, but we will provide updated results within the next few days. Additionally, we are downloading the IR and UV subsets from NIST—another source of real experimental spectra. Since the dataset is large, this download and subsequent evaluation will also require several days; we will follow up with results as soon as they are available.

---

> ### Author Response · Authors · 2025-11-22
> **Response to Reviewer 61jb (1/4)**
>
> We sincerely appreciate your positive assessment of our work. Below, we provide detailed responses to each of your comments and outline the corresponding revisions and improvements we have made based on your feedback.
>
> ### How sensitive is performance to the accuracy of the spectral-to-text mapping (ϕ)?
> We appreciate the reviewer's question on how sensitive our results are to the accuracy of the spectral-to-text mapping ϕ. To assess the sensitivity of the mapping ϕ to the accuracy of spectrum-to-text conversion and the influence of peak extraction on final test performance, we conducted a series of analyses. We first injected synthetic noise directly into the test-set spectra to examine the robustness of our model under perturbations. Three levels of noise were applied to the IR, Raman, and UV spectra in QM9S, with the specific parameterizations reported in Table 15 of the updated PDF.
>
> We designed 21 noise-injection configurations for multispectral prompts containing IR, Raman, and UV simultaneously. These include: applying three noise intensities to only one spectrum (e.g., IR) while keeping the other two clean; applying noise to two spectra while keeping one clean; and applying the same noise level to all three spectra. After noise injection, we reconstructed the corrupted test set using a dynamic filtering threshold equal to 1% of the maximum spectral intensity. For brevity, we present three representative configurations below, while Table 16 in the updated PDF includes nine representative patterns.
>
> The robustness results clearly show that performance degradation is monotonic with respect to both noise intensity and the number of perturbed spectra. For instance, under mild noise in a single modality (e.g., noisy IR with clean Raman and UV), performance declines only marginally relative to the clean setting. As noise becomes moderate or severe—or as more modalities are perturbed—the Tanimoto similarity decreases and MCES increases accordingly. Importantly, the model avoids catastrophic failure by leveraging undisturbed orthogonal spectral information for cross-modal compensation. Moreover, even when the model fails, the increase in MCES is relatively gentle, indicating that predictions tend to preserve the core molecular scaffold rather than collapsing into random outputs. This demonstrates a predictable and structurally conservative failure mode.
>
> |SpectraLLM QM9S Jointly|Noisy|Validity↑|Tanimoto↑|Cosine↑|MCES↓|Functional Group↑|Tanimoto(MACCS)↑|Fraggle↑|
> |-|-|-|-|-|-|-|-|-|
> |I+R+U|-|98.72%|0.3355|0.4560|4.9647|0.7934|0.5785|0.4117|
> |I*+R+U|mild|96.89%|0.3365|0.4659|5.3299|0.7819|0.5955|0.4011|
> ||moderate|97.99%|0.2688|0.3963|6.2112|0.6851|0.5212|0.3598|
> ||severe|87.18%|0.1712|0.2799|8.4433|0.5010|0.3774|0.2556|
> |I*+R*+U|mild|96.34%|0.2163|0.3398|7.0894|0.6063|0.4592|0.3115|
> ||moderate|89.56%|0.1546|0.2607|8.4029|0.4828|0.3653|0.2655|
> ||severe|57.88%|0.0749|0.1372|10.1472|0.3348|0.2466|0.1663|
> |I*+R*+U*|mild|96.70%|0.2205|0.3473|6.9820|0.6272|0.4676|0.3154|
> ||moderate|87.91%|0.1527|0.2556|8.4740|0.4659|0.3505|0.2548|
> ||severe|39.01%|0.0792|0.1427|10.2981|0.3342|0.2508|0.1718|
>
> We also evaluated how the peak-filtering threshold affects model performance. On QM9S, we applied threshold values of [0%, 1%, 2%, 5%, 10%, 20%] relative to the maximum peak intensity, re-fine-tuned SpectraLLM on each filtered dataset, and re-evaluated it. The results are shown below.
>
> |SpectraLLM QM9S Jointly|Validity↑|Tanimoto↑|Cosine↑|MCES↓|Functional Group↑|Tanimoto(MACCS)↑|Fraggle↑|
> |-|-|-|-|-|-|-|-|
> |0%|99.45%|0.4149|0.5388|4.36|0.8743|0.6769|0.4696|
> |**1%**|**99.82%**|0.4404|0.5613|4.1606|**0.8754**|**0.6978**|**0.4615**|
> |2%|99.63%|**0.4411**|**0.5622**|**3.9908**|0.8744|0.6917|0.4568|
> |5%|99.08%|0.4140|0.5357|4.3983|0.8650|0.6748|0.4495|
> |10%|99.08%|0.4071|0.5300|4.4732|0.8600|0.6683|0.4505|
> |20%|98.90%|0.3690|0.4988|4.7213|0.8513|0.6366|0.4441|
>
> Our experiments show that a dynamic threshold of 1% yields the strongest overall performance (Tanimoto = 0.4404), confirming that the 1% setting strikes the best balance between preserving chemically informative peaks and suppressing background noise. A more detailed analysis is provided in Appendix A.7 of the updated PDF.

---

### Author Response · Authors · 2025-11-28
**Response to All Reviewers (1/2)**

We thank all reviewers for their thoughtful feedback and in-depth engagement with our work.  A recurring concern across multiple reviews is the generalization capability of SpectraLLM to real-world experimental spectra.  To address this, we conducted two additional evaluations specifically designed to assess out-of-distribution robustness on purely experimental data. Now that the experimental results are available, we are providing a unified response here and have included the discussion results in Appendix A.14.

In the first study, we removed all MassSpecGym and MassBank spectra from the training corpus, training SpectraLLM exclusively on synthetic spectra. We then evaluated the resulting model on the held-out MassSpecGym and MassBank test sets, alongside several baselines. As summarized in the table below, SpectraLLM retains strong generalization despite having no exposure to these datasets during training. On MassSpecGym, it surpasses Spec2Mol across all metrics, while maintaining markedly higher chemical validity. On MassBank, SpectraLLM continues to exhibit substantial improvements over Spec2Mol in both structural and fragment-level metrics.

|Setting|State|Model|Validity↑|Tanimoto↑|Cosine↑|Functional Group↑|Tanimoto (MACCS)↑|Fraggle↑|
|-|-|-|-|-|-|-|-|-|
|MassSpecGym|trained|Diffms|57.16%|0.159705|0.242187|0.489004|0.430529|0.353878|
|||SpectraLLM|99.74%|0.1537|0.2565|0.5016|0.4735|0.362|
||untrained|Spec2Mol|62.86%|0.084861|0.151115|0.311157|0.270927|0.206558|
|||SpectraLLM|92.97%|0.1139|0.2039|0.4354|0.4005|0.2988|
|MassBank|trained|Diffms|23.63%|0.074269|0.100735|0.179588|0.156596|0.153949|
|||SpectraLLM|98.46%|0.1306|0.2264|0.454|0.3848|0.3201|
||untrained|Spec2Mol|71.63%|0.085741|0.153861|0.299915|0.241751|0.214962|
|||SpectraLLM|92.79%|0.1125|0.2019|0.187|0.3612|0.2922|

In the second experiment, we further examined cross-domain generalization on the NIST experimental collection. To eliminate any possibility of data leakage or memorization, we removed from NIST all molecules that either appeared in the training set or exhibited high structural similarity to training molecules based on MCES distance. This filtering resulted in 4,024 molecule–IR pairs and 911 molecule–UV–Vis pairs that share no close structural analogues with the training corpus. We then evaluated SpectraLLM and all baselines in a strict zero-shot setting. As shown in the table below, SpectraLLM maintains substantially stronger structural recovery than IR-to-Structure and Spectra2Structure across both modalities.

|Datasets|Model|Validity↑|Tanimoto↑|Cosine↑|MCES↓|Functional Group↑|Tanimoto (MACCS)↑|Fraggle↑|
|-|-|-|-|-|-|-|-|-|
|NIST-ir|IR-to-Structure|98.02%|0.0095|0.0657|33.4585|0.1309|0.0699|0.0720|
||Spectra2Structure|99.31%|0.0128|0.0769|29.5700|0.2104|0.0923|0.1267|
||SpectraLLM|99.43%|0.0727|0.1368|22.3229|0.3191|0.1964|0.2176|
|NIST-uv|IR-to-Structure|99.12%|0.0673|0.1147|25.9537|0.2799|0.1529|0.1101|
||Spectra2Structure|99.37%|0.0726|0.1285|23.3650|0.3058|0.1989|0.1407|
||SpectraLLM|99.56%|0.0744|0.1381|23.0612|0.3231|0.2084|0.1411|

Taken together, the two experiments collectively demonstrate that SpectraLLM does not rely on memorizing dataset-specific fingerprints but instead learns an abstraction of spectral peaks that transfers robustly across experimental domains, instruments, and spectral modalities. Even when trained solely on synthetic spectra or evaluated on molecules explicitly dissimilar to its training set, SpectraLLM preserves high chemical validity and returns structurally meaningful predictions. We hope these results address the reviewers’ concerns regarding real-world applicability and further clarify the generality of the proposed peak-language modeling and LLM-based reasoning framework.

---

> ### Author Response · Authors · 2025-12-03
> **Response to All Reviewers (2/2)**
>
> We additionally provide a consolidated summary of all revisions made in the manuscript:
> - In the main paper,
>    - we have incorporated three new references in the Introduction to better situate our work within the recent literature;
>    - clarified the definition of $\phi$ at line 176 in Section 2.2.2;
>    - and expanded the caption of Figure 2 to include explicit explanations of the visual symbols.
> - In the appendix,
>    - we added a more fine-grained EDA analysis of all datasets (Appendix A.2);
>    - defined the symbols (A) and (B) in the cosine similarity formulation (Appendix A.3);
>    - added details of training and testing to the baseline model (Appendix A.4);
>    - reported a grid search over decoding parameters used by generative models (Appendix A.6);
>    - provided experiments on the effects of peak-extraction thresholds and peak widths (Appendix A.7);
>    - analyzed the influence of background noise on peak extraction and model robustness (Appendix A.8);
>    - extended our ablations on multi-spectra interaction gains (Appendix A.9);
>    - examined the role of contextual metadata such as experiment descriptions (Appendix A.10);
>    - assessed pretrained LLMs’ prior familiarity with molecules (Appendix A.11);
>    - added a detailed efficiency analysis for both training and inference (Appendix A.12);
>    - included bad-case analyses (Appendix A.13);
>    - and reported new experiments evaluating generalization to real experimental spectra (Appendix A.14).
>
> All newly added or revised content is highlighted in blue in the revised version. We sincerely thank all reviewers and the area chair for their time, feedback, and careful evaluation of our work, and we hope these clarifications and additions help strengthen your confidence in the contributions of this paper.

---

### Author Response · Authors · 2025-12-03
**Summary of Rebuttal Updates and Reviewer Interactions (1/3)**

Dear Area Chair,

We sincerely thank you for the tremendous effort you have devoted to upholding the academic integrity and fairness of this conference. In such an exceptional period, your careful review of our rebuttal work has been an invaluable source of encouragement for us. We are also deeply grateful to the reviewers for their thoughtful, constructive, and academically rigorous feedback, which significantly strengthened the clarity and contribution of our work.

We have consistently adhered to ICLR's double-blind policy and engaged in active, substantive academic discussions with the reviewers throughout the rebuttal period. In light of the system rollback to the pre-discussion state, we would like to provide a factual summary of the substantial progress made during the discussion period (prior to the Nov 27th announcement).

We would like to note that, while we provided detailed responses and additional experiments to address the concerns of four reviewers (**61jb, MWCJ, myQs, m8ht**), we **did not receive further replies** from them during the discussion period before the system rollback.

To assist your quick assessment, we provide the following summary organized by reviewer timeline, key thematic resolutions, and specific paper revisions.

### 1. Timeline of Reviewer Interactions

|Reviewer|Score|Summary of the review and discussion|
|-|-|-|
|61jb|8|This reviewer expressed strong enthusiasm for the paper and explicitly supported acceptance, viewing SpectraLLM as **a novel and impactful direction with strong empirical results (near-perfect validity and robustness to missing modalities) and comprehensive evaluation (benchmarks on multiple datasets with diverse metrics and insightful qualitative analyses)**. The comments focused on constructive suggestions for further strengthening the work. In response, we expanded our discussion of spectrum-to-text interpretability, added detailed failure/ambiguity case analyses, and incorporated more recent related work. We also conducted extensive ablation studies clarifying the source of multi-spectrum gains, the sensitivity of the spectrum–text mapping, and the model's ability to generalize under unknown experimental conditions. All questions were fully addressed, though no further response was received.|
|MWCJ|4|This reviewer acknowledged **the importance of the problem (remains very challenging to solve) and found our dataset design (leverage Fraggle similarity from RDKit, which was not explored in related work before) and writing clear**, while expressing concerns regarding the rigor of the baseline comparisons and dataset preparation. We clarified that baselines were retrained under strictly fair settings, that our dataset splitting builds directly on the official MassSpecGym partitions with additional processing, and that we performed a thorough EDA to rule out any possibility of data leakage. We also added extensive experiments analyzing spectrum–text sensitivity, robustness to noise, the inability of untrained LLMs to perform spectrum–structure reasoning, and the contribution of experimental metadata. Each issue was fully addressed, though no further reply was received.|
|myQs|4|This reviewer **found the unified “spectrum → text token → LLM reasoning” architecture simple, extensible (straightforward to reproduce in principle), and empirically convincing (multiple datasets and metrics provide a more realistic view of structure recovery)**, but questioned whether the contribution goes beyond incremental improvement and requested efficiency/robustness reporting. We added detailed compute-efficiency analyses and a series of new experiments on tokenization sensitivity, transfer to real experimental spectra, the effect of metadata, and decoding-strategy sensitivity. All concerns were fully answered, though no follow-up response was received.|
|m8ht|4|This reviewer agreed that the method is **novel, experiments comprehensive (Appendix A.4), and performance strong (Table 2)**, but requested deeper analysis on peak-extraction robustness, the role of metadata, synthetic→real transfer, and clearer articulation of the task bottlenecks. We conducted all requested experiments and added a multi-perspective bad-case analysis identifying the dominant failure modes. All questions were thoroughly addressed, though no follow-up was received.|

---

> ### Author Response · Authors · 2025-12-03
> **Summary of Rebuttal Updates and Reviewer Interactions (2/3)**
>
> ### 2. Key Concerns Resolved via Clarification
>
> In our detailed responses, we resolved all individual inquiries. Below, we distill the resolution of the most critical and common topics shared by multiple reviewers:
> - **Generalization to real experimental conditions (raised by 61jb, myQs, m8ht)**: **Several reviewers** emphasized the importance of evaluating SpectraLLM under real laboratory conditions, rather than only on synthetic spectra. To address this, we added two new evaluations in Appendix A.14, using MassSpecGym, MassBank, and NIST experimental spectra for strict zero-shot testing. These results demonstrate that SpectraLLM maintains strong structural reasoning ability even when confronted with domain shifts in noise profiles, peak intensities, and acquisition parameters, thereby resolving the reviewers' concerns regarding real-world transferability.
> - **Model robustness and stability (raised by 61jb, myQs, m8ht)**: **All three reviewers** expressed strong interest in understanding how much SpectraLLM's performance depends on the upstream peak extraction process. In Appendix A.7, we systematically vary peak filtering thresholds and peak-width handling, directly addressing **Reviewer myQs's** question about sensitivity to peak detection hyperparameters. Appendix A.8 further evaluates robustness under increased spectral noise, providing concrete evidence for **Reviewer m8ht's** concerns on noise-induced degradation. These analyses jointly resolve **Reviewer 61jb's** broader question on the stability of our spectrum-to-text encoding pipeline. Additionally, multiple reviewers asked how much the experimental metadata (e.g., instrument descriptions, acquisition settings) actually contribute to model performance. The new ablation in Appendix A.10 directly isolates the effect of metadata and shows its measurable contribution, addressing this shared concern.
> - **Application relevance and practical significance (raised by MWCJ, myQs)**: **Reviewer MWCJ** questioned whether the task setting truly aligns with real analytical workflows. We addressed this through an expanded EDA (Appendix A.2), showing that the dataset indeed concentrates on small molecules—an area widely recognized as central in metabolomics, environmental analysis, and pharmaceutical research. Furthermore, we clarified that multi-spectral annotation and joint reasoning are already standard practice for chemists in real-world structural elucidation, thereby resolving doubts about the applicability of our problem formulation. For **Reviewer myQs's** concern regarding novelty, we clarified that the key contribution of SpectraLLM is conceptual: demonstrating that a single text-only interface can jointly reason over heterogeneous spectral modalities and map them directly to molecular structures without any modality-specific encoders. This establishes the feasibility of purely language-based multimodal scientific reasoning, thereby addressing concerns about the originality of our modeling framework.
> - **Experimental rigor and reproducibility (raised by MWCJ, myQs)**: Both reviewers sought assurance that the experiments were conducted rigorously and fairly. In Appendices A.1 and A.4, we expanded our explanation of dataset construction and baseline training, clarifying that our split builds directly upon the official MassSpecGym partition and confirming no data leakage issues. Appendix A.2 provides a detailed EDA that further strengthens transparency and dataset integrity. Finally, in the Reproducibility Statement, we explicitly commit to releasing all code, processed data, and model checkpoints upon publication, addressing **Reviewer myQs's** remaining concerns on reproducibility.

---

> ### Author Response · Authors · 2025-12-03
> **Summary of Rebuttal Updates and Reviewer Interactions (3/3)**
>
> ### 3. Summary of Key Revisions in the Uploaded PDF
> We have uploaded a revised paper incorporating extensive new experiments and analyses to address reviewer feedback. The 9 major updates are:
>
> - **Generalization to Real Experimental Spectra (New App. A.14)**: Across both MassSpecGym and MassBank/NIST zero-shot evaluations, SpectraLLM consistently maintains strong performance, demonstrating robust generalization to real experimental noise, unseen acquisition conditions, and molecular scaffolds absent from synthetic data.
> - **Robustness and Stability of Peak Selection (New App. A.7 & A.8 & A.10)**: Across all peak-selection and noise-level ablations, SpectraLLM remains stable under realistic spectral perturbations: optimal peak-intensity thresholds are essential for preserving the spectrum–structure mapping, degradation under corruption is smooth and predictable rather than catastrophic, and natural-language experimental metadata further strengthens multi-spectral alignment—together confirming that the model's performance boundary is shaped by coordinated disruptions across modalities rather than any single preprocessing factor.
> - **Bad Case Study (New App. A.13)**: The examined failure modes show that errors arise from a mix of intrinsic spectral ambiguity, limited scaffold coverage, and decoding imperfections—clarifying why metrics such as Tanimoto remain modest in challenging regimes and highlighting where future modeling and data augmentation efforts should focus.
> - **Modality Contributions and Multispectral Synergy (Revised Sec. 3.3 & New App. A.9)**: We find that single-spectrum training overfits to spectrum-specific cues and generalizes poorly across spectra, whereas multi-spectral training significantly improves robustness, preserving structure recovery under diverse spectral combinations and even missing-spectrum inputs.
> - **Exploration of Decoding Strategies (New App. A.6)**: Across all decoding configurations, the experiments reveal a consistent trade-off between diversity and correctness, and our final decoding setup provides the best balance—demonstrating that tailored decoding design is essential for reliable spectrum-to-structure generation.
> - **Zero-Shot Limitations of Pretrained LLMs on Spectral Interpretation (New App. A.11)**: Zero-shot evaluations confirm that pretrained LLMs do not understand spectral signals and cannot perform spectrum-to-structure inversion, indicating that SpectraLLM's gains come from dedicated spectrum–text alignment rather than chemical knowledge memorized during pretraining.
> - **Additional Details on Dataset Usage and Baseline Training(Revised App. A.1 & New App. A.2 & A.4)**: We clarified the construction and partitioning of all datasets, added full baseline training specifications, and conducted a comprehensive EDA to ensure that our evaluation is reproducible, fair, and free from unintended data leakage.
> - **Training and Inference Efficiency (New App. A.12)**: Despite higher computational costs than spectrum-specific baselines, SpectraLLM delivers significantly stronger cross-spectral generality and structural accuracy, allowing practitioners to trade off efficiency and performance based on application needs.
> - **Updated Conclusion (Revised Con.)**: Re-emphasizes our contributions and clarifies the scientific novelty.
>
> We hope this comprehensive summary assists you in evaluating the true state of our paper and the consensus forming among reviewers before the interruption. We remain available to answer any further questions you may have.
>
> Best regards,
>
> The Authors

---

### Meta-Review · Area_Chair_sjzF · 2026-01-17

**Summary:**

This paper proposes SpectraLLM, a unified “spectra → textual tokens → LLM” framework for molecular structure annotation from heterogeneous spectroscopy modalities (IR/Raman/UV/NMR/MS) via parameter-efficient tuning. The central value is a single tokenized interface that enables a consistent end-to-end pipeline for joint multi-spectra reasoning and structure generation (SMILES), including settings with missing modalities. The submission evaluates on multiple datasets and configurations, reporting structure-recovery metrics and near-perfect validity, and shows strong gains over baselines.

The rebuttal adds substantial evidence and clarifications that materially strengthen the credibility of the claims: (i) systematic sensitivity analyses over peak-selection thresholds and controlled noise injections, yielding interpretable degradation trends and stable recommended settings; (ii) modality contribution / synergy ablations that clarify which spectra drive improvements; (iii) decoding strategy comparisons (greedy vs stochastic/beam variants) that justify the reported generation protocol; (iv) experimental-metadata ablations (e.g., collision energy) demonstrating consistent benefits from context conditioning; (v) transparent reporting of training/inference compute and throughput, clarifying practical applicability.

Remaining concerns are mainly: (a) baseline fairness/comparability—some strong baselines are not retrained under the same data regime or budget, leaving room for disagreement on strict apples-to-apples comparisons; and (b) the most stringent sim-only→experimental transfer evaluation is still described as ongoing rather than fully closed. Overall, the added empirical evidence supports the paper’s main claims about a unified token interface and multi-spectra synergy; I lean toward acceptance as a poster, while recommending that the camera-ready carefully calibrate SOTA statements and improve transfer and reproducibility details.

**Reviewer Concerns:**

Concerns largely addressed by the rebuttal/revision:
1．Sensitivity/robustness of peak extraction and tokenization: The authors add systematic threshold sweeps and multi-level noise injections, showing interpretable degradation trends and a stable recommended setting, mitigating concerns that conclusions hinge on brittle upstream engineering.
2．Decoding strategy and evaluation protocol: Additional comparisons (greedy vs sampling/beam variants) justify the generation protocol for structure recovery and help explain differences against some benchmark reporting conventions.
3．Contribution of experimental metadata (e.g., collision energy): A “remove metadata” ablation (with retraining/evaluation) yields consistent drops, supporting that metadata conditioning is meaningful rather than cosmetic.

Concerns partially addressed / still outstanding:
4．Baseline fairness/comparability: Some strong baselines are not retrained under the same data regime/budget. The rebuttal argues retraining is costly and attempts to reduce bias via evaluation filtering and protocol clarification, which may not fully satisfy a strict apples-to-apples standard.
5．Closed-loop sim-only→experimental transfer: While robustness analyses and experimental datasets are included, the most stringent sim-only training → experimental-only testing transfer is still described as ongoing, leaving residual uncertainty about pure domain transfer.
6．Novelty framing: Some reviewers may still view the core as an incremental combination (tokenize spectra + tune an LLM). The rebuttal strengthens the conceptual framing (unified token interface and multi-spectra synergy), but subjectivity may remain.

**Reviewer Scores:**

Reviewer 61jb (original 8): Likely stays at 8 (possibly 9). The added modality ablations, robustness studies, and expanded analyses directly strengthen their main points without introducing new methodological concerns.
Reviewer myQs (original 4): Likely increases to 5. The rebuttal provides the requested sensitivity analyses, decoding comparisons, compute reporting, and more chemically grounded evaluations; novelty may still be viewed as incremental, limiting a larger jump.
Reviewer m8ht (original 4): Likely increases to 5. Quantified peak-threshold/noise robustness and metadata ablations substantially address the core concerns; remaining hesitation may be tied to the incomplete sim-only→experimental transfer closure.
Reviewer MWCJ (original 4): Likely, 4 → 4/5. Leakage/memorization concerns are reduced, but the lack of retrained baselines under matched budgets may remain decisive depending on their strictness about fairness/comparability.

---

### Decision · Program_Chairs · 2026-01-26

Accept (Poster)